Foreflipper and hindflipper muscle reconstructions of Cryptoclidus eurymerus in comparison to functional analogues: introduction of a myological mechanism for flipper twisting

Krahl Anna 1 2 3 anna.krahl@uni-tuebingen.de
Witzel Ulrich 1
1 Biomechanics Research Group, Lehrstuhl für Produktentwicklung, Faculty of Mechanical Engineering, Ruhr-Universität Bochum , Bochum , Germany
2 Section of Paleontology, Institute of Geoscience, Rheinische Friedrich-Wilhelms-Universität Bonn , Bonn , Germany
3 Paläontologische Sammlung, Fachbereich Geowissenschaften, Eberhard-Karls-Universität Tübingen , Tübingen , Germany
Hutchinson John
Electronic publication date: 2021 Dec 15
Publication date: 2021
Volume: 9
Electronic Location ID: e12537
Received 2021 Jan 21; Accepted 2021 Nov 3
Copyright: © 2021 Krahl and Witzel
Copyright year: 2021
Copyright holder: Krahl and Witzel
License: This is an open access article distributed under the terms of the Creative Commons Attribution License, which permits unrestricted use, distribution, reproduction and adaptation in any medium and for any purpose provided that it is properly attributed. For attribution, the original author(s), title, publication source (PeerJ) and either DOI or URL of the article must be cited.
License URL: https://creativecommons.org/licenses/by/4.0/

Keywords: Muscle reconstructions, Extant phylogenetic bracket, Flipper twisting, Underwater flight, Plesiosaur, Cryptoclidus eurymerus, Flipper beat cycle

Funding: Deutsche Forschungsgemeinschaft (DFG) WI1389/8-1 This work was funded by the Deutsche Forschungsgemeinschaft (DFG grant WI1389/8-1). The funders had no role in study design, data collection and analysis, decision to publish, or preparation of the manuscript.

==============================
Background

Plesiosaurs, diapsid crown-group Sauropterygia, inhabited the oceans from the Late Triassic to the Late Cretaceous. Their most exceptional characteristic are four hydrofoil-like flippers. The question whether plesiosaurs employed their four flippers in underwater flight, rowing flight, or rowing has not been settled yet. Plesiosaur locomotory muscles have been reconstructed in the past, but neither the pelvic muscles nor the distal fore- and hindflipper musculature have been reconstructed entirely.

Methods

All plesiosaur locomotory muscles were reconstructed in order to find out whether it is possible to identify muscles that are necessary for underwater flight including those that enable flipper rotation and twisting. Flipper twisting has been proven by hydrodynamic studies to be necessary for efficient underwater flight. So, Cryptoclidus eurymerus fore- and hindflipper muscles and ligaments were reconstructed using the extant phylogenetic bracket (Testudines, Crocodylia, and Lepidosauria) and correlated with osteological features and checked for their functionality. Muscle functions were geometrically derived in relation to the glenoid and acetabulum position. Additionally, myology of functionally analogous Chelonioidea, Spheniscidae, Otariinae, and Cetacea is used to extract general myological adaptations of secondary aquatic tetrapods to inform the phylogenetically inferred muscle reconstructions.

Results

A total of 52 plesiosaur fore- and hindflipper muscles were reconstructed. Amongst these are flipper depressors, elevators, retractors, protractors, and rotators. These muscles enable a fore- and hindflipper downstroke and upstroke, the two sequences that represent an underwater flight flipper beat cycle. Additionally, other muscles were capable of twisting fore- and hindflippers along their length axis during down- and upstroke accordingly. A combination of these muscles that actively aid in flipper twisting and intermetacarpal/intermetatarsal and metacarpodigital/metatarsodigital ligament systems, that passively engage the successive digits, could have accomplished fore-and hindflipper length axis twisting in plesiosaurs that is essential for underwater flight. Furthermore, five muscles that could possibly actively adjust the flipper profiles for efficient underwater flight were found, too.

Introduction

Within Diapsida, Sauropterygia are either placed on the archosauromorph (Merck, 1997) (Fig. 1B) or lepidosauromorph lineage (Rieppel & Reisz, 1999) (Fig. 1C), or form a sister-group to both (Neenan, Klein & Scheyer, 2013). Sauropterygia comprise the Triassic Placodontia, Pachypleurosauria, Nothosauroidea, and the Pistosauroidea from which the Plesiosauria emerge in the Late Triassic (Wintrich et al., 2017). Plesiosaurs diversified throughout the Jurassic and Cretaceous and died out by the end of the Cretaceous (Bardet, 1994; Motani, 2009; Vincent et al., 2011; Vincent et al., 2013). All sauropterygians are secondary aquatic tetrapods (Neenan, Klein & Scheyer, 2013).

Figure 1 Cryptoclidus eurymerus (IGPB R 324) mounting at the Goldfuß Museum, Section of Paleontology, Institute of Geosciences, Rheinische Friedrich-Wilhelms-Universität Bonn, Germany and extant phylogenetic bracket of Sauropterygia.

(A) Overview over the skeleton that has been remounted in 2013 to depict underwater flight. The skeleton is mostly complete except for large parts of the skull which are made of plaster (picture by G. Oleschinski). (B) Extant phylogenetic bracket of Plesiosauria, if Sauropterygia are early Archosauromorpha based on Rieppel & Reisz (1999), amended are Testudines as the sister-group of Crocodylia (Pereira et al., 2017). (C) Extant phylogenetic bracket of Plesiosauria, if Sauropterygia are early Lepidosauromorpha (including Sphenodontia and Squamata) based on Merck (1997), amended are Testudines as the sister-group of Crocodylia (Pereira et al., 2017).

The most unique character of plesiosaurs is the highly derived locomotory system. The pectoral and pelvic girdle is formed by much expanded, ventrally, flat-lying bones (scapula, coracoid, pubis, ischium). The dorsally projecting scapular blade and the ilium are greatly reduced in size in comparison to early Sauropterygia (Krahl, 2021 for review). Plesiosaurs have four very similarly shaped flippers which taper from the base to the flipper tip and form a hydrofoil (Robinson, 1975, 1977) with an asymmetrical flipper profile (Robinson, 1975; Caldwell, 1997; Fig. 1A).

Functionally comparable to plesiosaurs are the convergently evolved Chelonioidea, Spheniscidae, Otariinae, and Cetacea which have evolved lift-producing hydrofoil-like foreflippers (Walker, 1971; Davenport, Munks & Oxford, 1984; Wyneken, 1997; Rivera, Wyneken & Blob, 2011; Rivera, Rivera & Blob, 2013; Neu, 1931; Clark & Bemis, 1979; Miklosovic et al., 2004; Weber et al., 2009; Weber et al., 2014; English, 1976b; Feldkamp, 1987). The flipper profiles are asymmetrical in Chelonioidea and Spheniscidae and symmetrical in Otariinae and Cetacea (Fish & Battle, 1995; Fish, 2004). Chelonioidea and Spheniscidae employ underwater flight (Walker, 1971; Davenport, Munks & Oxford, 1984; Wyneken, 1997; Rivera, Wyneken & Blob, 2011; Rivera, Rivera & Blob, 2013; Neu, 1931; Clark & Bemis, 1979). Contrastingly, Cetacea mainly propel themselves by caudal oscillation of the fluke (Fish, 1996; Woodward, Winn & Fish, 2006) while the foreflippers act in maneuvering (Fish, 2002; Woodward, Winn & Fish, 2006). Otariinae evolved a swimming style which is termed rowing-flight, in which large lift-based elements of true underwater flight and drag-based elements from rowing are combined. The symmetrical hydrofoil-like foreflippers of sea lions show specialized adaptations for this and provide the main propulsion, while the hindlegs act as control surfaces (English, 1976b; Feldkamp, 1987) and aid in terrestrial locomotion (Berta, Sumich & Kovacs, 2005).

How plesiosaurs swam and how the two flipper pairs were moved in relation to each other has remained debated for over two centuries. It has been suggested that the flippers were used for rowing (Williston, 1914; Tarlo, 1958; Araújo & Correia, 2015; Araújo et al., 2015), underwater flight (Robinson, 1975, 1977; Lingham-Soliar, 2000; Carpenter et al., 2010; Liu et al., 2015; Muscutt et al., 2017; Krahl, 2021) or in a combination of both styles (Godfrey, 1984; Lingham-Soliar, 2000; Liu et al., 2015; Krahl, 2021).

Generally, during rowing, the main plane of flipper movement is anteroposterior. In underwater flight, the main direction of movement is dorsoventrally (Rivera, Rivera & Blob, 2013). These different movement patterns are due to different hydrodynamic mechanisms: drag-based (rowing) and lift-based propulsion (underwater flight/flapping) (see Krahl, 2021 for review). Otariinae and Carettochelys insculpta highlight that underwater flight vs. rowing is a false dichotomy. Instead, underwater flight and rowing span a spectrum of locomotory modes and sea lions and Carettochelys insculpta fall in intermediate positions. They both use water drag and lift and show phases of underwater flight and rowing in their flipper beat cycle (Feldkamp, 1987; Rivera, Rivera & Blob, 2013; Krahl, 2021). Sea lions have a proportionally longer flight phase than Carettochelys insculpta (Rivera, Rivera & Blob, 2013; Krahl, 2021).

Krahl (2021) concluded that based on the bone and joint morphology of plesiosaurs it is most likely that plesiosaurs were underwater fliers agreeing with e.g., the hydrodynamic experiments of Muscutt et al. (2017). Muscutt et al. (2017) showed that in phase or slightly out of phase beating of the fore- and hindflippers in an underwater flight flipper beat cycle is highly efficient in plesiosaurs. Contrastingly, asynchronous flipper beating is significantly less efficient. Krahl (2021) emphasizes that flipper twisting, additionally to flipper rotation, briefly mentioned by Robinson (1975) and Liu et al. (2015) has been largely neglected in studies on locomotion in plesiosaurs. Nevertheless, flipper twisting is an essential component for efficient lift-based underwater flight in vertebrates (Davenport, 1987; Walker & Westneat, 2000; Walker & Westneat, 2002) including plesiosaurs (Witzel, Krahl & Sander, 2015; Witzel, 2020).

The aim of this study is to examine whether it is possible to reconstruct locomotory muscles for a plesiosaur (Fig. 1A) which can perform an underwater flight flipper beat cycle, including flipper rotation and flipper length axis twisting. Flipper twisting has been found to be crucial for underwater flight by hydrodynamical studies. The extant phylogenetic bracket (EPB) (Testudines, Crocodylia, and Lepidosauria) (Figs. 1B and 1C) provides a sound phylogenetic inference for all reconstructed plesiosaur muscles (Figs. 2A, 2B, 3A and 3B). The extant groups chosen for the EPB are mostly functionally different to plesiosaurs (terrestrial locomotion (lepidosaurs, tortoises, crocodiles), rowing (turtles), laterally undulatory swimming (crocodiles). Therefore, functional analogues to plesiosaurs (Chelonioidea, Spheniscidae, Otariinae, and Cetacea) are chosen that (largely) rely on lift-based propulsion to help identify myological characters that are common amongst highly aquatic underwater flying secondarily aquatic Tetrapoda. Further, Cryptoclidus eurymerus (IGPB R 324) fore- and hindflipper muscles are assigned to osteological correlates. Muscle functions are obtained geometrically, i.e., by their relative arrangement in relation to the glenoid in the foreflipper and to the acetabulum in the hindflipper. This resulted in the reconstruction of 52 plesiosaur fore- and hindflipper muscles. Humeral and femoral depressors, elevators, retractors, protractors, and rotators were identified that were able to power underwater flight. Further, muscles were found that twist the fore- and hindflipper along its length axis. Six muscles were found to be possibly responsible for actively inducing asymmetry, i.e., cambered flipper profiles, which would have increased the efficiency of underwater flight in plesiosaurs (Figs. 4A, 4B, 5A and 5B; Tables 1 and 2).

Figure 2 Cryptoclidus eurymerus (IGPB R 324) foreflipper muscle attachment sites.

Muscle attachment sites in (A) ventral and (B) dorsal view. Dotted lines and ?, illustrate muscle attachment areas that are not as well supported by the EPB as the ones marked with solid lines, but make sense from a functional perspective. Black stars mark muscles that are associated with osteological correlates. Abbreviations: c, coracoid; cl, clavicular remains; h, humerus; int, intermedium; r, radius; rad, radiale; s, scapula; u, ulna; uln, ulnare.

Figure 3 Cryptoclidus eurymerus (IGPB R 324) hindflipper muscle attachment sites.

Muscle attachment sites in (A) ventral and (B) dorsal view. Dotted lines and ?, illustrate muscle attachment areas that are not as well supported by the EPB as by the solid lines, but make sense from a functional perspective. Black stars mark muscles that are associated with osteological correlates. Abbreviations: f, femur; fi, fibula; fib, fibulare; il, ilium; is, ischium; int, intermedium; p, pubis; t, tibia; tib, tibiale.

Figure 4 Muscle functions, lines of action, and the myological flipper twisting mechanism of the foreflipper of Cryptoclidus eurymerus (IGPB R 324).

Pectoral limb in (A) ventral and (B) dorsal view. (A) Muscles originating from the ventral pectoral girdle function as flipper depressors. blue, muscles that twist the flipper leading edge downwards during foreflipper downstroke. dark green, muscles that twist the flipper trailing edge downwards during upstroke. (B) muscles originating from the dorsal pectoral girdle/vertebral column are humeral elevators. blue, muscles that twist the flipper leading edge upwards during upstroke. dark green, muscles that twist the flipper trailing edge upwards during the downstroke. (A) and (B) black dashed line (~ in humerus long axis direction) marks boundary for humeral protractors/retractors (anterior/posterior to it). Edc and fdlf/fdls have an aponeurosis, preventing individual digital flexion, which is schematically represented by a line crossing the hand. Some muscle lines of action are represented with a kink so that they are closely associated with the bones which is the normal condition for extant tetrapod taxa. The posterior line of action of the ld originates outside of a bony area, i.e., from the vertebral column. Abbreviations of bones in italic: c, coracoid; cl, clavicular remains; h, humerus; int, intermedium; r, radius; rad, radiale; s, scapula; u, ulna; uln, ulnare. Abbreviations of muscles: abdV, Musculus abductor digiti V; adm, Musculus adductor digiti minimi; apb, Musculus abductor pollicis brevis; b, Musculus brachialis; bb, Musculus biceps brachii; cb, Musculus coracobrachialis brevis; cl, Musculus coracobrachialis longus; dc, Musculus deltoideus clavicularis; ds, Musculus deltoideus scapularis; ecu, Musculus extensor carpi ulnaris; edbp, Musculi extensores digitores breves profundi; edbs, Musculi extensores digitores breves superficialis; edc, Musculus extensor digitorum communis; fcr, Musculus flexor carpi radialis; fcu, Musculus flexor carpi ulnaris; fdlf, Musculus flexor digitorum longus (foreflipper); ld, Musculus latissimus dorsi; p, Musculus pectoralis; pte, Musculus pronator teres; sc, Musculus supracoracoideus; scs, Musculus subcoracoscapularis; shp, Musculus scapulohumeralis posterior; sl and ecr, Musculus supinator longus and Musculus extensor carpi radialis; sm, Musculus supinator manus; tb, Musculus triceps brachii.

Figure 5 Muscle functions and lines of action of the hind flipper of Cryptoclidus eurymerus (IGPB R 324).

Pelvic limb in (A) ventral and (B) dorsal view. (A) Muscles originating from the ventral pelvic girdle function as flipper depressors. Blue, muscles that twist the flipper leading edge downwards during hindflipper downstroke. Dark green, muscles that twist the flipper trailing edge downwards during upstroke. (B) muscles originating from the dorsal pelvic girdle or the vertebral column are femoral elevators. Blue, muscles that twist the flipper leading edge upwards during the hindflipper upstroke. Dark green, muscles that twist the flipper trailing edge upwards during downstroke. (A) and (B) black dashed line lying approximately in femur long axis direction marks the boundary for femoral protractors/retractors (anterior/posterior to it) Fdlh/fdb have an aponeurosis, preventing individual digital flexion which is schematically represented by a line crossing the foot. Some muscle lines of action are represented with a kink so that they are closely associated with the bones which is the normal condition for extant tetrapod taxa. Lines of action originating outside of a bony area represent origins from the vertebral column. M. ambiens, m. iliofibularis, and m. pubotibialis as well: may rotate anterior humeral edge up or downwards depending on the limb cycle phase. M. ischiotrochantericus originates dorsally and inserts posteroventrally on the femur; m. ambiens originates ventrally on the pubis but insert dorsally into the tibia inserts posteriorly rather ventrally on femur. Abbreviations of bones in italic: f, femur; fi, fibula; fib, fibulare; il, ilium; is, ischium; int, intermedium; p, pubis; t, tibia; tib, tibiale. Abbreviations of muscles: a, Musculus ambiens; addV, Musculus adductor digiti quinti; af, Musculus adductor femoris; cfb, Musculus caudifemoralis brevis; cfl, Musculus caudifemoralis longus; edb, Musculus extensores digitores breves; edl, Musculus extensor digitorum longus; ehp, Musculus extensor hallucis proprius; f, Musculus femorotibialis; fdb, Musculus flexores digitores breves; fdlh, Musculus flexor digitorum longus (hindflipper); fh, Musculus flexor hallucis; fte, Musculus flexor tibialis externus; fti, Musculus flexor tibialis internus; gi and ge, Musculus gastrocnemius internus and Musculus gastrocnemius externus; i, Musculus ischiotrochantericus; ife, Musculus iliofemoralis; ifi, Musculus iliofibularis; it, Musculus iliotibialis; pb and pl, Musculus peroneus brevis and Musculus peroneus longus; pe, Musculus puboischiofemoralis externus; pi, Musculus puboischiofemoralis internus; pit, Musculus puboischiotibialis; pti, Musculus pubotibialis; pp, Musculus pronator profundus; ta, Musculus tibialis anterior.

Table 1 Cryptoclidus eurymerus (IGPB R 324) foreflipper muscle functions in comparison to literature.

Muscle	Muscle abbrevi-ation	Function (Watson, 1924)	Function (Tarlo, 1958)	Function (Robinson, 1975)	Function (Lingham-Soliar, 2000)	Function (Carpenter et al., 2010)	Function (Araújo & Correia, 2015)	Function after current study	
m. latissimus dorsi (+ teres major)	ld	Retractor, rotator (anterior edge up)	Stabilizer	Elevator, eventually retractor, rotator (anterior edge down)	/	Main elevator rotator, protractor	/	Eventually anteriormost portion protraction, posterior portion retraction, elevation; rotation (leading edge upwards)	
m. subcoracoscapularis	scs	Posterior portion: retractor, rotator (anterior edge up); anterior portion: protractor and rotator (anterior edge down)	Protractor, rotator (anterior edge down)	Elevator, rotator (anterior edge up)	/	Pulls humerus into glenoid (stabilizer), eventually elevator	Stabilization	Anterior portion protraction, posterior portion retraction, both elevation, anterior portion rotation (leading edge downwards); posterior portion rotation (leading edge upwards)	
m. scapulohumeralis posterior	shp	–	–	–	/	Protraction, rotation (anterior edge down)	Glenohumeral joint stabilizer	Eventually minor elevation, rotation (leading edge downwards)	
m. deltoideus clavicularis	dc	Protractor, rotator (into the horizontal)	Rotator, protractor	Rotation (anterior edge down), protractor	/	/	Protractor	Protraction, depression, rotation (leading edge downwards)	
m. deltoideus scapularis	ds	Rotation (anterior edge up) or abduction	/	/	Stablílizer	Protraction, elevation, rotation (leading edge upwards)	
m. triceps brachii	tb	–	–	Adjustment of flipper trim, rotator (anterior edge up)	–	–	/	Elevation, rotation (leading edge downwards)	
m. pectoralis	p	Retractor, depressor, rotator (anterior side down)	Prevents anterior flipper movement	Depressor, rotator (anterior side down)	Depressor	Main depressor, rotator (anterior side down)	–	Anterior portion protraction, posterior portion retraction, both depression, posterior portion rotation (leading edge downwards); anterior portion rotation (leading edge upwards)	
m. supracoracoideus	sc	Retractor, rotator (anterior edge down), depressor	/	Rotator (anterior edge down)	Rotator	Rotator (anterior edge up)	Retractor or glenohumeral joint stabilizer	Anterior portion protraction, posterior portion retraction, both depression	
m. coracobrachialis brevis	cb	Retractor, depressor	„adducted backwards…“ (p.199, line 4)	Depressor	Rotator (diretion not specified)	Depressor, eventually retraction during down stroke	Mainly retractor	Retraction, depression, rotation (leading edge upwards)	
m. coracobrachialis longus	cl	Retraction, depression, rotation (leading edge upwards)	
m. biceps brachii + brachialis	bb + b	–	–	Adjustment of flipper trim	–	/	/	Retraction, depression, rotation (leading edge downwards)	
m. extensor carpi ulnaris	ecu	–	–	–	–	–	–	Displaces ulna dorsally/although weakly supported by EPB an insertion to metacarpal V would allow extension of metacarpal V on the adjacent distal carpal	
m. extensor digitorum communis	edc	–	–	–	–	–	–	Extends metacarpals on distal carpals	
m. supinator longus and extensor carpi radialis	sl and ecr	–	–	–	–	–	–	Displaces radius slightly dorsally/weakly supported insertion that expands onto the radiale would allow to displace the whole radial side of the carpus slightly	
m. supinator manus	sm	–	–	–	–	–	–	Abducts metacarpal I on adjacent distal carpal + minor extension	
m. pronator teres	pte	–	–	–	–	–	–	Displaces radius ventrally	
m. flexor carpi ulnaris	fcu	Influences flipper trim	–	–	–	–	–	Displaces ulnar side of carpus ventrally/badly supported possibly additional insertion to metacarpal V would allow to flex metacarpal V on the distal carpal element	
m. flexor digitorum longus (and flexores digitorum superficialis)	fdlf (and fdls)	–	–	–	–	–	–	Flexion of each digit	
m. flexor carpi radialis	fcr	Influences flipper trim	–	–	–	–	–	Flexes metacarpal I on adjacent distal carpal element/ equally well supported would be an insertion to the radial side of the carpus allowing to displace the radial side of the carpus slightly ventrally	
mm. extensores digitores breves superficialis and profundi	edbs and edbp	–	–	–	–	–	–	Extension of each digit	
m. abductor digiti V	abdV	–	–	–	–	–	–	Abducts and slightly flexes digit V	
m. abductor pollicis brevis	apb	–	–	–	–	–	–	Abducts and flexes digit I on metacarpophalangeal joint/might also insert to metacarpal I and would then allow flexion of it on the adjacent distal carpal element	
m. adductor digiti minimi	adm	–	–	–	–	–	–	Adducts and flexes digit V on metacarpophalangeal joint	
Note:

m., musculus; mm., musculi; – not reconstructed; /, reconstructed but no function determined.

Table 2 Cryptoclidus eurymerus (IGPB R 324) hindflipper muscle functions in comparison to literature.

Muscle	Muscle abbreviation	Function (Robinson, 1975)	Function (Carpenter et al., 2010)	Function after current study	
m. iliotibialis	it	Adjusting flipper trim and rotates anterior flipper edge up	–	Elevation, retraction, rotates anterior edge of the flipper up, slight dorsal displacement of tibia on distal femur	
m. femorotibialis	f			Slight dorsal displacement of tibia on distal femur	
m. ambiens	a			Protraction, (if femur depressed, similar to dc rotates anterior edge up; if elevated then rotates anterior edge down), slight dorsal displacement of tibia on distal femur	
m. iliofibularis	ifi	Adjusting flipper trim posteriorly	–	Elevation, rotates anterior edge down, retraction, rotates anterior edge up (as long as fibula above origin)	
m. iliofemoralis	ife	Elevation	Rotates anterior edge up	Elevation, retraction, rotates anterior edge up	
m. puboischiofemoralis internus	pi	Elevator	Elevation	Four possible muscle bellies: elevation from pubis: elevation, rotates anterior edge down, protraction

from ischium: elevation, rotates anterior edge up, minorly retraction

from ilium: elevation, rotates anterior edge up, minorly retraction

from vertebral column: elevation, protraction

	
m. puboischiotibialis	pit	Adjusts flipper trim	–	Depression, rotates anterior edge down	
m. pubotibialis	pti	–	–	Protraction, (if femur depressed, similar to dc rotates anterior edge up; if elevated then rotate anterior edge down)	
m. flexor tibialis internus	fti	–	–	From ischium: retraction, depression, rotates anterior edge down
from ilium/sacral vertebrae/transverse processes of caudal vertebrae: retraction, rotates anterior edge down, elevation	
m. flexor tibialis externus	fte	–	–	From ilium: rotates anterior edge down, retraction, elevation
from ischium: rotates anterior edge down, retraction, depression	
m. caudifemoralis brevis and m. caudifemoralis longus	cfb	Elevation, rotates anterior flipper edge down	Rotates anterior flipper edge down	elevation, retraction, rotates anterior edge down	
cfl	retraction, elevation, rotates anterior edge down	
m. ischiotrochantericus	i	Rotates anterior flipper edge down, elevation, retraction	–	Retraction, depression, rotation of anterior edge down	
m. adductor femoris	af	Depressor, rotation anterior flipper edge down”	–	From anterior ischium: depression
from lateroposterior ischium: adduction, retraction	
m. puboischiofemoralis externus	pe	Depressor	Depressor	From pubis: depression, protraction, rotates anterior edge up
from ischium: depression, retraction, rotates anterior edge down	
m. extensor digitorum longus	edl	–	–	Extension of digits I–IV (on tarsometatarsal joints)	
m. peroneus longus and m. peroneus brevis	pb and pl	Adjusts flipper trim	–	Extends tarsometatarsal joint of digit V, abduct metatarsal V	
m. tibialis anterior	ta	Adjusts flipper trim	–	Abducts metatarsal I	
m. gastrocnemius internus and m. gastrocnemius externus	gi and ge	–	–	Flexors of all 5 digits in all phalangeal joints, also acting on metatarsal I and V	
m. flexor digitorum longus	fdlh	–	–	Long flexors of all 5 digits	
m. pronator profundus	pp	–	–	Flexion of carpometacarpal joints of digit I (eventually digit II and III)	
mm. extensores digitores breves	edb	–	–	Extension of all phalangeal joints in all V digits	
mm. flexores digitores breves	fdb	–	–	Flexors of digits I–IV	
m. extensor hallucis proprius	ehp	–	–	Extension of extends or adducts metatarsal I (on tarso-metatarsal joint)	
m. adductor digiti quinti	addV	–	–	Flexor of digit V	
m. flexor hallucis	fh	–	–	Flexor of digit I	
Note:

m., musculus; mm, musculi; -, not reconstructed.

Abbreviations

Foreflipper: abdV, Musculus abductor digiti V; adm, Musculus adductor digiti minimi; apb, Musculus abductor pollicis brevis; b, Musculus brachialis; bb, Musculus biceps brachii; cb, Musculus coracobrachialis brevis; cl, Musculus coracobrachialis longus; dc, Musculus deltoideus clavicularis; ds, Musculus deltoideus scapularis; ecu, Musculus extensor carpi ulnaris; edbp, Musculi extensores digitores breves profundi; edbs, Musculi extensores digitores breves superficialis; edc, Musculus extensor digitorum communis; fcr, Musculus flexor carpi radialis; fcu, Musculus flexor carpi ulnaris; fdlf, Musculus flexor digitorum longus (foreflipper); fdls, Musculi flexores digitorum superficialis; ld, Musculus latissimus dorsi; p, Musculus pectoralis; pte, Musculus pronator teres; sc, Musculus supracoracoideus; scs, Musculus subcoracoscapularis; shp, Musculus scapulohumeralis posterior; sl and ecr, Musculus supinator longus and Musculus extensor carpi radialis; sm, Musculus supinator manus; tb, Musculus triceps brachii.

Hindflipper: a, Musculus ambiens; addV, Musculus adductor digiti quinti; af, Musculus adductor femoris; cfb, Musculus caudifemoralis brevis; cfl, Musculus caudifemoralis longus; edb, Musculi extensores digitores breves; edl, Musculus extensor digitorum longus; ehp, Musculus extensor hallucis proprius; f, Musculus femorotibialis; fdb, Musculi flexores digitores breves; fdlh, Musculus flexor digitorum longus (hindflipper); fh, Musculus flexor hallucis; fte, Musculus flexor tibialis externus; fti, Musculus flexor tibialis internus; gi and ge, Musculus gastrocnemius internus and Musculus gastrocnemius externus; i, Musculus ischiotrochantericus; ife, Musculus iliofemoralis; ifi, Musculus iliofibularis; it, Musculus iliotibialis; pb and pl, Musculus peroneus brevis and Musculus peroneus longus; pe, Musculus puboischiofemoralis externus; pi, Musculus puboischiofemoralis internus; pit, Musculus puboischiotibialis; pti, Musculus pubotibialis; pp, Musculus pronator profundus; ta, Musculus tibialis anterior.

Materials & Methods

Material

Muscles were reconstructed for the plesiosaur Cryptoclidus eurymerus (Phillips, 1871) exhibited at the Goldfuß Museum, Section of Paleontology, Institute of Geosciences (IGPB), Rheinische Friedrich-Wilhelms-Universität Bonn, Germany (Fig. 1A). In 1909, the specimen (IGPB R 324) was excavated by Alfred Leeds from the Lower Oxford Clay (Middle Jurassic) of Whittlesea near Peterborough, UK. In 1911, the University of Bonn bought the almost complete skeleton with a fragmentary skull via Berhard Stürtz from Leeds.

Amongst other characters, the anteriorly and posteriorly much expanded humerus are diagnostic for Cryptoclidus eurymerus (Brown, 1981). According to Benson & Druckenmiller (2014), Cryptoclidus eurymerus is part of the Cryptoclididae, which belong to the Plesiosauroidea. Cryptoclidids are rather derived plesiosauroids and do not represent plesiosmorphic plesiosaurians. Nevertheless, we chose to reconstruct the musculature for Cryptoclidus eurymerus because it is a taxon which is known from various specimens, its skeleton is relatively completely preserved and known, there are different ontogenetic stages known which may be interesting to study in the future, and for comparability reasons, i.e., Lingham-Soliar (2000), Robinson (1975) and Araújo & Correia (2015) based their muscle reconstructions partially on this taxon, as well as e.g., Godfrey’s (1984) discussion on plesiosaur locomotion.

Homologies

Pectoral girdle homology in Plesiosauria

Plesiosaur shoulder girdle homology followed (Araújo & Correia, 2015), which is used to establish a comparative basis to the extant taxa used for the EPB. Araújo & Correia (2015) proposed three possible hypotheses for how the plesiosaur pectoral girdle could have evolved from that of basal Eosauropterygia: The coracoids constantly keep their median contact while they are relocated posteriorly (hypothesis I). The coracoids loose contact with the scapula and the median suture between the coracoids. Then, the coracoids are displaced posteriorly and the median coracoid contact is reestablished again (hypothesis II). The coracoids are rotated backwards so that the anterior side of the coracoid comes to lie medially and the medial side posteriorly (hypothesis III). Placodonts seem to support hypothesis II, but their locomotory adaptations differ a lot from those of other Eosauropterygia, so Araújo & Correia (2015) conclude that this hypothesis is not their preferred one. For hypothesis III the muscles that originate from the coracoid would need to be reoriented fundamentally. So, hypothesis II and III involve more evolutionary steps than I, and I seems to be supported by the developmental patterns of recent sauropsids and fossil early Neodiapsida like e.g., Claudiosaurus, therefore hypothesis I is the preferred hypothesis (Araújo & Correia, 2015).

Pelvic girdle homology in Plesiosauria

Homology of the pelvic girdle of Plesiosauria has yet to be established. The authors presume that anterior and posterior sides of the ischium and pubis correspond to the same sides as in extant Sauropsida. In extant sauropsids ischium and pubis are somewhat inclined dorsoventrally. For plesiosaurs, the acetabulum may have been moved ventrally while the suture of the opposing sides of pubis and ischium in the body mid-line have been shifted dorsally in comparison to other Eosauropterygia. This way, pubis and ischium have become two almost flat-lying bones on the plesiosaur belly. From the lateral side to the body mid-line, pubis and ischium slant slightly v-shaped (Andrews, 1910). The lateral concavity anterior to the acetabulum on the pubis may be convergent to the lateral process in turtles (Walker, 1973) or the pubic tubercle in lepidosaurs (Russell & Bauer, 2008). A lateral process (called like that in turtles Walker (1973)) or an ischiadic tuberosity (called like that in lepidosaurs (Russell & Bauer, 2008) is not present in the plesiosaur ischium.

Muscle homology in Plesiosauria

Muscle homology was established for all plesiosaur flipper musculature terminology published in previous papers (Watson, 1924; Tarlo, 1958; Robinson, 1975; Lingham-Soliar, 2000; Carpenter et al., 2010; Araújo & Correia, 2015) so far, based on topology (Data S1). This means, that we decided to synonymize muscles that have similar muscle attachment areas and a similar hypothetical muscle course, but different names and based on their relative position in relation to other muscles (Holliday & Witmer, 2007).

Muscle homology in Sauropsida

Muscle homologies have been largely established within Crocodylia (Meers, 2003, Suzuki & Hayashi, 2010; Suzuki et al., 2011), Testudines (Walker, 1973), and Lepidosauria (Russell & Bauer, 2008) based on ontogeny, neurology, and topology. For this study, we established primary homology amongst Crocodylia, Lepidosauria, and Testudines based on topological criteria (i.e., muscle attachments and muscle courses, relative topological relationship to other muscles) following e.g., Rieppel & Kearney (2002) and Richter (2005). Ontogeny and neurology are important in establishing homology as well but the authors would like to point out that they may be considered as a form of topology as well (see, e.g., Agnarsson & Coddington, 2007 for a review of the definition of homology and homology criteria, Holliday & Witmer, 2007). For information on forelimb myological homologies across Sauropsida Remes (2007) was sometimes consulted and was cited if this was the case. Homology lists with citations were prepared for each muscle (please see section “muscle reconstructions” below).

Terminology for bone orientation in Plesiosauria and Sauropsida

Bone orientational terminology for Sauropsida was aimed to match the result, the locomotory musculature of plesiosaurs, and leans on Romer (1976): Directions within the vertebral column are described with cranial and caudal. Otherwise, orientations in the pectoral and pelvic limb are given with dorsal and ventral, anterior and posterior, and proximal and distal. The dorsal projection of the scapula and the ilium are described with dorsal vs. ventral, medial vs. lateral, and anterior vs. posterior.

Extant phylogenetic bracket

Extant phylogenetic bracket of Plesiosauria

Plesiosaur muscles were reconstructed with the extant phylogenetic bracket (EPB) (Bryant & Seymour, 1990; Bryant & Russel, 1992; Witmer, 1995). For the EPB of Plesiosauria, Lepidosauria, Archosauria (i.e., Crocodylia), and Testudines were chosen as extant taxa. As the origin of Sauropterygia may be within the archosauromorph or the lepidosauromorph clade, crocodiles and lepidosaurs could interchangibly be the upper or lower bracket. Turtles are the sister-taxon of Crocodylia according to genetical analyses and therefore archosaurs (Crawford et al., 2015; Pereira et al., 2017) (Figs. 1A and 1B; Table 3). The turtle shoulder girdle has been folded beneath the ribs and shell; accordingly, some muscular connections have been rearranged and some have been kept. Nevertheless, the fundamental restructuring of the turtle bauplan, has not affected lower arm and leg, hand, and foot musculature (Nagashima et al., 2009). Therefore, turtles can provide valuable information on plesiosaur musculature.

Table 3 Presence of muscles and citations on which muscle homology and muscle reconstructions are based on of extant Testudines, Crocodylia, and Lepidosauria. No citation = muscle not present in the respective taxon.

Muscle	Testudines	Crocodylia	Lepidosauria	
fore limb				
m. latissimus dorsi (+ teres major)	Walker, 1973	Meers, 2003; Suzuki & Hayashi, 2010	Russell & Bauer, 2008; Zaaf et al., 1999	
m. subcoracoscapularis	Walker, 1973	Meers, 2003; Suzuki & Hayashi, 2010	Russell & Bauer, 2008; Anzai et al., 2014	
m. scapulohumeralis posterior		Meers, 2003; Suzuki & Hayashi, 2010	Russell & Bauer, 2008; Zaaf et al., 1999; Jenkins & Goslow, 1983	
m. deltoideus clavicularis	Walker, 1973, Wyneken, 2001	Meers, 2003; Suzuki & Hayashi, 2010	Russell & Bauer, 2008; Zaaf et al., 1999; Anzai et al., 2014	
m. deltoideus scapularis	Walker, 1973	Meers, 2003; Suzuki & Hayashi, 2010	Russell & Bauer, 2008; Zaaf et al., 1999; Anzai et al., 2014	
m. triceps brachii	Walker, 1973	Meers, 2003; Suzuki & Hayashi, 2010	Russell & Bauer, 2008; Zaaf et al., 1999; Jenkins & Goslow, 1983; Anzai et al., 2014	
m. pectoralis	Walker, 1973	Meers, 2003; Suzuki & Hayashi, 2010	Russell & Bauer, 2008; Zaaf et al., 1999; Anzai et al., 2014	
m. supracoracoideus	Walker, 1973	Meers, 2003; Suzuki & Hayashi, 2010	Russell & Bauer, 2008	
m. coracobrachialis brevis	Walker, 1973, Abdala et al., 2014	Meers, 2003; Suzuki & Hayashi, 2010	Russell & Bauer, 2008; Zaaf et al., 1999	
m. coracobrachialis longus	Walker, 1973	Meers, 2003; Suzuki & Hayashi, 2010	Russell & Bauer, 2008; Zaaf et al., 1999; Anzai et al., 2014; Jenkins & Goslow, 1983	
m. biceps brachii + brachialis	Walker, 1973	Meers, 2003; Suzuki & Hayashi, 2010	Russell & Bauer, 2008; Zaaf et al., 1999; Anzai et al., 2014	
m. extensor carpi ulnaris	Walker, 1973	Meers, 2003; Suzuki & Hayashi, 2010	Russell & Bauer, 2008	
m. extensor digitorum communis	Walker, 1973; Abdala, Manzano & Herrel, 2008	Meers, 2003; Suzuki & Hayashi, 2010	Russell & Bauer, 2008	
m. supinator longus and extensor carpi radialis	Walker, 1973	Meers, 2003; Suzuki & Hayashi, 2010	Russell & Bauer, 2008	
m. supinator manus	Walker, 1973	Meers, 2003; Suzuki & Hayashi, 2010	Russell & Bauer, 2008	
m. pronator teres	Walker, 1973; Abdala, Manzano & Herrel, 2008	Meers, 2003; Suzuki & Hayashi, 2010	Russell & Bauer, 2008	
m. flexor carpi ulnaris	Walker, 1973	Meers, 2003; Suzuki & Hayashi, 2010	Russell & Bauer, 2008	
m. flexor digitorum longus (and flexores digitorum superficialis)	Walker, 1973; Abdala, Manzano & Herrel, 2008	Meers, 2003; Suzuki & Hayashi, 2010	Russell & Bauer, 2008	
m. flexor carpi radialis	Walker, 1973		Russell & Bauer, 2008	
mm. extensores digitores breves superficialis and profundi	Walker, 1973; Abdala, Manzano & Herrel, 2008	Meers, 2003	Russell & Bauer, 2008	
m. abductor digiti V	Walker, 1973; Abdala, Manzano & Herrel, 2008	Meers, 2003	Russell & Bauer, 2008	
m. abductor pollicis brevis	Walker, 1973; Abdala, Manzano & Herrel, 2008	Meers, 2003	Russell & Bauer, 2008	
m. adductor digiti minimi	Walker, 1973; Abdala, Manzano & Herrel, 2008	Meers, 2003	Russell & Bauer, 2008	
hind limb				
m. iliotibialis	Walker, 1973, Zug, 1971	Romer, 1923; Otero, Gallina & Herrera, 2010; Suzuki et al., 2011, Gatesy, 1997	Snyder, 1954; Russell & Bauer, 2008	
m. femorotibialis	Walker, 1973, Zug, 1971	Romer, 1923; Otero, Gallina & Herrera, 2010; Suzuki et al., 2011, Gatesy, 1997	Snyder, 1954; Russell & Bauer, 2008	
m. ambiens	Walker, 1973, Zug, 1971	Romer, 1923; Otero, Gallina & Herrera, 2010; Suzuki et al., 2011	Snyder, 1954; Russell & Bauer, 2008	
m. iliofibularis	Walker, 1973, Zug, 1971	Romer, 1923; Otero, Gallina & Herrera, 2010; Suzuki et al., 2011, Gatesy, 1997	Snyder, 1954; Russell & Bauer, 2008	
m. iliofemoralis	Walker, 1973, Zug, 1971	Romer, 1923; Otero, Gallina & Herrera, 2010; Suzuki et al., 2011, Gatesy, 1997	Snyder, 1954; Russell & Bauer, 2008	
m. puboischiofemoralis internus	Walker, 1973, Zug, 1971	Romer, 1923; Otero, Gallina & Herrera, 2010; Suzuki et al., 2011, Gatesy, 1997	Snyder, 1954; Russell & Bauer, 2008	
m. puboischiotibialis	Walker, 1973, Zug, 1971	Romer, 1923; Otero, Gallina & Herrera, 2010; Suzuki et al., 2011, Gatesy, 1997	Snyder, 1954; Russell & Bauer, 2008	
m. pubotibialis	Walker, 1973, Zug, 1971		Snyder, 1954; Russell & Bauer, 2008	
m. flexor tibialis internus	Walker, 1973, Zug, 1971	Romer, 1923; Otero, Gallina & Herrera, 2010; Suzuki et al., 2011, Gatesy, 1997	Snyder, 1954; Russell & Bauer, 2008	
m. flexor tibialis externus	Walker, 1973, Zug, 1971	Romer, 1923; Otero, Gallina & Herrera, 2010; Suzuki et al., 2011, Gatesy, 1997	Snyder, 1954; Russell & Bauer, 2008	
m. caudifemoralis brevis	Walker, 1973, Zug, 1971	Romer, 1923; Otero, Gallina & Herrera, 2010; Suzuki et al., 2011, Gatesy, 1997	Snyder, 1954; Russell & Bauer, 2008	
m. caudifemoralis longus		Romer, 1923; Otero, Gallina & Herrera, 2010; Suzuki et al., 2011, Gatesy, 1997	Snyder, 1954; Russell & Bauer, 2008	
m. ischiotrochantericus	Walker, 1973, Zug, 1971	Romer, 1923; Otero, Gallina & Herrera, 2010; Suzuki et al., 2011, Gatesy, 1997	Snyder, 1954; Russell & Bauer, 2008	
m. adductor femoris	Walker, 1973, Zug, 1971	Romer, 1923; Otero, Gallina & Herrera, 2010; Suzuki et al., 2011	Snyder, 1954; Russell & Bauer, 2008	
m. puboischiofemoralis externus	Walker, 1973, Zug, 1971	Romer, 1923; Otero, Gallina & Herrera, 2010; Suzuki et al., 2011, Gatesy, 1997	Snyder, 1954; Russell & Bauer, 2008	
m. extensor digitorum longus	Walker, 1973, Zug, 1971	Suzuki et al., 2011	Snyder, 1954; Russell & Bauer, 2008	
m. peroneus longus and m. peroneus brevis	Walker, 1973, Zug, 1971	Suzuki et al., 2011	Snyder, 1954; Russell & Bauer, 2008	
m. tibialis anterior	Walker, 1973, Zug, 1971	Suzuki et al., 2011	Snyder, 1954; Russell & Bauer, 2008	
m. gastrocnemius internus and m. gastrocnemius externus	Walker, 1973, Zug, 1971	Suzuki et al., 2011, Otero, Gallina & Herrera, 2010	Snyder, 1954; Russell & Bauer, 2008	
m. flexor digitorum longus	Walker, 1973, Zug, 1971	Suzuki et al., 2011	Snyder, 1954; Russell & Bauer, 2008	
m. pronator profundus	Walker, 1973, Zug, 1971	Suzuki et al., 2011	Snyder, 1954; Russell & Bauer, 2008	
mm. extensores digitores breves	Walker, 1973, Zug, 1971	Suzuki et al., 2011	Snyder, 1954; Russell & Bauer, 2008	
mm. flexores digitores breves	Walker, 1973, Zug, 1971	Suzuki et al., 2011	Snyder, 1954; Russell & Bauer, 2008	
m. extensor hallucis proprius	Walker, 1973, Zug, 1971	Suzuki et al., 2011	Russell & Bauer, 2008	
m. adductor digiti quinti			Russell & Bauer, 2008	
m. flexor hallucis		Suzuki et al., 2011	Russell & Bauer, 2008	

Pectoral and pelvic limb myology of lepidosaurs relies mostly on Russell & Bauer (2008) who present each locomotory muscle for Iguana, but extensively review lepidosaur myological research and homologies including Sphenodon. In case of doubt or additional questions on lepidosaur forelimb musculature, Zaaf et al. (1999) (on two gekkotans (Eublepharis macularius and Gekko gecko)), Anzai et al. (2014) (on various Anolis species), Jenkins & Goslow (1983) (on Varanus exanthematicus) were considered. Additional information on lepidosaur hindlimb myology was drawn from Snyder (1954) who studied hindlimb musculature of Iguanidae and Agamidae.

Crocodilian forelimb myology is based on Meers (2003) who sampled and compared various crocodilian taxa (Alligator mississippiensis, Crocodylus siamensis, C. acutus, Osteolaemus tetraspis, and Gavialis gangeticus). Suzuki & Hayashi (2010) were also consulted for crocodilian muscle attachments on the pectoral girdle, humerus, and radius and ulna. They sampled Caiman crocodilus and Crocodylus siamensis and C. niloticus. Crocodilian hindlimb myology is largely based on Suzuki et al. (2011) who studied Caiman crocodilus fuscus, Crocodylus siamensis, and C. porosus. Supplementary and comparative information on pelvic muscles inserting into the femur or spanning it were taken from Otero, Gallina & Herrera (2010) (on Caiman latirostris), Romer (1923) (Alligator mississippiensis), and Gatesy (1997) (Alligator mississippiensis).

Turtle forelimb and hindlimb myology is based on Walker (1973) who primarily describes fore- and hindlimb myology of Pseudemys scripta elegans, but he compares them to other turtles he dissected including terrestrial, semi-aquatic, and marine ones. Further, Walker (1973) also extensively reviews turtle myological literature and included muscle homologies. Abdala, Manzano & Herrel (2008) who studied lower arm and hand muscles of several terrestrial and semi-aquatic Testudines were additionally considered.

Hindlimb myology of Testudines was also based on Zug (1971) who depicts and describes variability to the pictured musculature of Pseudemys by Walker (1973) who despite describing variability of muscle attachments, did not figure them.

Criteria for reconstruction of a plesiosaur muscle

Generally, we found most favourable for the reconstruction of a muscle attachment area (Figs. 2 and 3), was a support by all three extant taxa, Lepidosauria, Crocodilia, and Testudines. A support by two extant taxa was given priority over support by just one extant taxon. Further, it should be noted that support from either lepidosaurs and crocodilians (Archosauria) or lepidosaurs and turtles (Archosauria) is considered stronger than one from turtles and crocodiles (both Archosauria). If a support from crocodiles and turtles is considered in the results section, a reason is given why this support is phylogenetically weaker than e.g., one from lepidosaurs and turtles.

Yet, in detail, the muscle reconstructions are more complex than this, because other criteria are worth to be considered as well: 1. the presence of actual muscle attachment surfaces, i.e., osteological correlates, in the fossil Cryptoclidus eurymerus, 2. general conclusions drawn from the functional analogues because their pectoral girdle has been subjected to similar selective pressures, 3. the overall spatial arrangement of extant tetrapod shoulder girdle musculature, 4. the functionality of each plesiosaur muscle, 5. Sphenodon musculature, because it is the long ago diverged sister group to all other extant lepidosaurs. 1. The limb skeleton of Cryptoclidus eurymerus (IGPB R 324) was examined for osteological correlates. The osteological correlates considered were rugosities, pits, striations, and ridges. If an osteological correlate was present in Cryptoclidus eurymerus, a description of its appearance and which muscle was assigned to it is given in the muscle subchapters in the results section.

2. A closer look at the myology of the functional analogues (Chelonioidea, Spheniscidae, Otariinae, and Cetacea) of plesiosaurs helped to identify traits that these secondary aquatic tetrapods share or diverge in. This is relevant because the extant EPB taxa mostly do not share the same locomotory style (underwater flight) and ecology (a highly marine lifestyle) with plesiosaurs but are instead mostly terrestrial, i.e., walking or climbing, or semi-aquatic, i.e., lateral undulation, rowing, bottom-walking (Russell & Bauer, 2008; Rivera, Rivera & Blob, 2013; Manter, 1940).

3. The general three-dimensional arrangement of sauropsid pectoral myology was considered as well because it is quite similar on a large scale: In lateral view of a sauropsid, superficially lying m. pectoralis (p) fans out from the humerus ventrally to posteroventrally. M. latissimus dorsi (ld), also lying superficially, fans out from the humerus dorsally and caudodorsally. Sauropsids also have in common, that the m. deltoideus scapularis (ds) muscle belly runs rather dorsally above the humerus, while the m. deltoideus clavicularis (dc) portion runs anteriorly (Walker, 1973; Jenkins & Goslow, 1983; Meers, 2003; Russell & Bauer, 2008; Suzuki & Hayashi, 2010; personal observation of Caretta caretta dissection). If ld is dissected off a lepidosaur, m. supracoracoideus (sc) and m. subcoracoscapularis (scs) become visible (that take an anterior to anterodorsal course Jenkins & Goslow, 1983; Russell & Bauer, 2008). This is similar in crocodilians and turtles, except that the latter lack m. scapulohumeralis posterior (shp) (Walker, 1973; Meers, 2003; Suzuki & Hayashi, 2010; personal observation). If p is dissected off in ventral view, the deltoids and sc can always be found anteriorly. Variable across sauropsids appears to be the deeper musculature that follows from anterior to posterior: in crocodilians m. biceps brachii (bb), m. coracobrachialis brevis (cb), and scs (Meers, 2003; Suzuki & Hayashi, 2010), in lepidosaurs cb, bb, and m. coracobrachialis longus (cl) (Jenkins & Goslow, 1983), and in turtles cb, cl, and bb, also visible in turtles is the sc due to its peculiar origin on the ventral coracoid (Walker, 1973; personal observation).

Despite variable origins and insertions of extensors and flexors that are on the ent- and ectepicondyle of the humerus, their course is the same. Across Sauropsida m. supinator longus and m. extensor carpi radialis (sl and ecr), m. extensor digitorum communis (edc), and m. extensor carpi ulnaris (ecu) fan out over the lower arm from anterior to posterior (digit I to digit V) (Walker, 1973; Meers, 2003; Russell & Bauer, 2008; Suzuki & Hayashi, 2010). Similarly, the flexors originating from the humerus also fan out over the lower arm. From digit I to digit V these are m. flexor carpi radialis (fcr), fdlf, m. flexor carpi ulnaris (fcu). Pte lies deep to fcr and m. flexor digitorum longus (foreflipper)) (fdlf) in lepidosaurs and turtles (Walker, 1973; Russell & Bauer, 2008). Crocodilians pose the exception, as such that fcr is reduced and m. pronator teres (pte) is situated in its place.

So, during the course of plesiosaur muscle reconstructions, above mentioned generalized three-dimensional arrangement of muscles in Sauropsida was considered to hold true for plesiosaurs, too.

4. Every muscle needs to have a function, otherwise it would have become vestigial and reduced. So, we thoroughly tried to find functions for each reconstructed muscle which may deviate from those of extant taxa. Plesiosaur muscles may have different functions because the fore- and hindflipper of plesiosaurs are dorsoventrally flattened and broadened and diverge from the average extant terrestrial lepidosaur, tortoise, and the semi-aquatic crocodiles, i.e., the EPB taxa used for phylogenetic inference.

Further, we tried to find muscles that would enable underwater flight, i.e., protraction/retraction, elevation/depression, clockwise and counter-clockwise length-axis rotation of humerus and femur, and flipper twisting along the flipper length axis. It happens, that some muscle attachment areas can be supported by two of the extant sauropsid groups by the EPB, but favorable for flipper twisting is a muscle attachment area which is only supported by one of the extant sauropsid taxa. In the result-section it is always mentioned when one option is less well supported by the EPB but instead implied and supported by its functionality.

5. If the EPB turned out to be little informative, i.e., three different equally likely options were received, Sphenodon myology was considered as well. Sphenodon is the only extant species of Sphenodontia, which pose the long-diverged sister group to all other recent squamates. Therefore, Sphenodon adds important information on the myology of extant Sauropsida and aids in finding a preferred hypothesis on plesiosaur muscle reconstructions. Sphenodon is missing the ilioischiadic ligament, presumably like plesiosaurs, but unlike to Iguana. So, Sphenodon can inform the plesiosaur muscle reconstructions on where the muscle attachments from the ilioischiadic ligament may have spread to, unlike to Iguana.

Determining muscle functions of muscles originating from the pectoral and the pelvic girdle in plesiosaurs

Different functions were assigned to muscles that developed subportions, i.e., that extend from the glenoid/acetabulum cranially or anteriorly and caudally and posteriorly. It is likely that the muscles that are placed cranially or anteriorly to the glenoid/acetabulum play a role in protraction and the muscles that are placed caudally or posteriorly to the glenoid/acetabulum in retraction. Also, muscles that originate dorsally to the glenoid/acetabulum or on the dorsal pectoral or pelvic girdle have an elevational function, contrary to muscles that originate ventrally to the glenoid/acetabulum which act as depressors. Rotators rotate the humerus or femur length axis and distal bony elements. A potential rotatory function is given, if the hypothetical course a muscle takes between its origin and insertion does not lie parallel to the axis of rotation of the humerus and femur but is angled to it (Figs. 4 and 5; Tables 1 and 2). The whole flipper is rotated by approximately 19°, as suggested by hydrodynamic studies, by flipper rotators and twisted by flipper twisting muscles (Witzel, Krahl & Sander, 2015; Witzel, 2020). Flipper rotators can rotate the humerus and femur in two different ways during downstroke and upstroke: During the downstroke, the flipper leading edge is rotated downwards and the flipper trailing edge upward. During the upstroke, the flipper leading edge is rotated upwards and the flipper trailing edge downwards. Hence, muscles that rotate effectively the flipper leading edge downward can have two different origin areas. They either originate posterior to the glenoid/acetabulum from the ventral coracoid/ischium or anterior/cranially to the glenoid/acetabulum from the dorsal pectoral/pelvic girdle or dorsally from the vertebral column. For an upward rotation of the flipper leading edge the opposite is true. In the following text the terms anterior and posterior portion of a certain pectoral muscle will be used, because in the pectoral girdle anterior and posterior portions of a muscle do not necessarily correspond to an origin from scapula or coracoid (Table 1). For pelvic musculature they will be termed pubic or ischial portion, as they do seem to correspond well with the bony elements (Table 2).

In the following text muscle functions as the authors themselves interpreted them are discussed, not secondary interpretations of other authors as e.g., done by Carpenter et al. (2010). Watson (1924) poses the exception, as he writes that every muscle that originates ventral to the glenoid has probably a depressional function, but does not list them. Therefore, we deduced that these are: cb, p, the deltoids, scapulohumeralis anterior, and sc (which was depicted as depressor by Watson (1924) himself). Further adductor/abductor is used by following authors (Robinson, 1975; Lingham-Soliar, 2000; Carpenter et al., 2010; Araújo & Correia, 2015) for muscles that move the plesiosaur flippers ventrally below the body midline or dorsally above the body midline. Instead, depression and elevation are used in this study because it highlights the concept of underwater flight, as used by e.g., Rivera, Wyneken & Blob (2011) and Rivera, Rivera & Blob (2013); Krahl et al. (2019) for sea turtle underwater flight.

Figures

For the muscle attachment figures (Figs. 2 and 3), the bone margins of all fore- and hindflipper bones of Cryptoclidus eurymerus were traced. Then, the muscle attachment areas were projected onto the flipper tracings. Dotted lines in the colors of muscles and an associated question marks highlight visually that these muscle attachment areas are less likely than those attachment areas with solid lines but that they are worth to be reconstructed often due to functional reasons.

Stars in Figs. 2 and 3 highlight muscles that are associated with osteological correlates. Sometimes two or more attachments were reconstructed onto one muscle scar: (1) the insertion of pectoralis and supracoracoideus into the proximoventral humerus (Fig. 2A), (2) the latissimus dorsi, subcoracoscapularis, and scapulohumeralis posterior insertion into the proximodorsal humerus (Fig. 2B), (3) the insertion of puboischiofemoralis externus and ischiotrochantericus into the proximoventral femur (Fig. 3A), and (4) the puboischiofemoralis internus and iliofemoralis insertion into the dorsal and proximal femur (Fig. 3B). Osteological correlates of the humerus and femur have been nicely figured by Brown (1981; Figure 15, p.275 and Figure 16, p. 277).

The same line tracings that are used in Figs. 2 and 3 are re-used for Figs. 4 and 5. To show the muscle functions, the lines of action for all muscles and subportions are added. Lines of action are a direct connection of muscle origin and insertion in a straight line. This is often just a broad approximation to the muscle courses that often wrap around boney structures during parts of the limb cycle (see e.g., Krahl et al., 2019). Further, muscles tend to take a course that is relatively close to the respective parts of the skeleton, so generally muscle lines of action are arranged fan-shaped from the pectoral/pelvic girdle towards the limbs. Nevertheless, muscle courses that run very far posteriorly, e.g., m. biceps brachii are represented by a kinked line of action because otherwise it would be impossible to keep them within the body outline. These muscles were possibly held closer to the body by connective tissue.

Results

Myology of functional analogues and implications for plesiosaur muscle reconstructions

Generally, all four functional analogues suggest that locomotory muscles spanning the shoulder joint do not experience reduction in the land-water transition, independent of the locomotory mode they employ (Walker, 1973; English, 1977; Schreiweis, 1982; Wyneken, 2001; Cooper et al., 2007. The set of muscles they have is determined by their phylogeny. So, depending on whether plesiosaurs are on the archosaur or on the lepidosaur lineage they either could have a scapulohumeralis anterior, shp, or a second m. flexor tibialis externus (fte) head (s. Abbreviations). A reduction takes place in the two-joint muscles that span the glenoid and the elbow, bb and m. triceps brachii (tb): in penguins and whales bb is fully reduced (Schreiweis, 1982; Cooper et al., 2007). In sea turtles tb is either much reduced or entirely reduced depending on the species (Walker, 1973). Sea lions and fur seals have both muscles well developed (English, 1977).

Cetacea have extremely reduced flexors and extensors of the lower arm and hand in comparison to Chelonioidea, Spheniscidae, and Otariinae (Walker, 1973; English, 1976a; Schreiweis, 1982; Louw, 1992; Cooper et al., 2007). In Cetacea, mainly the long digital flexor and extensor are exempt from complete reduction (Cooper et al., 2007). For plesiosaurs that swim paraxially (Krahl, 2021), a similar musculature arrangement is unlikely because cetaceans mainly swim with their fluke (Fish, 1996; Woodward, Winn & Fish, 2006) and the foreflippers are control surfaces (Fish, 1996, 2002; Woodward, Winn & Fish, 2006).

Contrastingly, Otariinae have well developed digital muscles. They even have muscles that spread the digital webbing during the rowing phase of the forflipper beat cycle (English, 1976a). These muscles that are employed in individual digital movement are lacking in penguins, sea turtles, and cetaceans that entirely rely on lift-based locomotion (Walker, 1973; English, 1976a; Schreiweis, 1982; Cooper et al., 2007). Due to the flipper shape (hydrofoil, tapers from the base to the flipper tip), plesiosaurs were most likely relying only on lift-based locomotion (underwater flight) unlike sea lions (see Krahl, 2021 for review). Therefore, muscles employed in individual digital movement are not reconstructed in plesiosaurs, comparable to sea turtles, penguins, and whales.

Extensors and flexors are generally reduced in size in sea turtles and penguins (Walker, 1973; Schreiweis, 1982; Louw, 1992; Cooper et al., 2007) in comparison to their terrestrial relatives. During the ontogeny of sea turtles, extensors and flexors show an increase in fascia development and in connective tissue (Walker, 1973; Wyneken, 2001; Abdala, Manzano & Herrel, 2008). Penguins overall show development of longer tendons and reduced muscle belly size. Muscle fusion with dermis is reported for sea turtles, penguins, and otariines (Walker, 1973; English, 1976a; Louw, 1992; Wyneken, 2001; Abdala, Manzano & Herrel, 2008). In penguins, skin even fuses to bone (Louw, 1992). Overall individual digital movement is reduced in penguins and sea turtles (Walker, 1973; Louw, 1992). This implys that it is likely that muscles in plesiosaurs flippers similarly tended to become aponeurotically or develop longer tendons, or fused with the dermis.

Fcu is hypertrophied in sea turtles and Otariinae and present in Spheniscidae and Cetacea (Walker, 1973; English, 1976a; Schreiweis, 1982; Louw, 1992; Cooper et al., 2007). It is possible that it rotates the flipper leading edge up in both former taxa and therefore needed to be relatively stronger than in a terrestrial environment due to the higher viscosity of water in comparison to air.

Further, in comparison to other turtles, which have rather straight and only slightly distally expanding humeri, sea turtle humeri are anteriorly straight and posteriorly curved and expanded (Walker, 1973; Wyneken, 2001; Krahl et al., 2019). Extensors originate anteriorly from the radial epicondyle just proximal to the joint capsule in cheloniids and other turtles alike (Walker, 1973; Krahl et al., 2019). On the posterior side, flexors usually arise in turtles in the same fashion as the extensors anteriorly. In Cheloniidae the origin areas have migrated proximally up to approximately half the shaft length (Walker, 1973; Krahl et al., 2019). One of the most apparent features of Cryptoclidus eurymerus fore- and hindflippers is the hammer shape of its humeri and femora, which is more pronounced in the former than in the latter (Andrews, 1910). In comparison to the sea turtle humeri, it was decided to place the origins of extensors and flexors on the humerus and femur of Cryptoclidus eurymerus rather proximal onto the curved and expanding epicondyles from approximately half the shaft length on further distally.

The general assumptions from the foreflippers of the extant functional analogues are transferred to the muscle reconstructions of the plesiosaur foreflipper and hindflipper. So in sum, the functional analogues suggest all muscles spanning the glenoid and acetabulum in the EPB taxa (Lepidosauria, Testudines, Crocodylia) can be reconstructed for plesiosaurs. The functional analogues further show, that most lower arm/leg and hand/foot flexors and extensors should be reconstructed based on the EPB taxa for plesiosaurs. Contrastingly, extensors responsible for digital spreading of webbed hands and feet should instead not be reconstructed.

Muscle reconstructions

Foreflipper musculature

Ligaments of the pectoral girdle and limb

Araújo & Correia (2015) reconstructed a scapulohumeral ligament in the plesiosaur pectoral girdle with which this study agrees. The scapulosternal ligament they reconstructed takes a ventral course in their reconstructions, although it is reported to take a course dorsal to the shoulder girdle in extant lepidosaurs (compare to Russell & Bauer, 2008 Figure 1.8, p. 97, or Figure 1.25 p. 237). The current study refrains from reconstructing a scapulosternal ligament for plesiosaurs, because during its course it would mostly lie on or wrap around the surface of the dorsal pectoral girdle. Because the course the ligament would take is identical with the plane the restructured pectoral girdle of plesiosaurs extends in, it is likely that the ligament is reduced because it has lost its function and muscles originating from it have been shifted onto adjacent bony areas.

We reconstructed an extensor retinaculum, although this ligament is weakly supported by the EPB. This is because because it interconnects to the flipper twisting mechanism (s. below) and mirrors the plesiosaur hindflipper. An extensor retinaculum, a ligament which ties the extensors at about the hight of the dorsal wrist, was reported for lepidosaurs (Russell & Bauer, 2008, p. 263, Figure 1.27). It is a derivative of the subdermal fascia. The extensor retinaculum was neither reported for crocodylians (Meers, 2003), nor for Testudines (Walker, 1973). The figures by Russell & Bauer (2008) suggest an attachment at a relatively similar position as the flexor retinaculum on the ventral wrist, from radiale to ulnare.

A ventral flexor retinaculum is well supported by the EPB and is therefore reconstructed for Cryptoclidus eurymerus (IGPB R 324). This is based on the ventral annular ligament or flexor retinaculum in crocodilians (Meers, 2003), in turtles (Abdala, Manzano & Herrel, 2008), and in lepidosaurs (Russell & Bauer, 2008 for Iguana, Abdala, Manzano & Herrel, 2008 for Liolaemus). The flexor retinaculum attaches to the radiale and the pisiform in lepidosaurs and it connects with the aponeurosis from which flexores digiti breves originate (Abdala & Moro, 2006). Further, a similar arrangement of ligaments (intermetacarpal ligaments and metacarpodigital ligaments) that connect successive metacarpals and metacarpals with phalanx I of bordering digits as described in the lepidosaur carpus and metacarpus (Russell & Bauer, 2008 Figure 113, p. 119) is reconstructed for plesiosaurs as part of the flipper twisting mechanism (s. below).

Pectoral muscles

Dorsal group

Musculus latissimus dorsi (ld)

-latissimus dorsi (Walker, 1973; Zaaf et al., 1999; Meers, 2003; Russell & Bauer, 2008; Suzuki & Hayashi, 2010; Anzai et al., 2014)

-teres major (Walker, 1973; Meers, 2003; Suzuki & Hayashi, 2010)

Teres major is considered a derived portion of latissimus dorsi (Remes, 2007). It was treated together with latissimus dorsi, because of their closely associated insertion tendons (Walker, 1973; Meers, 2003; personal observation of Caretta caretta dissection). It is not reported by Russell & Bauer (2008), Zaaf et al. (1999), and Anzai et al. (2014). Tm lies beneath ld (Table 3).

Crocodiles and lepidosaurs suggest that ld arises from at least the first to the sixth dorsal vertebra in plesiosaurs (Meers, 2003; Russell & Bauer, 2008), but it may have extended further caudally along the vertebral column up to at least the 12th dorsal vertebra based on the EPB. In crocodiles and lepidosaurs, ld originates from the neural spines of the vertebral column by an aponeurosis (Meers, 2003; Russell & Bauer, 2008). In crocodiles, the ld origin area extends from approximately the first dorsal vertebra caudally to the sixth rib (Meers, 2003). In lepidosaurs, the aponeurosis of origin of the ld begins with the first cervical vertebra. The number of vertebrae involved in the origin area of this muscle varies across taxa from three to four in chameleons to 12 in e.g., Sphenodon and Iguana (Russell & Bauer, 2008). Turtles pose the exception, in which the muscle origin is on the dorsal scapula and has spread laterally onto the carapace reaching the posterior border of the first peripheral plate (Walker, 1973).

The ld insertion was reconstructed on the anterodorsal tuberosity of the plesiosaur humerus (as supported by all three taxa) associated with part of the very rugose and deeply striated muscle scar on the tuberosity. Ld attachment site is distally to scs (as in all three EPB taxa) and anterior to shp (as in crocodiles and lepidosaurs) (Fig. 2B). In lepidosaurs, crocodiles, and turtles, the ld attachment is on the proximal dorsal humerus (Walker, 1973; Meers, 2003; Russell & Bauer, 2008; Suzuki & Hayashi, 2010). In crocodiles and some lepidosaurs, it is placed anteriorly to shp (Zaaf et al., 1999; Meers, 2003; Russell & Bauer, 2008; Suzuki & Hayashi, 2010). The insertion of ld is positioned on the humerus posteriorly to the deltoid insertions, distally to the scs insertion and distally bordered by the humeral tb head in all three taxa (Walker, 1973; Meers, 2003; Russell & Bauer, 2008; Suzuki & Hayashi, 2010).

Musculus subcoracoscapularis (scs)

-subcoracoscapularis (Russell & Bauer, 2008)

-subscapularis (Walker, 1973; Meers, 2003; Suzuki & Hayashi, 2010)

Term coined for most plesiomorphic (origin areas on coracoid and on scapula) taxon employed in this study, Lepidosauria, is given priority over the probably derived states in crocodilians (Meers, 2003; Suzuki & Hayashi, 2010) and Testudines (Walker, 1973), which show only a scapular portion (Table 3).

A scapula portion of the scs is well supported for plesiosaurs by all three taxa, while a coracoid portion is supported by lepidosaurs. Yet, a large coracoid portion is possible, if one considers that in lepidosaurs and turtles the dorsal coracoid is well covered by muscles (in crocodilians merely to a lesser degree) (Fig. 2B). In crocodiles, scs originates from the medial scapula (Meers, 2003; Suzuki & Hayashi, 2010) and from the lateral scapular blade (Walker, 1973). In lepidosaurs, scs takes its origin on most of the medial and dorsal scapulocoracoid and spreads partially around the scapula onto its posterolateral side (Russell & Bauer, 2008; Anzai et al., 2014).

In plesiosaurus, the scs insertion area was reconstructed on the posterodorsal proximal plesiosaur humerus as in all three taxa, relatively closer to the glenoid than the ld insertion. It was correlated with part of the large, rugose, and deeply striated muscle scar on the dorsal tuberosity of the plesiosaur humerus (Fig. 2B). In lepidosaurs, scs inserts posterodorsally into the proximal humerus (into the lesser tubercle), in crocodiles (into the medial protuberance), and in turtles (into the medial process) (Walker, 1973; Meers, 2003; Russell & Bauer, 2008; Suzuki & Hayashi, 2010). In turtles, the scs insertion is bordered anterodistally by the ld insertion and posteriorly by the cl insertion (Walker, 1973). In lepidosaurs and crocodilians scs is the most posterior insertion of a pectoral muscle on the humerus (Meers, 2003; Russell & Bauer, 2008; Suzuki & Hayashi, 2010). In all three taxa scs inserts proximally to the ld insertion on the humerus (Walker, 1973; Meers, 2003; Russell & Bauer, 2008; Suzuki & Hayashi, 2010).

Musculus scapulohumeralis posterior (shp)

-scapulohumeralis posterior (Zaaf et al., 1999; Russell & Bauer, 2008)

-scapulohumeralis caudalis (Meers, 2003; Suzuki & Hayashi, 2010)

We chose the term shp because the term is used in the lepidosaur articles this work is based on, paying tribute to lepidosaurs showing the more plesiomorphic condition than Crocodylia (in which the anterior part is reduced). Has not been observed in Testudines (Walker, 1973) (Table 3).

In plesiosaurs, shp arises from the posterior edge of the scapula and from the lower part of the small scapular blade, spreading around onto the dorsal and ventral surface (Fig. 2B). Dorsally it is bordered by scs as in crocodiles and ventrally by ds as in lepidosaurs. The origin surface of shp is located posteriorly on the lower half of the scapula in crocodiles and some lepidosaurs (Sphenodon (Russell & Bauer, 2008), Varanus (Jenkins & Goslow, 1983)) and reaches around onto the medial and lateral surface of the scapula (Jenkins & Goslow, 1983; Meers, 2003; Russell & Bauer, 2008; Suzuki & Hayashi, 2010). Towards the glenoid shp borders the tb origin in crocodiles and lepidosaurs (Jenkins & Goslow, 1983; Meers, 2003; Russell & Bauer, 2008; Suzuki & Hayashi, 2010). Medially, shp flanks scs in crocodiles (Meers, 2003; Suzuki & Hayashi, 2010). On the lateral lepidosaur scapula shp is paralleled anteriorly by ds (Jenkins & Goslow, 1983).

In plesiosaurs, a small insertion site was reconstructed on the proximodorsal humerus, posteriorly on the humeral tuberosity. It was, like scs and ld, associated with the heavily striated, rugose large muscle scar on the humeral tuberosity (Fig. 2B). Shp inserts differently in lepidosaurs and crocodiles. In the former it attaches to the lesser tubercle of the humerus posterodorsally (Jenkins & Goslow, 1983; Russell & Bauer, 2008) and in the latter its insertion area is large and on the proximodorsal humerus (Meers, 2003; Suzuki & Hayashi, 2010). In crocodiles and lepidosaurs, shp inserts more distally than scs, posterior and at about the same level as the ld, and posterior to ds (Meers, 2003; Russell & Bauer, 2008; Suzuki & Hayashi, 2010).

Musculus deltoideus clavicularis (dc)

-deltoideus clavicularis (Walker, 1973; Meers, 2003; Suzuki & Hayashi, 2010)

-clavodeltoideus (Zaaf et al., 1999; Russell & Bauer, 2008; Anzai et al., 2014)

The term deltoideus clavicularis was chosen because it is the most commonly used one in recent works on Sauropsida (Walker, 1973; Meers, 2003; Suzuki & Hayashi, 2010) (Table 3). In the following chapters, muscle names will only be discussed, if the authors did not decide to give a muscle the name that is most commonly used in literature.

The origin area of dc is on the very reduced clavicular remains ventrally and posteriorly in plesiosaurs (Fig. 2A). Anteriorly to dc attaches visceral arch musculature (Russell & Bauer, 2008) which will not be further discussed in this paper as it is beyond the scope of this work. Dc arises from the ventral, dorsal, and medial clavicula in lepidosaurs (Russell & Bauer, 2008). In Testudines, dc originates from the dorsal acromion (Walker, 1973). Due to loss of the clavicula in crocodiles, dc arises from the anterolateral scapula (Meers, 2003; Suzuki & Hayashi, 2010). In lepidosaurs and turtles, this is the most anterior muscle origin area of a locomotor muscle on the ventral pectoral girdle (Walker, 1973; Wyneken, 2001; Russell & Bauer, 2008).

For description of the insertion of dc in plesiosaurs, turtles, crocodiles and lepidosaurs please view the section on the insertion of ds below (Fig. 2B).

Musculus deltoideus scapularis (ds)

-scapulodeltoideus (Zaaf et al., 1999; Russell & Bauer, 2008; Anzai et al., 2014)

-deltoideus scapularis (Walker, 1973; Meers, 2003; Suzuki & Hayashi, 2010)

In plesiosaurs, ds originates on the anteroventral and lateral scapula (supported by all three taxa) extending posteriorly towards the scapular glenoid portion (Fig. 2A). Its attachment site, and that of dc, on the pectoral girdle is demarcated posteriorly by a ridge that expands from the body midline anteriorly posterolaterally to the glenoid. In turtles, lepidosaurs, and crocodiles, ds originates from the ventral anterolateral scapula (Walker, 1973; Russell & Bauer, 2008; Meers, 2003; Suzuki & Hayashi, 2010) and from the suprascapula adjacently in lepidosaurs (Russell & Bauer, 2008) and in crocodiles (Meers, 2003; Suzuki & Hayashi, 2010).

Both deltoid muscle bellies insert by a common tendon in plesiosaurs, as suggested by turtles and lepidosaurs. In plesiosaurs, the insertion site has been reconstructed on the anterior plesiosaur humerus shaft, adjacently to all other pectoral girdle musculature. Ds is partially associated with the anteroventral rugose muscle scar at approximately humeral mid-shaft (Fig. 2B). Ds inserts via a shared tendon with dc into the deltopectoral crest in Testudines (Walker, 1973) and lepidosaurs (Russell & Bauer, 2008). In crocodiles, the insertion tendons of ds and dc separately insert into the anterodorsal deltopectoral crest of the humerus proximal to the dc insertion (Meers, 2003; Suzuki & Hayashi, 2010).

Musculus triceps brachii (tb)

-triceps complex: subdivision into scapular head, coracoid head, lateral humeral head, and medial humeral head (Russell & Bauer, 2008)

-triceps (Zaaf et al., 1999)

-triceps brachii (subdivision into: triceps longus lateralis (Meers, 2003; Suzuki & Hayashi, 2010), triceps longus caudalis (Meers, 2003)–longus medialis (Suzuki & Hayashi, 2010), triceps brevis cranialis, triceps brevis intermedius, triceps brevis caudalis (Meers, 2003; Suzuki & Hayashi, 2010)

-triceps brachii: humeral head and scapular head (Walker, 1973)

If taking crocodiles and Iguana iguana into consideration, an origin posterior and anterior to the glenoid is possible in plesiosaurs (Figs. 2A and 2B). This is because the scapular blade has been displaced cranially relative to the glenoid and the coracoid relatively posteriorly (please view Araújo & Correia, 2015 for discussion of homology of the sauropterygian pectoral girdle). If taking Testudines and chameleons (Lepidosauria) into account, which are both possibly better functionally comparable to plesiosaurs because both have a stiffened trunk region, only the scapular origin of the tb remains. Also, a restrictive anteroposterior function may be obsolete in plesiosaurs, as the glenoid shape seems to restrict anteroposterior motion of the humerus already. All extant EPB groups share a tb origin area on the scapula dorsally, just above the glenoid (Walker, 1973; Jenkins & Goslow, 1983; Meers, 2003; Russell & Bauer, 2008; Suzuki & Hayashi, 2010). Lepidosaurs may have a second tb head which arises from the sternoscapular ligament (Jenkins & Goslow, 1983; Russell & Bauer, 2008). Crocodiles have a second and third tendinous tb origin on the posterolateral scapula and the posteromedial coracoid just below the glenoid (Meers, 2003; Suzuki & Hayashi, 2010). In lepidosaurs, the two heads are involved in a complex sling mechanism (also called the cruciate ligaments of the shoulder joint) that spans the insertion tendon of ld. The sling mechanism consists of four ligaments (cranio-dorsal, caudo-dorsal, cranio-ventral, and caudo-ventral ligament). The cranio-dorsal ligament is partially joined by a triceps tendon. The caudo-dorsal and the caudo-ventral ligament are fused with the joint capsule. In lepidosaurs, the two tb heads are probably involved in reducing anteroposterior humeral movement (Jenkins & Goslow, 1983; Russell & Bauer, 2008). The tb head from the coracoid is lost in chameleons and therefore the sling mechanism is reduced (see Russell & Bauer, 2008 for review). Russell & Bauer (2008) suggest that this might be due to a loss of the restricting mechanism in chameleons. We suggest, it could also be connected to a loss in lateral undulation. This is because not only chameleons have only one tb head which arises from a region just dorsally of the glenoid, but also Testudines (Walker, 1973; Russell & Bauer, 2008). In crocodiles, the two additional tb tendons join into a common tendon distally (Meers, 2003; Suzuki & Hayashi, 2010).

Origin on humerus:

The humeral tb origin was on the dorsal humerus (as in turtles, lepidosaurs, and crocodilians) distal to the proximal pectoral musculature adjacent to the extensor origins in plesiosaurs (Fig. 2B). It can be correlated with the tendentially fan-shaped, striated, and rugose dorsal distal surface of the plesiosaur humerus. This osteological correlate covers the shaft and delineates the anterior and posterior distal expansions of the humerus. In recent sauropsids, several portions of the humeral head are often recognized (medial and lateral head in lepidosaurs (Zaaf et al., 1999; Russell & Bauer, 2008) and triceps brevis cranialis, t. b. intermedius, and t. b. caudalis in crocodiles (Meers, 2003; Suzuki & Hayashi, 2010). In this study it was impossible to find evidence for muscle portions, so it was reconstructed undivided. In all three taxa it arises from a large origin area situated on the dorsal humerus distal to the insertions of the proximal pectoral musculature and proximal to or reaching distally the most proximal origins of the brachial and antebrachial extensors (Walker, 1973; Zaaf et al., 1999; Meers, 2003; Russell & Bauer, 2008; Suzuki & Hayashi, 2010). In crocodiles, the tb origin on the humerus spreads around the humeral shaft anteriorly and posteriorly, thus the antagonistic bb on the ventral humerus is markedly smaller (Meers, 2003; Suzuki & Hayashi, 2010).

The tb insertion in plesiosaurs is on the posterodorsal edge of the ulna and adjacent bony areas, due to the lack of an olecranon, according to all three extant EPB taxa (Fig. 2B). The insertion area of tb is posterodorsally on the olecranon of the ulna via a common tendon in lepidosaurs (Zaaf et al., 1999; Russell & Bauer, 2008; Anzai et al., 2014), turtles (Walker, 1973), and crocodiles (Meers, 2003; Suzuki & Hayashi, 2010

Ventral group

Musculus pectoralis (p)

No synonyms employed in articles used in this study (Walker, 1973; Zaaf et al., 1999; Meers, 2003; Russell & Bauer, 2008; Suzuki & Hayashi, 2010; Anzai et al., 2014; for a list of synonyms see Discussion in Remes, 2007). Subdivisions are common (see Remes, 2007 for review) (Table 3).

The p origin often spreads onto different skeletal elements in various tetrapod groups and keep its relative position in the body (s. below), therefore it was reconstructed in Cryptoclidus eurymerus (IGPB R 324) along the midline of the scapula and coracoid along the ventral crest that each element of both side forms at the body midline (Fig. 2A), superficial to sc and cb. Substantiated by the EPB, it is possible that it might have spread onto adjacent gastralia caudally as in crocodilians. In tetrapods in general, p is a large fan-shaped muscle, often subdivided into various portions, which arises ventrally from the middle axis of the body and often spreads posteriorly onto adjacent bony or cartilagous elements: In lepidosaurs and crocodiles, it originates from the sternal elements (Zaaf et al., 1999; Meers, 2003; Russell & Bauer, 2008; Anzai et al., 2014). Additionally, it arises from the lepidosaur interclavicula (Zaaf et al., 1999; Russell & Bauer, 2008; Anzai et al., 2014) and from the crocodilian thoracal ribs (Meers, 2003). As there is no interclavicula or sternum in turtles, p has spread onto the plastron. The attachment surface is situated posterior to the ligamentous articulation of the acromion to the plastron and extends posteriorly and curves in an arc laterally (Walker, 1973; personal observation).

In plesiosaurs, the muscle insertion of p was on the posteroventral proximal humerus associated with part of the rugose muscle scar on the ventral humerus (Fig. 2A). This is supported by all three extant EPB taxa. The attachment site of p via a large tendon is in crocodiles, lepidosaurs, and turtles on the deltopectoral crest and relatively posterodistally to the attachment site of sc and anteriorly to coracobrachialis insertions (Walker, 1973; Meers, 2003; Russell & Bauer, 2008; Suzuki & Hayashi, 2010).

Musculus supracoracoideus (sc)

-supracoracoideus (Walker, 1973; Russell & Bauer, 2008)

-supracoracoideus + coracobrachialis brevis dorsalis (Meers, 2003; Suzuki & Hayashi, 2010)

-subdivision into supracoracoideus longus, intermedius, brevis in crocodilians (Meers, 2003; Suzuki & Hayashi, 2010)

The sc origin site was on the posterior portion of the scapula in plesiosaurs (supported by lepidosaurs and crocodilians) behind the ridge that demarcates the posterior border of ds. Sc also arises from the anterior portion of the coracoid (as in lepidosaurs and crocodilians) posteriorly bordered by a bulging rounded ridge that runs from the posteroventral glenoid medially towards the body midline. It is also presumed that it covers the coracoid foramen (Fig. 2A) (Araújo & Correia, 2015), as it is known to cover two fenestrae in the lepidosaur shoulder girdle (Russell & Bauer, 2008). In none of the three groups used for EPB (Walker, 1973; Meers, 2003; Russell & Bauer, 2008; Suzuki & Hayashi, 2010), sc origin area contacts the glenoid, so in the plesiosaur muscle reconstruction it does not either. In Crocodylia, sc originates from the anteroventral and anterodorsal coracoid, and the anterolateral and anteromedial scapula (Meers, 2003; Suzuki & Hayashi, 2010). In Testudines, sc arises from the ventral side of the coracoid and scapula (Walker, 1973). In lepidosaurs, sc originates usually from the anteroventral coracoid (Russell & Bauer, 2008) while a scapular origin poses the exception (Russell & Bauer, 2008). The sc origin lies in lepidosaurs and crocodiles anteriorly to cb and cl (Meers, 2003; Russell & Bauer, 2008; Suzuki & Hayashi, 2010). The pectoral girdle of turtles seems to show the derived condition.

The sc insertion is on the anterior to anteroventral proximal plesiosaur humerus (Fig. 2A), anteriorly to the p, cl and cb insertions (as in all three EPB taxa) but at about the same level as the deltoid insertion (as in crocodiles). This is due to a relative displacement of the sc insertion further distally determined by its correlation with part of the ventral rugose muscle scar on the plesiosaur humerus. The insertion of sc is anteroventrally proximally on the proximal border of the deltopectoral crest on the humerus in Lepidosauria (Russell & Bauer, 2008) and Crocodylia (Meers, 2003; Suzuki & Hayashi, 2010). Contrastingly, in Testudines the insertion is positioned proximally to the deltopectoral crest but anteriorly extending slightly dorsally and more ventrally (Walker, 1973). In turtles and lepidosaurs, the insertion of sc is proximal to the deltoid insertion (Walker, 1973; Russell & Bauer, 2008). In crocodiles, the proximal extension of the deltoids reaches the same level as the sc insertion (Meers, 2003; Suzuki & Hayashi, 2010). It is positioned anteriorly to the cb and cl insertions (Walker, 1973; Meers, 2003; Russell & Bauer, 2008; Suzuki & Hayashi, 2010) and proximal to the p insertion (lepidosaurs, turtles) or at the same level as the p (Russell & Bauer, 2008).

Musculus coracobrachialis brevis (cb)

-coracobrachialis brevis (Walker, 1973; Russell & Bauer, 2008)

-coracobrachialis (Zaaf et al., 1999)

-coracobrachialis brevis ventralis (Meers, 2003; Suzuki & Hayashi, 2010)

Nomenclature by Russell & Bauer (2008) was chosen because it describes the geometry of cb and cl well (Table 3).

In plesiosaurs, the origin area of cb is on the ventral coracoid surface as in Crocodylia and lepidosaurs. It originates posterior to sc, anterior to cl, covering about four-fifths of it, presuming that the state found in the turtle pectoral girdle is highly derived due to its position inside the rib cage and the shell (Nagashima et al., 2012). Cb origin is placed posteriorly behind a broad bulging ridge that expands from the posteroventral glenoid to the body midline (Fig. 2A). Cb takes its large origin on the anteroventral coracoid posterior to the sc and reaches far back to meet the cl origin posteriorly in Crocodylia (Meers, 2003; Suzuki & Hayashi, 2010) and lepidosaurs (Russell & Bauer, 2008). In Testudines, cb attaches to the proximal part of the posterolateral rim of the coracoid posterior to the glenoid (Walker, 1973).

The EPB suggests that an insertion similar to the state seen in crocodiles and turtles is likely in plesiosaurs, i.e., proximal to b insertion on the ventral humerus. This is because the insertion of cb can be correlated with the rough rugosities on the ventral to posteroventral plesiosaur humerus proximal to cl insertion (Fig. 2A). Cb attaches posteroventrally to the humeral head and extends proximodistally in lepidosaurs (Zaaf et al., 1999; Russell & Bauer, 2008) and into the ventral intertrochanteric fossa in Crocodylia and Testudines (Walker, 1973; Meers, 2003; Suzuki & Hayashi, 2010). P and sc insertions are positioned anteriorly to cb in all three taxa (Walker, 1973; Meers, 2003; Suzuki & Hayashi, 2010; Abdala et al., 2014). In turtles and crocodiles, the b insertion lies distal to it (Walker, 1973; Meers, 2003; Suzuki & Hayashi, 2010) while it borders b proximoposteriorly in lepidosaurs (Russell & Bauer, 2008).

Musculus coracobrachialis longus (cl)

-coracobrachialis longus (Zaaf et al., 1999; Russell & Bauer, 2008; Anzai et al., 2014)

-coracobrachialis magnus (Walker, 1973)

Name was chosen for the same reason as discussed for cb. This muscle is not reported in Crocodylia (Table 3).

For Cryptoclidus eurymerus (IGPB R 324) the origin area of the cl was on the posterior coracoid (Fig. 2A) as reported for lepidosaurs and along with the other parts of this muscle, cb, which also derives from the lateral side of the pectoral girdle in crocodiles (Meers, 2003; Suzuki & Hayashi, 2010) and lepidosaurs (Russell & Bauer, 2008). In lepidosaurs, cl originates from the ventral posterior coracoid, posteriorly to cb (Jenkins & Goslow, 1983; Russell & Bauer, 2008). In Testudines, cl covers most of the dorsal coracoid (Walker, 1973). The state of the pectoral girdle of turtles is considered the derived state, in comparison to Lepidosauria, due to its placement inside the shell and the rib cage (Nagashima et al., 2012).

In plesiosaurs, the insertion site of cl was on the posteroventral and distal humerus shaft (Fig. 2A), similarly to lepidosaurs (Russell & Bauer, 2008). This is due to the observation of a rugose muscle scar that expands relatively far distally along the posterior shaft of the plesiosaur humerus. In lepidosaurs, the insertion of cl is situated posteroventrally on the humerus, distal to the cb insertion, and extends far distally, reaching almost the epicondyle in lepidosaurs (Russell & Bauer, 2008). Cl attaches posteroventrally into the turtle humerus and proximally into the medial process, posterior to the cb insertion (Walker, 1973).

Musculus biceps brachii (bb)

-biceps (Zaaf et al., 1999)

-biceps brachii (Meers, 2003; Russell & Bauer, 2008; Suzuki & Hayashi, 2010; Anzai et al., 2014)

-biceps brachii, subdivisions into biceps profundus and biceps superficialis (Walker, 1973)

In plesiosaurs, the bb origin area was on the posterior coracoid (Fig. 2A), supported by lepidosaurs and partially Testudines (Meers, 2003; Suzuki & Hayashi, 2010). Bb originates from the posterior coracoid in lepidosaurs posterior to cl and cb (Russell & Bauer, 2008) and from the posterolateral coracoid in Testudines. In turtles, cl lies posterior to cb. The origin area of bb is bordered ventrally by sc and dorsally by cl (Walker, 1973). In crocodiles, it arises from the anterior coracoid placed between suparcoracoideus anteriorly and cb posteriorly. The crocodilian condition is thought to be the derived one because crocodiles have become secondarily aquatic and are able to sprawl, but also to employ a “high walk” (Reilly & Elias, 1998).

Musculus brachialis (b)

-brachialis anticus (Russell & Bauer, 2008; Anzai et al., 2014)

-brachialis inferior (Walker, 1973; Meers, 2003)

-brachialis (Zaaf et al., 1999; Suzuki & Hayashi, 2010)

The term brachialis will be employed in this work according to suggestion of Remes (2007) (Table 3).

In plesiosaurs, the attachment surface of b was placed on the ventral surface of the humerus distally to the shoulder musculature (Fig. 2A), like all three extant EPB taxa suggest, bordering extensors and flexors. The very “veiny” or tendentially fan-shaped striated and slightly rugose surface in this area suggests an association with a muscular covering. A similar arrangement, where the flexors spread proximally onto the humeral shaft was observed in Caretta caretta (Cheloniidae) by dissection (Krahl et al., 2019) in comparison to non-marine turtles by Walker (1973). In lepidosaurs and turtles, b covers most of the ventral humerus shaft. Proximally it is flanked by shoulder musculature and distally it extends to and partially proximally borders the extensors and flexors of the antebrachium and brachium which originate from the ect- and entepicondyle (Walker, 1973; Zaaf et al., 1999; Russell & Bauer, 2008; Anzai et al., 2014). In crocodiles, b is distinctly smaller than the antagonistic tb, which reaches around the humeral shaft anteriorly and posteriorly onto the ventral side and displaced the b origin somewhat anteriorly (Meers, 2003; Suzuki & Hayashi, 2010).

In plesiosaurs, the insertion of the common bb and b tendon could be reconstructed on the proximal posterior radius and anterior proximal ulna (Fig. 2A), a placement on their shafts is impossible due to the derived bone shapes radius and ulna have in plesiosaurs. An attachment solely on the radius is just as well supported by the EPB (Fig. 2A). In extant sauropsids, bb and b insert by a common tendon which attaches to either radius or radius and ulna. In turtles, they insert into the posterior radius and anterior ulna at about mid-length on shaft (Walker, 1973). Zaaf et al. (1999) and Russell & Bauer (2008) report that in geckos and Iguana iguana the common tendon attaches to the posterior radius and the anterior ulna, too but more proximally and may even be associated with the elbow joint capsule (Russell & Bauer, 2008). Contrastingly, in crocodiles (Meers, 2003; Suzuki & Hayashi, 2010) and in various Anolis species (Anzai et al., 2014), the insertion tendon only attaches to the proximal radius.

Antebrachial muscles

Dorsal group

Musculus extensor carpi ulnaris (ecu)

-extensor carpi ulnaris (Walker, 1973; Russell & Bauer, 2008)

-extensor ulnaris (Suzuki & Hayashi, 2010)

-flexor ulnaris (Meers, 2003)

Here we go along with the homology established by Suzuki & Hayashi (2010) and not with Remes (2007) for flexor ulnaris of Meers (2003) (Table 3).

The attachment area of ecu is on the anterodorsal ectepicondyle of the plesiosaur humerus (Fig. 2B), as crocodiles and lepidosaurs suggest. Ecu origin was reconstructed to be the most proximal extensor origin on the plesiosaur humerus so that it meets the criteria discussed above about the overall arrangement in all Sauropsida of the extensors originating from the humerus. Ecu originates tendinously from the ectepicondyle of the humerus in lepidosaurs, crocodiles, and turtles (Walker, 1973; Meers, 2003; Russell & Bauer, 2008; Suzuki & Hayashi, 2010). In lepidosaurs the origin area is situated dorsally (Russell & Bauer, 2008), in crocodiles anterodorsally (Meers, 2003; Suzuki & Hayashi, 2010), and in turtles anteroventrally (Walker, 1973). Additionally, ecu also arises in lepidosaurs from the olecranon of the ulna dorsally (Russell & Bauer, 2008). In turtles, the ecu origin lies slightly posterodistally to that of sl and ecr and in between the origins of sl and ecr proximally and edc (Walker, 1973). In crocodilians, ecu is situated anterodorsally just above the elbow joint capsule (Meers, 2003; Suzuki & Hayashi, 2010). In lepidosaurs, ecu is the most distal extensor (Russell & Bauer, 2008).

In plesiosaurs, an attachment on the pisiform (according to turtles and lepidosaurs), or adjacent areas is probable and an attachment on the ulna (according to crocodiles and turtles) seems likely because this muscle displays a large insertion surface in all extant sauropsids (Fig. 2B). There is no designated pisiform in the plesiosaur flipper, but accessory ossicles are regularly found in a similar relative position of the carpus in plesiosaurs (see Krahl, 2021 for review). These accessory ossicles may well have been involved in the ecu insertion area. It is also possible, but only supported by Iguana iguana that ecu inserted into metacarpal V. This results in a relatively large muscle insertion area for ecu in plesiosaurs. In turtles and crocodiles, ecu inserts into a large area of the shaft of the ulna (Walker, 1973; Meers, 2003; Suzuki & Hayashi, 2010). In crocodiles the attachment area lies anterodorsally (Meers, 2003; Suzuki & Hayashi, 2010), while it covers most of the dorsal surface of the turtle ulna (Walker, 1973). Ecu inserts into the dorsal pisiform in lepidosaurs and turtles as well (Walker, 1973; Russell & Bauer, 2008). In lepidosaurs, a second muscle belly inserts into the shaft of metacarpal V posteriorly (Russell & Bauer, 2008). In turtles, lepidosaurs, and crocodiles, the ecu insertion is bordered proximoposteriorly by the tb insertion (Walker, 1973; Meers, 2003; Russell & Bauer, 2008; Suzuki & Hayashi, 2010). The m. supinator manus (sm) origin lies anteriorly to it in turtles and lepidosaurs (Walker, 1973; Russell & Bauer, 2008).

Musculus extensor digitorum communis (edc)

-extensor digitorum longus (Russell & Bauer, 2008)

-extensor digitorum communis (Walker, 1973; Suzuki & Hayashi, 2010)

-extensor carpi ulnaris (Meers, 2003)

In plesiosaurs, edc arises from the ectepicondyle of the humerus anterodorsally and in between the origin of ecu proximally and sl and ecr distally (Fig. 2B). This is based on the EPB, in which edc arises from the ectepicondyle dorsally in turtles and lepidosaurs (Walker, 1973; Abdala, Manzano & Herrel, 2008; Russell & Bauer, 2008) and anteriorly in crocodiles (Meers, 2003; Suzuki & Hayashi, 2010). In lepidosaurs, the origin of edc is closely associated with the elbow joint capsule and with the origin of ecr. In turtles, edc is the most distal extensor (Walker, 1973). It arises from a similar position from the humerus as in crocodilians (Meers, 2003; Suzuki & Hayashi, 2010). In lepidosaurs, this muscle arises in between sl and ecr proximally and ecu distally (Russell & Bauer, 2008).

Insertions of the edc in plesiosaurs, are reconstructed proximally and posterodorsally on metacarpal I, anterodorsally and posterodorsally on metacarpal II, III, and IV, and anterodorsally on metacarpal V (Fig. 2B), similar to recent turtles. The most elaborated insertion pattern of edc is seen in turtles, where the muscle belly gives way distally to a tendon. This tendon in turn splits up into four motor tendons that take course in between the digits. Here, the four motor tendon split up again into two tendons which finally insert by the following pattern into the metacarpals: posterodistally on the dorsal metacarpal I, antero- and posterodistally onto metacarpal II, III, IV and anterodistally onto metacarpal V (Walker, 1973; Abdala, Manzano & Herrel, 2008). In Iguana iguana, edc gives way to three tendons which attach to metacarpals II, III, and IV proximally and posterodorsally. Additional tendons and attachments on metacarpal I and V are observed in Sphenodon punctatum for the former and in chameleons for the latter (see Russell & Bauer, 2008 for review). For Crocodylia, a single attachment on the proximodorsal metacarpal II is described, except for C. acutus where edc inserts into the extensor fascia at the level of digit I (Meers, 2003). The state of edc as represented by turtles is assumed by the authors to represent the more plesiomorphic condition. This is because we assume that it is developmentally more likely and involves fewer evolutionary steps that in lepidosaurs and crocodiles muscular connections are lost, than that in lepidosaurs and turtles at least four times independently comparable muscular connections evolved anew convergently.

Musculus supinator longus and Musculus extensor carpi radialis (sl and ecr)

-supinator + extensor carpi radialis (Russell & Bauer, 2008)

-tractor radii (Walker, 1973; Meers, 2003; Suzuki & Hayashi, 2010) (s. Remes, 2007)

-extensor carpi radialis superficialis (Walker, 1973)

-extensor carpi radialis intermedius (Walker, 1973)

-extensor carpi radialis profundus (Walker, 1973)

-supinator (Meers, 2003)

-brachioradialis (Suzuki & Hayashi, 2010)

-supinator (Suzuki & Hayashi, 2010)

-abductor radialis (Meers, 2003)

-extensor carpi radialis longus (Meers, 2003; Suzuki & Hayashi, 2010)

It was decided to go along with the simplifying terminology of Russell & Bauer (2008), because it is impossible to reconstruct this muscle for a fossil in such detail as in e.g., turtles in which it has three subportions (Table 3).

The origin area of sl and ecr is on the ectepicondyle of the plesiosaur humerus (Fig. 2B); as suggested by all three extant EPB groups. The origin of sl and ecr is the most distal one of the three extensors that arise from the plesiosaur humerus, to achieve the general sauropsid fan-like arrangement described above. Sl and ecr arises from the humeral ectepicondyle anteriorly in turtles (Walker, 1973) and lepidosaurs (Russell & Bauer, 2008) and slightly anteroventrally in crocodiles (Meers, 2003; Suzuki & Hayashi, 2010). Sl and ecr originates from the humerus as most proximal extensor in turtles and lepidosaurs (Walker, 1973; Russell & Bauer, 2008).

The attachment surface for sl and ecr is on the dorsal and anterior plesiosaur radius, possibly extending onto the radiale (Fig. 2B), because no other muscles occupy the space. Sl and ecr inserts into the dorsal and anterior radius in lepidosaurs, turtles, and crocodiles (Walker, 1973; Meers, 2003; Russell & Bauer, 2008; Suzuki & Hayashi, 2010). In lepidosaurs, it also extends onto the radiale (Russell & Bauer, 2008).

Musculus supinator manus (sm)

-supinator manus (Walker, 1973; Russell & Bauer, 2008)

-extensor carpi radialis brevis pars radialis and pars ulnaris (Meers, 2003; Suzuki & Hayashi, 2010).

Sm is used as described by Remes (2007) (Table 3).

In plesiosaurs, the sm origin area is on the anterodorsal ulna (Fig. 2B), as in all three EPB taxa, and on the adjacent carpal element proximally and anteriorly, as observed in turtles and crocodilians. Sm arises from the anterodorsal edge of the ulna in turtles, lepidosaurs, and crocodiles anterior to ecu insertion (Walker, 1973; Russell & Bauer, 2008; Suzuki & Hayashi, 2010). The origin area of sm extends onto the proximal intermedium in Testudines (Walker, 1973). In crocodiles, it also originates from the posterodorsal and distal dorsal radius (Meers, 2003; Suzuki & Hayashi, 2010).

The insertion area for sm was reconstructed on the proximal and anterodorsal metacarpal I in plesiosaurs (Fig. 2B). The insertion was correlated with the anterior prominence on metacarpal I. In turtles and lepidosaurs, sm inserts into the proximal anterodorsal metacarpal I (Walker, 1973; Russell & Bauer, 2008) and to the radiale in crocodiles (Meers, 2003).

Ventral group

Musculus pronator teres (pte)

No synonyms in literature used in this study (Walker, 1973; Meers, 2003; Abdala, Manzano & Herrel, 2008; Russell & Bauer, 2008; Suzuki & Hayashi, 2010) (Table 3).

Pte is reconstructed to arise from the posteroventral surface of the plesiosaur humerus where it fans out and bends caudally (Fig. 2A). Pte is placed as the most proximal flexor arising from the plesiosaur humerus as in lepidosaurs and crocodiles, distally bordered by fcr. In turtles, lepidosaurs, and crocodiles, pte originates from the humeral entepicondyle posteroventrally (Walker, 1973; Meers, 2003; Abdala, Manzano & Herrel, 2008; Russell & Bauer, 2008; Suzuki & Hayashi, 2010). In crocodiles and lepidosaurs, its origin area is the most proximal of the flexors that originate on the humerus, distally followed by fcr (Meers, 2003; Russell & Bauer, 2008; Suzuki & Hayashi, 2010). In Testudines, pte is distally and ventrally situated, but at the same level as fcu origin area and proximally bordered by fdlf (Walker, 1973).

The insertion area of pte was reconstructed on the lower half of the plesiosaur radius (Fig. 2A). In turtles, lepidosaurs, and crocodiles, pte inserts into the ventral radius. In turtles, it inserts distally (Walker, 1973; Abdala, Manzano & Herrel, 2008), in lepidosaurs anteroventrally and for approximately half the distal length of the radius (Russell & Bauer, 2008), and in crocodiles it covers most of the ventral radius shaft (Meers, 2003).

Musculus flexor carpi ulnaris (fcu)

-epitrochleoanconeus (Russell & Bauer, 2008)

-flexor carpi ulnaris (Walker, 1973; Meers, 2003; Russell & Bauer, 2008; Suzuki & Hayashi, 2010)

Epitrochleoanconeus is a portion of flexor carpi ulnaris according to Remes (2007). The detailed differentiation into several muscle bellies was not undertaken (Table 3).

In plesiosaurs, fcu is reconstructed to arise as the distalmost flexor from the humerus, proximally bordered by fdlf (Fig. 2A). This is because fcu arises from the entepicondyle of the humerus posteroventrally in all three taxa used for the EPB (Walker, 1973; Meers, 2003; Russell & Bauer, 2008; Suzuki & Hayashi, 2010). Fcu is the most distal flexor that originates from the posterodorsal turtle humerus, at the same level as the pte origin area posteroventrally (Walker, 1973). In crocodylians and lepidosaurs, fcu arises as the most distal flexor from the humerus, proximally bordered by fdlf (Meers, 2003; Russell & Bauer, 2008; Suzuki & Hayashi, 2010).

Fcu insertion is reconstructed in plesiosaurs on the posterodistal ulna as supported by turtles and on the forming accessory ossicle (s. Andrews, 1910, which forms on the proximoposterior ulnare) which is in a similar position as the pisiform is in crocodiles, turtles, and lepidosaurs. An insertion into metacarpal V is favorable for flipper twisting in plesiosaurs. In turtles, crocodiles, and lepidosaurs, fcu inserts ventrally into the pisiform (Walker, 1973; Meers, 2003; Russell & Bauer, 2008). In lepidosaurs, fcu also attaches to metacarpal V (Russell & Bauer, 2008). In turtles and lepidosaurs fcu inserts into the posteroventral ulna (Walker, 1973; Russell & Bauer, 2008), anteriorly bordered by the ulnar origin area of fdlf as in crocodiles, lepidosaurs, and turtles (Walker, 1973; Meers, 2003; Russell & Bauer, 2008).

Musculus flexor digitorum longus (fdlf)

-flexor digitorum longus (Meers, 2003; Russell & Bauer, 2008)

-palmaris longus of turtles (Walker, 1973) is homologous to humeral head/s of crocodylians and lepidosaurs (Meers, 2003; Russell & Bauer, 2008)

The humeral head of fdlf is reconstructed to arise from the ventral ulnar epicondyle in plesiosaurs (Fig. 2A), as supported by lepidosaurs, crocodiles, and turtles but bordered by fcr proximally and by fcu distally as seen in lepidosaurs. The ulnar head arises in plesiosaurs from an extensive origin ventrally (Fig. 2A) as in turtles and lepidosaurs. Fdlf has two bellies, one of which arises posteroventrally from the entepicondyle of the humerus and the other one ventrally from the ulna (Walker, 1973; Meers, 2003; Russell & Bauer, 2008). The ulnar origin in crocodilians is confined to the distal ulna, but additionally a carpal muscle belly arises from the ulnar side (Meers, 2003). In crocodiles and lepidosaurs, fdlf arises from the humerus proximal to fcu (Meers, 2003; Russell & Bauer, 2008; Suzuki & Hayashi, 2010), but distally to fcr in lepidosaurs (Russell & Bauer, 2008) and to pte in crocodiles (Meers, 2003; Suzuki & Hayashi, 2010). In turtles, it originates distally to fcr but proximally to fcu and pte (Walker, 1973).

In plesiosaurs, it seems likely that, as in turtles and lepidosaurs, the common tendon of fdlf contributes to a flexor aponeurosis which sends five tendons to the terminal phalanx of digit I–V (Fig. 2A) (Walker, 1973; Russell & Bauer, 2008). Contrastingly, in crocodiles the tendon splits into three smaller tendons (Meers, 2003) which insert into the penultimate phalanges of digit I–III (Meers, 2003).

Musculus flexor carpi radialis (fcr)

No synonyms known for taxa and literature studied in this work (Walker, 1973; Russell & Bauer, 2008). This muscle is reduced in crocodilians (Meers, 2003; Remes, 2007) (Table 3).

In plesiosaurs, the origin area of fcr is on the entepicondyle of the humerus distal to pte origin and proximal to fdlf origin (Fig. 2A). In turtles and lepidosaurs, fcr originates posteroventrally from the entepicondyle of the humerus (Walker, 1973; Russell & Bauer, 2008). At its origin, this muscle is associated in lepidosaurs with pte, which arises proximal to it (Russell & Bauer, 2008). In turtles, fcr is the most proximal flexor arising from the humerus (Walker, 1973). Distally to fcr arises fdlf in lepidosaurs and turtles (Walker, 1973; Russell & Bauer, 2008).

Based solely on the EPB, the insertion type seen in turtles and lepidosaurs are equally likely for plesiosaurs. Yet, a lepidosaur-like insertion to metacarpal I is functionally favorable as it contributes to flipper twisting in plesiosaurs (Fig. 2A). In lepidosaurs, fcr inserts into the proximal metacarpal I (Russell & Bauer, 2008). In Testudines, fcr attaches to the anterodistal radiale-centrale and adjacently to the proximal distal carpal (Walker, 1973).

Manual muscles

Dorsal group

Musculi extensores digitores breves superficialis (edbs)

-extensores digitores breves superficialis (Russell & Bauer, 2008)

-extensores digitorum breves (Walker, 1973)

-extensor digitorum superficialis and extensor pollicis superficialis et indicus proprius (Meers, 2003)

The authors decided to choose extensores digitores breves superficiales after Russell & Bauer (2008) as it clarifies that this muscle group consists of a superficial and a deeper muscle layer (Table 3).

EPB suggests to reconstruct the origin area of edbs is on the plesiosaur ulnare (Fig. 2B) as seen in turtles and lepidosaurs. Although, an origin on ulna, radiale, and intermedium is also possible because no muscles originate or insert here otherwise in plesiosaurs, except for ecu which inserts into part of the ulnare. In turtles and lepidosaurs, edbs arise dorsally from the ulnare (Walker, 1973; Russell & Bauer, 2008) and from the distal ulna in turtles (Walker, 1973). In crocodiles, muscles to digit I, II, and III originate from the radiale, the muscle for digit IV from both, radiale and ulnare, and the muscle for digit V originates from ulnare and distal most ulna (Meers, 2003).

In plesiosaurs, the tendon insertions are proximodorsally on the terminal phalanx of each digit (Fig. 2B) as suggested by crocodiles and lepidosaurs. All five tendons insert into the proximal dorsal terminal phalanges of digit I–V in crocodiles and lepidosaurs (Meers, 2003; Russell & Bauer, 2008) and into the penultimate phalanges in turtles (Walker, 1973). Additionally, extensor pollicis superficialis et indicus proprius attach to the first phalanx of digit I and II.

Musculi extensores digitores breves profundi (edbp)

-extensores digitores breves profundi (Russell & Bauer, 2008)

-interossei dorsales (Walker, 1973)

-dorsometacarpalis (Abdala, Manzano & Herrel, 2008)

-extensor digitorum profundi (Meers, 2003)

Please view explanation for extensores digitores breves superficiales above (Table 3).

For plesiosaurs, an origin area on the metacarpals is supported by turtles and lepidosaurs (Walker, 1973; Abdala, Manzano & Herrel, 2008; Russell & Bauer, 2008), while an origin on adjacent carpal elements is supported by turtles and crocodiles (Walker, 1973; Meers, 2003). An origin on the metacarpals is reconstructed for plesiosaurs (Fig. 2B), but the origin areas might have been spread over the adjacent distal carpal elements, as they appear to be free of muscles yet. Both options are equally well supported, because none would change the line of action and therefore the muscle function. In lepidosaurs and turtles, edbp originate from the proximal dorsal metacarpals (Walker, 1973; Abdala, Manzano & Herrel, 2008; Russell & Bauer, 2008) and in turtles from bordering areas of the adjacent distal carpal I–V distally as well (Walker, 1973; Abdala, Manzano & Herrel, 2008). Origin areas of these muscles in crocodiles are quite complex: Extensor digiti III has three muscle bellies, while the other four only have one muscle belly. Extensor digiti I arises from the proximal anterodorsal metacarpal I and extensor digiti II from the proximal posterodorsal metacarpal I. Extensor digiti III originates from the ulnare and the radial distal carpal, and the metacarpal II. Extensor digiti IV arises from the proximal dorsal metacarpal III. Extensor digiti V originates from the distal carpal to metacarpal V. Involvement of origin areas with the ligaments of the carpus are common (Meers, 2003).

The insertions of edbp are reconstructed on the unguals of digit I–V in plesiosaurs (Fig. 2B). Both layers of extensores digitorum breves (superficialis and profundi) are reconstructed as they are necessary for digital extension and because they are well supported by the EPB. Yet, it is likely that both portions are fused or undifferentiated in plesiosaurs as observed in chelonioids due to a reduction of digital mobility (Walker, 1973). In crocodilians and lepidosaurs, edbp attach to the terminal phalanges in all five digits (Meers, 2003; Russell & Bauer, 2008) except for the fifth digit in crocodiles (Meers, 2003). How edbp attach in turtles is reported differently by Walker (1973), who states an attachment on the penultimate phalanx (Walker, 1973), and Abdala, Manzano & Herrel (2008) who report insertions on the terminal phalanges.

Ventral group

Musculi flexores digitorum superficialis (fdls)

-flexores digitores breves (Russell & Bauer, 2008)

-flexor brevis superficialis (Walker, 1973; Abdala, Manzano & Herrel, 2008)

-flexor digitorum brevis superficialis I–IV (Meers, 2003)

This study agrees on the established homology by Remes (2007) (Table 3).

In plesiosaurs, if fdls are differentiated from the fdlf, they arise from a tendinous structure and not from a bony area. Therefore, they are not marked in Fig. 2. Fdls originate from the annular ligament in lepidosaurs (Russell & Bauer, 2008) and the flexor retinaculum (which appears to be topologically homologous to the annular ligament in lepidosaurs) in turtles according to Abdala, Manzano & Herrel (2008). Walker (1973) describes it as originating from the flexor plate in turtles which would be similar to the situation described for Crocodylia (Meers, 2003).

For plesiosaurs, an insertion into the paenultimate phalanx paralleling fdlf displayed by turtles and partially by lepidosaurs is most likely (and possibly plesiosmorphic in diapsids) (see Fig. 2A fdlf insertions). The insertions of this muscle are highly variable across Sauropsida. In turtles (Walker, 1973), crocodilians (Meers, 2003), and lepidosaurs (Russell & Bauer, 2008) it may insert into digit I–IV and in turtles and lepidosaurs also to digit V). In crocodylians in attaches anteriorly and posteriorly to phalanx I in digit I, III, and IV, and to phalanx II in digit II (Meers, 2003). In turtles, the fdlf insertion may be into phalanx I, the paenultimate phalanx, or the tendon sheath of each digit. Additionally, the portion to digit I or digit V may be lost, or in Cheloniidae all but the portion to digit V are reduced (Walker, 1973). In lepidosaurs, the portion to digit I inserts into the proximal phalanx I, the ones to digit II–V insert into phalanx II. The tendon to digit III shows an additional insertion into phalanx III and the tendons to digit IV and V additionally insert into the penultimate phalanges (Russell & Bauer, 2008).

Musculus abductor digiti V (abdV)

-abductor digiti quinti (Russell & Bauer, 2008)

-abductor digiti minimi (Walker, 1973)

-abductor digitorum V (Abdala, Manzano & Herrel, 2008)

-abductor metacarpi V (Meers, 2003)

The term abductor digiti V by Remes (2007) will be followed due to reasons discussed in Remes (2007) (Table 3).

The origin area of this muscle is placed on the accessory ossicle adjacent to ulnare, respectively the adjacent ulnare (compare to Andrews, 1910, Figure C, p. 182) in the plesiosaur foreflipper (Fig. 2A), which are in a similar position as the pisiform in extant sauropsids. AbdV originates ventrally from the pisiform in crocodilians (Meers, 2003). In turtles, its origin is situated on the fifth distal carpal according to Walker (1973) and from the pisiform as reported by Abdala, Manzano & Herrel (2008). Contrastingly, in lepidosaurs it arises from the tendon of fcu and the annular ligament (Russell & Bauer, 2008).

AbdV inserts into the first phalanx of digit V proximally in plesiosaurs (Fig. 2A) based on lepidosaurs and turtles (Abdala, Manzano & Herrel, 2008; Russell & Bauer, 2008). The muscle inserts proximoposteriorly in turtles (Walker, 1973) and ventrally in lepidosaurs (Russell & Bauer, 2008). In crocodiles, its insertion area is situated along the shaft of metacarpal V (Meers, 2003).

Musculus abductor pollicis brevis (apb)

-abductor metacarpi I (Meers, 2003)

-abductor pollicis brevis (Walker, 1973; Abdala, Manzano & Herrel, 2008)

-anteriormost belly of interossei ventrales to digit I (Russell & Bauer, 2008)

The established name by Remes (2007), abductor pollicis brevis, will be followed here as well (Table 3).

In plesiosaurs, the origin surface of apb is on the plesiosaur radiale (Fig. 2A) as reported for crocodiles and lepidosaurs. In Testudines apb arises from the distal carpal adjacent to digit I (Walker, 1973) or from the distal radius (Abdala, Manzano & Herrel, 2008). In crocodilians it originates from the radiale anterodistally (Meers, 2003) and in lepidosaurs from a ligament at the level of the radiale and from distal carpal IV (Russell & Bauer, 2008).

The insertion site of apb is on phalanx I of digit I of the plesiosaur foreflipper (Fig. 2A) as reported for turtles and lepidosaurs because this is of advantage for individually flexing digit I during flipper twisting, although an attachment on metacarpal I is equally possible. In turtles and lepidosaurs, apb inserts into phalanx I of digit I anteroproximally (Walker, 1973; Russell & Bauer, 2008) and distally into metacarpal I in lepidosaurs, too (Russell & Bauer, 2008). In Crocodylia, it attaches to metacarpal I anteroproximally (Meers, 2003).

Musculus adductor digiti minimi (adm)

-adductor digiti minimi (Walker, 1973)

-flexor digiti quinti pars superficialis and profundus (Meers, 2003)

-(no actual name given) mesial lumbricales branch from metacarpal I to digit V (Russell & Bauer, 2008)

The term adductor digiti minimi will be used in the following text, to underline its different function to abdV (Table 3).

The origin of adm is on the plesiosaur radiale (Fig. 2A) as in crocodiles. This way it is ensured that this muscle takes a course similar to that observed across sauropsids from anteroproximal to posterodistal (Walker, 1973; Meers, 2003; Russell & Bauer, 2008). In lepidosaurs, adm originates from the anteroproximal metacarpal I (Russell & Bauer, 2008), in turtles from the distal carpal I and II (Walker, 1973; Abdala, Manzano & Herrel, 2008), and in crocodiles from the posterodistal radiale (Meers, 2003).

In plesiosaurs, the insertion of adm is on metacarpal V as suggested by all three taxa from the EPB (Fig. 2A), with a possible insertion into phalanx III as in crocodilians. Adm inserts anteroproximally into metacarpal V in lepidosaurs (Russell & Bauer, 2008), turtles (Walker, 1973; Abdala, Manzano & Herrel, 2008), and crocodiles (Meers, 2003). Additionally, in crocodilians it also attaches to phalanx three of digit V (Meers, 2003).

Hindflipper musculature

Ligaments of the pelvic girdle and limb

A puboischiadic ligament seems unlikely for plesiosaurs: First of all, in plesiosaurs there is no distinctive lateral ischial process/ischiadic tuberosity or lateral pubic process that provide the attachment surfaces for this ligament in turtles and lepidosaurs (Walker, 1973; Russell & Bauer, 2008). Second, the crocodilian ischium and pubis which lacks this ligament also does not show these processes. Third, a connection of the hypothetical attachment surfaces of the ilioischiadic ligament in plesiosaurs would lie in the same plane as the ischium and pubis themselves. Therefore, an ilioischiadic ligament is not reconstructed for plesiosaurs. An iliopubic and ilioischiadic ligament are possible in plesiosaurs, although the course of an iliopubic ligament would be quite close to the glenoid and the pubis eventually leaving not enough room for the large m. puboischiofemoralis internus (pi) portions that originate from the dorsal pelvic girdle and insert into the proximal femur to pass ventrally to it. Suitable osteological correlates, that are present in those taxa having these ligaments are lacking in plesiosaurs which speaks against their reconstruction in plesiosaurs. Therefore, none of the three ligaments is reconstructed for the plesiosaur pelvic girdle as the plesiosaur pelvic girdle does not show the morphologies correlated with their presence, although EPB would support all three of them. This is despite the EPB supporting relatively well the reconstruction of a puboischiadic, an ilioischiadic, and an iliopubic ligament in plesiosaurs. The presence of ligaments in the pelvic girdle varies considerably in Sauropsida: Testudines only have a puboischiadic ligament that connects the posterior ischial symphyseal region with its lateral process and with the lateral process of the pubis (Walker, 1973) which is similarly described for lepidosaurs by Russell & Bauer (2008). Contrastingly, this ligament is reduced in crocodilians (Romer, 1923). Lepidosaurs in general and crocodilians have an ilioischiadic ligament, which connects the lateral process of the ischium with the posterior ilium. Additionally, an iliopubic ligament, that spans from the anterior ilium to the lateral process of the pubis is found in crocodilians and lepidosaurs (Romer, 1923; Russell & Bauer, 2008). Yet, Sphenodon does not have an ilioischiadic ligament (Russell & Bauer, 2008).

A flexor retinaculum is reconstructed for plesiosaurs. This is founded on the EPB. Crocodilians have a flexor retinaculum which is associated with the tibiocalcaneal tendon (Suzuki et al., 2011). Similar structures, associated with the gastrocnemial heads are visible in lepidosaurs (compare to Russell & Bauer, 2008 Figure 1.43, p. 347) and turtles (compare to e.g., Walker, 1973 Figure 25, p. 71). For plesiosaurs, the authors reconstructed an extensor retinaculum/annular ligament. An extensor retinaculum/annular ligament that extends between pl and m. tibialis anterior (ta) is known from lepidosaurs and crocodilians (Russell & Bauer, 2008; Suzuki et al., 2011) but not from turtles (Walker, 1973).

Intermetatarsal and metatarsodigital ligaments, comparable to those of the plesiosaur foreflipper, are reconstructed for Cryptoclidus eurymerus. This is based on the description by Russell & Bauer (2008) for lepidosaurs. These ligaments are part of the flipper twisting mechanism in plesiosaurs (s. below).

Muscles of the pelvis

Dorsal group

Musculus iliotibialis (it)

-iliotibialis (Snyder, 1954; Zug, 1971; Walker, 1973; Suzuki et al., 2011)

-iliotibiales (Otero, Gallina & Herrera, 2010)

-ilio-tibialis (Romer, 1923; Gatesy, 1997)

The biggest consensus for plesiosaurs is found, if it arises from the anterodorsal lateral ilium, anteriorly to fte and dorsally to m. iliofemoralis (ife) (Fig. 3B) as supported by all three taxa. A posterior origin as in lepidosaurs and some turtle taxa is nonetheless possible. It originates across Sauropsida broadly similarly, i.e., from the dorsal rim of the lateral ilium (Snyder, 1954; Russell & Bauer, 2008; Otero, Gallina & Herrera, 2010; Suzuki et al., 2011). In crocodilians it arises by three heads from the approximately first two thirds of the dorsal ilium (Romer, 1923; Otero, Gallina & Herrera, 2010; Suzuki et al., 2011). In lepidosaurs it originates anteriorly fleshy and posteriorly aponeurotically from the lateral ilium (Russell & Bauer, 2008) dorsally to the origin areas of m. iliofibularis (ifi) and ife (Snyder, 1954; Russell & Bauer, 2008). In crocodilians and lepidosaurs it arises cranially/anteriorly to fte and dorsal to ife (Romer, 1923; Snyder, 1954; Russell & Bauer, 2008; Otero, Gallina & Herrera, 2010; Suzuki et al., 2011). Walker (1973) reports that in Testudines it arises from the posterodorsal rim of the lateral ilium and tendinously from its anterior border (Walker, 1973) dorsally or partially from the same level as the ife origin (Zug, 1971; Walker, 1973). This is supported partially by observations of Zug (1971) who observed this bifurcated origin for it as well. Yet, Zug (1971) reports that its origin is more variable than this though, i.e., some taxa were studied that only have either one or the other origin area.

M. iliotibialis inserts in concert with a, and f on the proximodorsal tibia (Fig. 3B) because a cneminal crest is lacking in plesiosaurs. The insertion of it is very uniform across Sauropsida. It contributes to a common tendon with m. femorotibialis (f) and m. ambiens (a) and attaches to the proximodorsal tibia (Otero, Gallina & Herrera, 2010; Suzuki et al., 2011) or to the cneminal crest (Snyder, 1954; Zug, 1971; Walker, 1973; Russell & Bauer, 2008).

Musculus femorotibialis (f)

-femorotibialis (Snyder, 1954; Zug, 1971; Walker, 1973; Suzuki et al., 2011)

-femorotibiales (Otero, Gallina & Herrera, 2010)

-femoro-tibialis (Romer, 1923; Gatesy, 1997)

An origin area on the dorsal femoral shaft that reaches around it anteriorly and posteriorly onto its ventral side in plesiosaurs is well supported by crocodilians, turtles, and lepidosaurs (Figs. 3A and 3B). The origin site is associated with the fan-shaped striations and rugosities on the distal dorsal plesiosaur femur. In turtles, crocodilians, and lepidosaurs, f originates from the femoral shaft dorsally, but reaches around it anteriorly and posteriorly onto the ventral side of the femur (Romer, 1923; Snyder, 1954; Zug, 1971; Walker, 1973; Russell & Bauer, 2008; Otero, Gallina & Herrera, 2010; Suzuki et al., 2011). In crocodilians two subportions are discerned (externus and internus) (Romer, 1923; Otero, Gallina & Herrera, 2010) and in turtles three (vastus internus, medialis, and externus) (Walker, 1973). In sauropsids, f origin area is situated on the femur distally to those insertions of the pelvic musculature that insert into the femur, except for m. adductor femoris (af), which may ventrally separate the overlapping origin area of f across the EPB (Romer, 1923; Snyder, 1954; Zug, 1971; Walker, 1973; Russell & Bauer, 2008; Otero, Gallina & Herrera, 2010; Suzuki et al., 2011).

F shares a common tendon of insertion with it and a (for more information see above it insertion) (Fig. 3B).

Musculus ambiens (a)

No synonyms reported in the literature on which this study is based (Romer, 1923; Snyder, 1954; Zug, 1971; Walker, 1973; Russell & Bauer, 2008; Otero, Gallina & Herrera, 2010; Suzuki et al., 2011) (Table 3).

The most common origin area for a in plesiosaurs is the origin area on the pubic tubercle anterior to the acetabulum as reported for all three extant taxa. This way the a origin site is bordered by m. puboischiotibialis externus (pe) as reported for all three sauropsid taxa. This arrangement is most closely to the turtle condition and is somewhat similar to and mirrors the bb arrangement of the pectoral limb. In plesiosaurs, an attachment on the area ventrally to the acetabulum where pubis and ischium meet is well supported by crocodilians and lepidosaurs (Figs. 3A and 3B). In a superficial, geometrical way, a originates from the pelvis relatively similar across crocodilians, turtles, and lepidosaurs namely anteroventrally to the acetabulum, but in detail they vary (Romer, 1923; Snyder, 1954; Zug, 1971; Walker, 1973; Russell & Bauer, 2008; Otero, Gallina & Herrera, 2010; Suzuki et al., 2011). From the puboischiadic ligament anteriorly and/or the lateral pubic process posteriorly originates the a in Testudines (Zug, 1971; Walker, 1973). Crocodilians have two tendinous a origin areas, with Caiman latirostris posing the exception having one head arising from a region anterior to the acetabulum (Otero, Gallina & Herrera, 2010). In the other crocodilians, two a origin tendons arise from the suture between pubis and ischium anteroventral to the acetabulum and from the pubic peduncle on the dorsal side ventrally to the acetabulum (Romer, 1923; Suzuki et al., 2011). In lepidosaurs, two a tendons arise laterally from the pelvic girdle just ventrally and anteriorly to the acetabulum, they soon converge and are followed by the muscle belly which soon merges towards the knee into a tendon which joins the patellar tendon. Ventrally it adds to the joint capsule (Russell & Bauer, 2008). In turtles, crocodilians, and lepidosaurs it is flanked by the pe origin area (Romer, 1923; Zug, 1971; Walker, 1973; Russell & Bauer, 2008; Suzuki et al., 2011), in the former also by the pi origin area (Zug, 1971; Walker, 1973).

A inserts together with it and f in sauropsids and plesiosaurs as described above (see it insertion for more details) (Fig. 3B).

Musculus iliofibularis (ifi)

-iliofibularis (Snyder, 1954; Zug, 1971; Walker, 1973; Russell & Bauer, 2008; Otero, Gallina & Herrera, 2010; Suzuki et al., 2011)

-ilio-fibularis (Romer, 1923; Gatesy, 1997)

The origin area of ifi is reconstructed on the posterior plesiosaur ilium ventrally to it and fte (Fig. 3A). In crocodilians, lepidosaurs, and turtles ifi originates from the posterolateral ilium (Romer, 1923; Snyder, 1954; Zug, 1971; Walker, 1973; Russell & Bauer, 2008; Otero, Gallina & Herrera, 2010; Suzuki et al., 2011). Its exact origin area is slightly variable across all three taxa, but it is situated generally posterior to ife origin and not on the dorsal border of the ilium and ventrally to it and fte origin (Romer, 1923; Snyder, 1954; Walker, 1973; Russell & Bauer, 2008; Otero, Gallina & Herrera, 2010; Suzuki et al., 2011), except for various turtles as reported by Zug (1971).

In plesiosaurs, the insertion area of ifi is on the proximal dorsal fibula and proximal to m. peroneus brevis and m. peroneus longus (pb and pl) origin site in plesiosaurs (Fig. 3B). In crocodilians, turtles, and lepidosaurs ifi inserts into the proximal third of the dorsal fibula proximal to the origins of pb and pl (Romer, 1923; Snyder, 1954; Zug, 1971; Walker, 1973; Russell & Bauer, 2008; Otero, Gallina & Herrera, 2010; Suzuki et al., 2011). The insertion area of ifi is displaced distally at approximately half the fibula length in turtles (Walker, 1973).

Musculus iliofemoralis (ife)

-iliofemoralis (Snyder, 1954; Zug, 1971; Walker, 1973; Russell & Bauer, 2008; Otero, Gallina & Herrera, 2010; Suzuki et al., 2011)

-ilio-femoralis (Romer, 1923; Gatesy, 1997)

The ife origin area is confidently placed on the lateral plesiosaur ilium (Fig. 3A), as supported by all three taxa, below the it origin area and above the acetabulum. Additionally it probably spread onto the adjacent vertebral column as described in turtles. Ife origin area lies on the lateral ilium dorsal to the acetabulum deep to it in crocodilians, turtles, and lepidosaurs (Romer, 1923; Otero, Gallina & Herrera, 2010; Suzuki et al., 2011). In turtles, it additionally arises from the last one to two dorsal vertebrae, from the first sacral vertebra, and from bordering areas on the carapace (Zug, 1971; Walker, 1973). In lepidosaurs, ife also arises from a ventral septal origin it shares with pit (Russell & Bauer, 2008).

A dorsal insertion of ife on the proximal plesiosaur femur is well supported for plesiosaurs by the EPB (Fig. 3B). Although a posterior insertion would be equally well supported, the former is preferred, as it matches well with part of the rugose and deeply striated muscle scar on the dorsal femoral trochanter. In Testudines, ife inserts dorsally into the trochanter major (Zug, 1971; Walker, 1973). In lepidosaurs, its insertion area is a comparatively large surface that covers much of the proximal and ventral femur and wraps around the posterior femur onto its posterodorsal side (Snyder, 1954; Russell & Bauer, 2008). In crocodilians, ife inserts posteriorly along the femoral shaft (Romer, 1923; Gatesy, 1997; Otero, Gallina & Herrera, 2010; Suzuki et al., 2011).

Musculus puboischiofemoralis internus (pi)

-puboischiofemoralis internus (Snyder, 1954; Zug, 1971; Walker, 1973; Russell & Bauer, 2008; Otero, Gallina & Herrera, 2010; Suzuki et al., 2011)

-pubo-ischio-femoralis internus (Romer, 1923; Gatesy, 1997)

Four origin areas for pi in plesiosaurs are discerned, of which two are better supported by the EPB and two yield a functional advantage for hindflipper elevation: 1. an origin on most of the dorsal plesiosaur pubis as in turtles and lepidosaurs, 2. an origin on the anterior ischium as in crocodilians and lepidosaurs (Fig. 3B), 3. an origin on the sacrum and it may have spread onto the first caudal or the last dorsal vertebrae, 4. an origin on the medial ilium and from the vertebral column as in crocodilians and turtles. In lepidosaurs, pi has three heads in lepidosaurs (pi 1–3) (Russell & Bauer, 2008) and two heads in crocodilians (pi I and pi 2) and turtles (Walker, 1973; Otero, Gallina & Herrera, 2010), although a single-headed state was described by Zug (1971) for Testudines, too. In crocodilians pi I has its origin area situated on the medial ilium and ischium posteriorly at their symphyseal region below the sacral rib facets on the ilium (Romer, 1923; Suzuki et al., 2011). Pi 2 arises ventrally from up to seven lumbar vertebrae and their transverse processes (Romer, 1923; Gatesy, 1997; Otero, Gallina & Herrera, 2010). The posterodorsal portion of turtles arises from the medial median ilium and ventrally from the first or second sacral and the first two caudal vertebra and ribs (Zug, 1971; Walker, 1973). The anteroventral head arises from the epipubic cartilage and the pubis dorsally (also from the thyroid fenestra). (Zug, 1971; Walker, 1973). In lepidosaurs Russell & Bauer (2008) describe three portions of this muscle (pi 1–3, from posterior to anterior). Pi 3 arises from most of the dorsal pubis extending posteriorly to the thyroid fenestra. Pi 1 arises from the symphysis of the ischia and posteriorly to the thyroid fenestra almost up to the ilium. Pi 2 arises in between pi 1 and 3 (Russell & Bauer, 2008).

A pi insertion on the anterodorsal proximal plesiosaur femur, proximal to f origin and distal to pe insertion, is well supported (Fig. 3B). Its attachment site was correlated with part of the large heavily striated and rugose muscle scar on the dorsal trochanter of the plesiosaur femur. In crocodilians, pi inserts in crocodilians and often in lepidosaurs into separate insertion areas on the femur anterodorsally to dorsally and posterodorsally (Romer, 1923; Snyder, 1954; Russell & Bauer, 2008; Otero, Gallina & Herrera, 2010; Suzuki et al., 2011). In turtles pi attaches to the dorsal to anterodorsal femur distally onto trochanter minor (Zug, 1971; Walker, 1973). In all three taxa it inserts proximally to f and distally to pe into the femur (Romer, 1923; Snyder, 1954; Zug, 1971; Walker, 1973; Russell & Bauer, 2008; Suzuki et al., 2011).

Ventral group

Musculus puboischiotibialis (pit)

-puboischiotibialis (Snyder, 1954; Zug, 1971; Walker, 1973; Russell & Bauer, 2008; Otero, Gallina & Herrera, 2010; Suzuki et al., 2011)

-pubo-ischio-tibialis (Romer, 1923; Gatesy, 1997)

For plesiosaurs the crocodilian state is reconstructed, as it is presumed, that the puboischiadic ligament is absent in plesiosaurs (s. above) (Fig. 3A). In crocodilians, the pit origin area is situated on the anterolateral ischium (Romer, 1923; Otero, Gallina & Herrera, 2010; Suzuki et al., 2011). Contrastingly, in turtles and lepidosaurs, pit arises from the puboischiadic ligament (Snyder, 1954; Walker, 1973; Russell & Bauer, 2008). In crocodilians and turtles, pit is a small muscle (Romer, 1923; Otero, Gallina & Herrera, 2010; Suzuki et al., 2011). Zug (1971) was not able to find it in turtle dissections. In lepidosaurs, pit is a large fan-shaped muscle (Snyder, 1954; Walker, 1973; Russell & Bauer, 2008).

The pit insertion is on the anterodorsal plesiosaur tibia, distal to the patellar tendon insertion (Fig. 3B) as in crocodilians and lepidosaurs. Pit inserts into the tibia anterodorsally distal to the patellar tendon formed by a, f, and m. iliotibialis (it) in lepidosaurs and crocodilians (Snyder, 1954; Russell & Bauer, 2008; Otero, Gallina & Herrera, 2010; Suzuki et al., 2011). In turtles it attaches anteroventrally to the tibia (Walker, 1973).

Musculus pubotibialis (pti)

No synonyms in the literature on which this muscle reconstruction is based on (Snyder, 1954; Zug, 1971; Walker, 1973; Russell & Bauer, 2008). Pti is not reported in crocodilians (Romer, 1923; Otero, Gallina & Herrera, 2010; Suzuki et al., 2011) (Table 3).

For plesiosaurs, the best supported hypothesis based on the EPB is that pti originated from the puboischiadic ligament, which is absent in plesiosaurs (s. above). So, instead pti must have spread onto the adjacent pubis (pubic tubercle) in plesiosaurs (Fig. 3A), similar to lepidosaurs. In Testudines and lepidosaurs, pti originates from the puboischiadic ligament (Zug, 1971; Walker, 1973; Russell & Bauer, 2008) anterior and superficial to pit (Walker, 1973; Russell & Bauer, 2008). A second pti head is known in lepidosaurs to arise ventrally from the pubis anteroventrally to a from the processus lateralis of the pubis (Snyder, 1954; Russell & Bauer, 2008).

An insertion on the proximoventral tibia, proximally to the insertion area of fte and m. flexor tibialis internus (fti), is well supported (Fig. 3A) in plesiosaurs. Pti attaches ventrally and proximally to the tibia, proximally to the attachment sites of fte and fti in turtles and lepidosaurs (Snyder, 1954; Zug, 1971; Walker, 1973; Russell & Bauer, 2008).

Musculus flexor tibialis internus (fti)

No synonyms in the literature on which this study is based on (Romer, 1923; Snyder, 1954; Zug, 1971; Walker, 1973; Gatesy, 1997; Otero, Gallina & Herrera, 2010; Suzuki et al., 2011; Russell & Bauer, 2008) (Table 3).

In plesiosaurs a ventral origin area of fti on the posteroventral ischium is well supported by all three taxa (Fig. 3A). A second fti head from a dorsal origin area was reconstructed for plesiosaurs on the first two to six caudals and possibly also on the sacrum based on turtles and Sphenodon. In turtles, one or two origins for fti are known (Zug, 1971; Walker, 1973). Two or three fti heads are described in lepidosaurs (Snyder, 1954; Russell & Bauer, 2008). For crocodilians three to four heads are observed (Romer, 1923; Gatesy, 1997; Otero, Gallina & Herrera, 2010; Suzuki et al., 2011). All three taxa share a ventrally situated fti origin on the posterior ischium (fti1 and fti3 in crocodilians) (Romer, 1923; Zug, 1971; Walker, 1973; Gatesy, 1997; Russell & Bauer, 2008; Otero, Gallina & Herrera, 2010; Suzuki et al., 2011). In turtles this head also originates from the posterior puboischiadic ligament (Zug, 1971; Walker, 1973) and in lepidosaurs also from the ilioischiadic ligament and the perineal region (Snyder, 1954; Russell & Bauer, 2008). Crocodilians, lepidosaurs, and turtles have a dorsally arising component of fti (Romer, 1923; Zug, 1971; Walker, 1973; Gatesy, 1997; Russell & Bauer, 2008; Otero, Gallina & Herrera, 2010; Suzuki et al., 2011). In crocodilians, an iliac origin of fti (heads 2 and 4) is present (Romer, 1923; Gatesy, 1997; Otero, Gallina & Herrera, 2010; Suzuki et al., 2011). In lepidosaurs this head originates from an intermuscular septum it shares with fte and from the ilioischiadic ligament (Russell & Bauer, 2008). In turtles, there may be an iliac origin and/or on the vertebral column, i.e., from one sacral and one to two or three caudal vertebrae (Zug, 1971; Walker, 1973). As all three states provided by the EPB are equally likely, Sphenodon was considered in addition. In Sphenodon, the origin of fti has spread onto the vertebral column (from the first six caudal vertebrae’s transverse processes (reviewed in Russell & Bauer, 2008)) as in turtles.

In plesiosaurs, the fti insertion is placed in common with the pit insertion on the proximal anterior tibia (Figs. 3A and 3B) as in crocodilians and lepidosaurs. An additional tendon to the gastrocnemius is equally well supported by these two taxa. In turtles, two muscle bellies converge into a common tendon which inserts proximoventrally to anteriorly into the tibia, distally to pit and pti insertion (Zug, 1971; Walker, 1973). In lepidosaurs and crocodilians the insertions of this muscle are highly complex, a complexity which cannot be reconstructed in detail for extinct plesiosaurs. In a simplified way, fti portions insert tendinously partially together with pit and partially by themselves into the proximal tibia (Gatesy, 1997; Russell & Bauer, 2008; Otero, Gallina & Herrera, 2010; Suzuki et al., 2011) posterodorsally in lepidosaurs (Russell & Bauer, 2008) and anterodorsally in crocodilians in common with fte (Gatesy, 1997; Otero, Gallina & Herrera, 2010; Suzuki et al., 2011). An additional tendon inserts into the gastrocnemius in lepidosaurs and crocodilians (Gatesy, 1997; Russell & Bauer, 2008; Suzuki et al., 2011).

Musculus flexor tibialis externus (fte)

-flexor tibialis externus

No synonyms in the literature on which this muscle reconstruction is based on (Romer, 1923; Snyder, 1954; Zug, 1971; Walker, 1973; Gatesy, 1997; Russell & Bauer, 2008; Otero, Gallina & Herrera, 2010; Suzuki et al., 2011) (Table 3).

The origin site of fte is reconstructed on the plesiosaur vertebral column as reported from turtles and Sphenodon and on the lateral ilium as in crocodilians and turtles (Fig. 3A).

Fte is single-headed in lepidosaurs and crocodilians (Romer, 1923; Snyder, 1954; Gatesy, 1997; Russell & Bauer, 2008; Otero, Gallina & Herrera, 2010; Suzuki et al., 2011) and may be double-headed in turtles (Zug, 1971; Walker, 1973). Laterally on the dorsal ilium border (Romer, 1923; Gatesy, 1997; Otero, Gallina & Herrera, 2010; Suzuki et al., 2011), posterior to it and ifi, the fte origin area is situated in crocodilians (Romer, 1923; Gatesy, 1997; Suzuki et al., 2011). In lepidosaurs, it arises from the ilioischiadic ligament, partially closely associated with the posterior portion of fti (Russell & Bauer, 2008). In Sphenodon, fte has spread onto the caudal vertebral column (Russell & Bauer, 2008). Contrastingly, in Testudines fte may have one or two heads which take their origin on the lateral posterodorsal ilium (Zug, 1971; Walker, 1973) or from the adjacent second sacral rib or from the first to the fifth caudal vertebrae (Zug, 1971) and posterodorsally from the ischium (Zug, 1971; Walker, 1973) or puboischiadic ligament (Zug, 1971).

An insertion of fte on the proximal tibia as in crocodilians, turtles, and lepidosaurs and posteroventrally as in crocodilians and turtles is reconstructed for plesiosaurs (Fig. 3A). In plesiosaurs, an additional tendon of fte inserts into the m. gastrocnemius internus (gi) as in lepidosaurs, turtles, and crocodilians. In lepidosaurs, turtles, and crocodilians, fte attaches by a bifurcated tendon to the tibia, and via a common tendon with fti, it converges with gi (Romer, 1923; Walker, 1973; Russell & Bauer, 2008; Otero, Gallina & Herrera, 2010; Suzuki et al., 2011). In turtles, a single tendon attachment on the posteroproximal tibia is possible (Zug, 1971). In crocodilians and turtles, fte inserts into the proximoventral tibia (Romer, 1923; Zug, 1971; Walker, 1973; Otero, Gallina & Herrera, 2010; Suzuki et al., 2011) and in lepidosaurs into the posteroventral tibia proximally (Snyder, 1954; Russell & Bauer, 2008).

Musculus caudifemoralis brevis (cfb)

-caudofemoralis brevis (Gatesy, 1997; Otero, Gallina & Herrera, 2010)

-caudifemoralis brevis (Snyder, 1954; Russell & Bauer, 2008; Suzuki et al., 2011)

-caudi-iliofemoralis (Zug, 1971; Walker, 1973)

-coccygeo-femoralis brevis (Romer, 1923)

A cfb origin on the first caudals is supported for plesiosaurs by all three taxa and one from the posterior ilium (Figs. 3A and 3B) and the sacrum by turtles and crocodilians. Thus, the first option is reconstructed because it is best supported by the EPB. The second option displays the potential in sauropsids to enlarge the surface of origin for this muscle, despite of the presence of a diminutive ilium in plesiosaurs. In crocodilians, cfb originates from the ventral posterolateral ilium (Romer, 1923) and either from the first caudal vertebra (Gatesy, 1997; Otero, Gallina & Herrera, 2010) or also from the last sacral vertebra (Gatesy, 1997). In turtles, the origin of cfb seems to be more variable. It involves the posteromedial ilium, the sacral vertebrae, and may spread onto up to two dorsal vertebrae and onto up to four caudal vertebrae (Zug, 1971; Walker, 1973). In lepidosaurs, cfb originates from the first postsacrals and does not involve the ilium as origin surface (Snyder, 1954; Russell & Bauer, 2008).

The insertion area of cfb is reconstructed on a much rugose scar, posteriorly on the plesiosaur femur, supported by all three EPB taxa. The attachment site is adjacently to the pe insertion as in turtles (Fig. 3B). Across all three taxa, cfb attaches posteroventrally to the femur (Romer, 1923; Zug, 1971; Walker, 1973; Gatesy, 1997; Russell & Bauer, 2008; Otero, Gallina & Herrera, 2010; Suzuki et al., 2011). In turtles, cfb attaches to trochanter major (Zug, 1971; Walker, 1973), in lepidosaurs to the femoral trochanter (Snyder, 1954; Russell & Bauer, 2008), and in crocodilians to the fourth trochanter (Romer, 1923; Gatesy, 1997; Otero, Gallina & Herrera, 2010; Suzuki et al., 2011). In lepidosaurs and crocodilians, cfb attaches distally to the pe insertion (Romer, 1923; Snyder, 1954; Russell & Bauer, 2008; Suzuki et al., 2011) and in turtles it attaches adjacently to it (Walker, 1973).

Musculus caudifemoralis longus (cfl)

-caudofemoralis longus (Gatesy, 1997; Otero, Gallina & Herrera, 2010)

-coccygeo-femoralis longus (Romer, 1923)

-caudifemoralis longus (Snyder, 1954; Russell & Bauer, 2008; Suzuki et al., 2011)

This muscle is reduced in turtles (Walker, 1973) (Table 3).

In plesiosaurs, cfl originates from the centra, lateral haemal arches, and ventral transverse processes as in crocodilians and lepidosaurs for at least up to 13 caudal vertebrae or even further caudally. In crocodilians and lepidosaurs, cfl is a large muscle mass that arises caudally to and partially along with cfb from the centra, ventral transverse processes, and the lateral haemal arches (Gatesy, 1997; Russell & Bauer, 2008; Otero, Gallina & Herrera, 2010) from 14 caudal vertebrae in lepidosaurs (Russell & Bauer, 2008) and from up to 13 (Otero, Gallina & Herrera, 2010) or 15 (Romer, 1923; Gatesy, 1997) caudal vertebrae in crocodilians.

An insertion in common with the cfb on the proximal femur is likely in plesiosaurs (Fig. 3B). The insertion of the tendon into either the ventral knee joint as in lepidosaurs, or in the complex crocodilian way are both equally likely for plesiosaurs. Cfl inserts via a long tendon together with cfb in the proximal femur in crocodilians and lepidosaurs (Russell & Bauer, 2008; Otero, Gallina & Herrera, 2010). Part of the tendon runs further distally to insert ventrally into the knee joint in lepidosaurs (Russell & Bauer, 2008). In crocodilians the insertion of cfl is complex: it splits up and one part attaches to the ventral and proximal fibula, another part converges with a tendon of ifi and with the tendon of m. gastrocnemius externus (ge) (Otero, Gallina & Herrera, 2010).

Musculus ischiotrochantericus (i)

-ischiotrochantericus (Snyder, 1954; Zug, 1971; Walker, 1973; Russell & Bauer, 2008; Otero, Gallina & Herrera, 2010; Suzuki et al., 2011)

-ichio-trochantericus (Romer, 1923; Gatesy, 1997)

In plesiosaurs, i arises from the medial ischium as in all three taxa and from approximately the posterior half of the ischium (Fig. 3B), posterior to pi as lepidosaurs and crocodilians. In turtles, lepidosaurs, and crocodilians, the i origin area is situated on the dorsal ischium (Romer, 1923; Snyder, 1954; Zug, 1971; Walker, 1973; Russell & Bauer, 2008; Otero, Gallina & Herrera, 2010; Suzuki et al., 2011). In the former, it occupies the first ~ two-thirds of the ischium, the ventralmost region of the medial ilium and the membrane covering the thyroid fenestra (Walker, 1973) and in the latter two the posterior ~ third to half of the ischium (Romer, 1923; Snyder, 1954; Russell & Bauer, 2008; Otero, Gallina & Herrera, 2010; Suzuki et al., 2011). In lepidosaurs, crocodilians, and turtles, the origin area of i lies posterior to pi (ischial head) (Romer, 1923; Snyder, 1954; Walker, 1973; Russell & Bauer, 2008; Otero, Gallina & Herrera, 2010; Suzuki et al., 2011). In turtles i arises anterior to fte (Zug, 1971; Walker, 1973).

I insertion area is reconstructed on the proximal posteroventral plesiosaur femur as in lepidosaurs and in turtles, and similar to crocodilians. The muscle insertion is associated with part of the posteroventral rugose muscle scar on the plesiosaur femur, which displaces it further distally as reported for extant sauropsids (Fig. 3A). I inserts proximally posteroventrally into the lepidosaur femur (Snyder, 1954; Russell & Bauer, 2008), posteroventrally or ventrally into the intertrochanteric fossa in turtles (Zug, 1971; Walker, 1973) and posteriorly to posteroventrally in crocodilians (Romer, 1923; Otero, Gallina & Herrera, 2010; Suzuki et al., 2011). I inserts into the femur in all three taxa as one of the most proximal pelvic muscles (Romer, 1923; Snyder, 1954; Zug, 1971; Walker, 1973; Russell & Bauer, 2008; Otero, Gallina & Herrera, 2010; Suzuki et al., 2011).

Musculus adductor femoris (af)

-adductor femoris (Zug, 1971; Walker, 1973)

-adductor femoris 1 and adductor femoris 2 (Romer, 1923; Otero, Gallina & Herrera, 2010; Suzuki et al., 2011)

The best supported origin area for af in plesiosaurs is, because the puboischiadic ligament is absent, on the posterolateral ischium (Fig. 3A). In crocodilians, af is two headed. One belly originates from the anterolateral and the other one from the posterolateral ischium. In between stretches the origin area of the pe3 (please view section on puboischiofemoralis externus (pe) below for more information). The former reaches up to the pit origin towards the acetabulum and the latter is bordered towards the body midline by fti1 (Romer, 1923; Otero, Gallina & Herrera, 2010; Suzuki et al., 2011). In turtles and lepidosaurs, af takes its origin on the puboischiadic ligament (Snyder, 1954; Zug, 1971; Walker, 1973; Russell & Bauer, 2008). In lepidosaurs, af arises superficial to pit and posterior to pti (Russell & Bauer, 2008). In turtles, its attachment also spreads onto adjacent areas of the lateral process of the ischium.

In plesiosaurs, af is confidently reconstructed onto the ventral femoral shaft in between the overlapping origins of f and distal to the pe insertion on the femur, as reported for the three extant EPB taxa (Fig. 3A). It can be correlated with parts of the striated plesiosaur femur shaft surface. Af inserts anteroventrally into the femoral shaft in lepidosaurs and turtles (Snyder, 1954; Zug, 1971; Walker, 1973; Russell & Bauer, 2008) and posteroventrally on the distal half in crocodilians (Romer, 1923; Otero, Gallina & Herrera, 2010; Suzuki et al., 2011).

Musculus puboischiofemoralis externus (pe)

-puboischiofemoralis externus (Snyder, 1954; Zug, 1971; Walker, 1973; Russell & Bauer, 2008; Otero, Gallina & Herrera, 2010; Suzuki et al., 2011)

-pubo-ischio-femoralis externus (Romer, 1923; Gatesy, 1997)

In plesiosaurs, pe origin area is on the ventral pubis and ischium as described in all three extant EPB taxa (Fig. 3A). Probably, pe also originates from the membrane covering the thyroid fenestra. Pe arises in crocodilians, lepidosaurs, and turtles from the ventral pubis and ischium (Romer, 1923; Zug, 1971; Walker, 1973; Gatesy, 1997; Russell & Bauer, 2008; Otero, Gallina & Herrera, 2010; Suzuki et al., 2011). In crocodilians, it also arises from the anterodorsal pubis (Romer, 1923; Gatesy, 1997; Otero, Gallina & Herrera, 2010; Suzuki et al., 2011). In turtles an anterior (pubis) and posterior portion (from thyroid fenestra and ischium) are reported (Zug, 1971; Walker, 1973). Pe1, 2, and 3 are described for crocodilians: 1 from anterodorsal pubis, 2 from anteroventral pubis, and 3 from lateral ischium bordered anteriorly by af 1 origin and posteriorly by af 2 and fti1 (Romer, 1923; Gatesy, 1997; Otero, Gallina & Herrera, 2010; Suzuki et al., 2011). In lepidosaurs there are two portions of pe described as well as in turtles, but they are subdivided into a1, a2, a3, and b. a1 and a2 originate from most of the pubis, a3 from posterior thyroid fenestra and ischium and b from posterior and medial ischium.

In plesiosaurs, pe inserts into the anteroventral proximal plesiosaur femur as observed in turtles, crocodilians, and lepidosaurs. It is associated with a large muscle scar on the proximoventral plesiosaur femur (Fig. 3B). In turtles and lepidosaurs, pe inserts anteriorly dorsally and ventrally (Snyder, 1954; Zug, 1971; Walker, 1973; Russell & Bauer, 2008) into the trochanter minor of the femur in the former (Zug, 1971; Walker, 1973) and the femoral trochanter in the latter. In lepidosaurs, the subportions are divisible at their insertions as well (Snyder, 1954; Russell & Bauer, 2008). In crocodilians, pe portions insert in common into the trochanter major anteroventrally (Romer, 1923; Gatesy, 1997; Otero, Gallina & Herrera, 2010; Suzuki et al., 2011).

Muscles of the crus

Dorsal group

Musculus extensor digitorum longus (edl)

-extensor digitorum communis + extensor hallucis longus (Walker, 1973)

-extensor digitorum communis (Zug, 1971)

-extensor digitorum longus (Snyder, 1954; Russell & Bauer, 2008; Suzuki et al., 2011)

The origin of edl is on the fibular epicondyle of the plesiosaur femur (Fig. 3B) like in all three EPB taxa in which edl arises from the fibular epicondyle dorsally, just proximal to the joint capsule (Snyder, 1954; Zug, 1971; Walker, 1973; Russell & Bauer, 2008; Suzuki et al., 2011).

The best supported edl insertion by all three EPB taxa would be posterodorsally on metatarsal II and III. However, due to the different arrangement of the plesiosaur pes and probable loss of individual toe movement, their insertions are placed on the dorsal metatarsals. Additionally, we reconstructed this muscle onto metacarpals I and IV as in turtles because we presume the turtles show the more plesiomorphic condition for Sauropsida than crocodylians and lepidosaurs (Fig. 3B). In lepidosaurs, edl attaches to metacarpal II and III (Snyder, 1954; Russell & Bauer, 2008), in crocodilians additionally to metacarpal IV (Suzuki et al., 2011), and in turtles additionally to that to metacarpal I (Walker, 1973). In turtles and lepidosaurs, edl inserts into the posterodorsal shaft of the metacarpals (Snyder, 1954; Walker, 1973; Russell & Bauer, 2008), while in crocodilians it attaches dorsally to them (Suzuki et al., 2011). In Testudines, there is also an anterodorsal insertion on metacarpal IV. In turtles, edl may become fascial, especially the slips to metatarsal I–III (Walker, 1973).

Musculus peroneus brevis and Musculus peroneus longus (pb and pl)

-peroneus anterior and peroneus posterior (Zug, 1971; Walker, 1973)

-peroneus brevis and peroneus longus (Snyder, 1954; Russell & Bauer, 2008; Suzuki et al., 2011)

In all three extant EPB taxa, pb and pl arise distally to the insertion ifi on the fibula in plesiosaurs. The EPB (crocodilians and turtles) suggest a pl origin on the posterodorsal distal half of the fibula and a pb origin on the posterodorsal distal fibula in plesiosaurs, distally to the attachment of ifi on the fibula (Fig. 3B). Pl arises from the distal dorsal and posterior half of the fibula in turtles (Walker, 1973; Suzuki et al., 2011). In crocodilians, the pl origin spreads further onto the ventral side than in turtles and reaches further proximal, too (Suzuki et al., 2011). In lepidosaurs, its origin area is situated on the fibular epicondyle of the femur (Russell & Bauer, 2008). Pb originates from a small area on the posterodorsal distal fibula in turtles and crocodilians (Walker, 1973; Suzuki et al., 2011). In lepidosaurs, the pb origin covers most of the femoral shaft except for a thin area extending proximodistally (Russell & Bauer, 2008).

In plesiosaurs, pb and pl insert into the dorsal metatarsal V. Pb attaches to the proximoposterior tubercle and pl on the distoposterior tubercle of metatarsal V (Fig. 3B). In crocodilians and turtles, pl and pb insert into dorsal metatarsal V (Walker, 1973; Suzuki et al., 2011). Pl attaches proximoposteriorly in crocodilians (Suzuki et al., 2011) and distoposteriorly in turtles and lepidosaurs (Walker, 1973; Russell & Bauer, 2008). Pb inserts posterodistally in crocodilians (Suzuki et al., 2011) and proximoposteriorly in turtles (Walker, 1973; Russell & Bauer, 2008).

Musculus tibialis anterior (ta)

No synonyms in the studies on which theses muscle reconstructions are based on (Snyder, 1954; Zug, 1971; Walker, 1973; Russell & Bauer, 2008; Suzuki et al., 2011) (Table 3).

In plesiosaurs, ta arises from the dorsal to anterodorsal tibia, distal to the patellar tendon insertion (Fig. 3B). Ta originates from the tibia across the groups used fort the EPB. In turtles, the ta origin area covers the distal two-thirds of the anterior tibia (Walker, 1973). In crocodilians, this muscle arises from the dorsal tibia relatively proximally (Suzuki et al., 2011). In lepidosaurs, ta originates from most of the dorsal tibial shaft (Russell & Bauer, 2008). In all three taxa, the origin of ta is distal to the insertion of the patellar tendon (Walker, 1973; Russell & Bauer, 2008; Suzuki et al., 2011).

In plesiosaurs, ta inserts into the proximal anterior metatarsal I as in lepidosaurs and turtles (Fig. 3B). In turtles and lepidosaurs, the insertion into metatarsal I is proximally and anteriorly and it spreads onto the ventral and dorsal side (Walker, 1973; Russell & Bauer, 2008). In crocodilians, ta inserts into proximal dorsal metatarsal I and II (Suzuki et al., 2011).

Ventral group

Musculus gastrocnemius internus and Musculus gastrocnemius internus (gi and ge)

-gastrocnemius internal/tibial and femoral/external head (Walker, 1973)

-gastrocnemius and anterior and posterior head (Zug, 1971)

-gastrocnemius extra lateral, lateral, and medial head (Suzuki et al., 2011)

-femorotibial and femoral gastrocnemius (Russell & Bauer, 2008)

-fibular and tibial gastrocnemius (Snyder, 1954)

-gastrocnemius externus and gastrocnemius internus (Otero, Gallina & Herrera, 2010).

Internal/tibial head of gastrocnemius in Testudines (Walker, 1973) equals the medial head in crocodilians (Suzuki et al., 2011) and the femorotibial/tibial one of lepidosaurs (Snyder, 1954; Russell & Bauer, 2008). External/femoral gastrocnemius head of gastrocnemius (Walker, 1973) is the same as the lateral and extralateral portion of gastrocnemius in crocodilians (Suzuki et al., 2011) and these are the same as the femoral/fibular head in lepidosaurs (Snyder, 1954; Russell & Bauer, 2008). Further the lateral portion after Suzuki et al. (2011) equals gastrocnemius externus (Otero, Gallina & Herrera, 2010) and the medial portion (Suzuki et al., 2011) is homologous to gastrocnemius internus (Otero, Gallina & Herrera, 2010). It was decided to go along with the terminology of turtles and Otero, Gallina & Herrera (2010) for crocodilians because a detachment from the various origins of gastrocnemius and to focus on its general position in the hindlimb helps to identify the muscular head more clearly (Table 3).

The origin area of ge is on the fibular epicondyle of the plesiosaur femur as was found across the groups used for EPB. There are two options to reconstruct gi in plesiosaurs: Either from the tibial epicondyle of the femur and the tibia, associated with the joint capsule, as is better supported by lepidosaurs and some turtles, or exclusively from the tibia, as found in crocodilians and some turtles (Fig. 3A). In turtles, crocodilians, and lepidosaurs, gastrocnemius comprises two large heads (Walker, 1973; Russell & Bauer, 2008; Otero, Gallina & Herrera, 2010; Suzuki et al., 2011). Ge arises distally from the fibular epicondyle of the femur (Walker, 1973; Russell & Bauer, 2008; Suzuki et al., 2011) posteriorly in crocodilians and turtles (Walker, 1973; Suzuki et al., 2011) and ventrally in lepidosaurs (Russell & Bauer, 2008). In lepidosaurs, it is also closely associated with the ventral knee joint capsule and there it interconnects with ifi. In crocodilians, an extralateral subportion of this muscle is present (Suzuki et al., 2011). The head of gi arises from the tibia in crocodilians, lepidosaurs, and turtles (Walker, 1973; Russell & Bauer, 2008; Suzuki et al., 2011). In turtles, gi arises by two subportions from the anterodorsal and anteroventral and the ventral tibia (Walker, 1973). In crocodilians, gi originates from the anterior and proximal tibia (Otero, Gallina & Herrera, 2010; Suzuki et al., 2011). In lepidosaurs, the gi origin area is on the anterior and anteroventral tibia and it is associated with the ventral knee joint capsule, the meniscus, and partially arises from the tibial femoral epicondyle (Russell & Bauer, 2008). For some turtles, a spreading of the origin area onto the distal femur is reported, too (Walker, 1973).

The insertion of the different bellies of gastrocnemius is very complex across sauropsids and therefore difficult to reconstruct for plesiosaurs. The gastrocnemial heads all have in common that at approximately ankle level, the separated muscle bellies become associated with each other and form tendinous structures (aponeuroses, tendons). These tendinous structures attach to metatarsal V posteroventrally and either to the astragalocalcaneum (in lepidosaurs) or to metatarsal I (in turtles). Further, gastrocnemius is closely associated with mm. flexores digitores breves (fdb) and inserts partially alone, partially in common with it into digit I to V in lepidosaurs and to digits I–IV in crocodilians and turtles (Fig. 3A). In turtles, crocodilians, and lepidosaurs, the gastrocnemial heads converge and become aponeurotic at approximately ankle level (Walker, 1973; Russell & Bauer, 2008; Suzuki et al., 2011) and form the plantar aponeurosis. The plantar aponeurosis is associated with the flexor retinaculum which inserts into the tubercle of metatarsal V posteroventrally and into the tubercle of metatarsal I anteroventrally (Walker, 1973; Suzuki et al., 2011). Further, it inserts into digits I–IV in common with fdb (Walker, 1973; Suzuki et al., 2011). Russell & Bauer (2008) show how complex the insertion of the gastrocnemii is in lepidosaurs. This amount of detail is impossible to reconstruct for a fossil animal, hence this muscle insertion will be treated in relatively superficially. Gastrocnemius has three, partially interconnected, partially independently acting layered subportions. The muscle inserts into metatarsal V and anteriorly onto the astragalocalcaneum. Additionally, it either sends out own motor tendons to all five digits or is associated with fdb. The gastrocnemius layers are associated amongst each other tendinously, with pl and pb, and with fte (Russell & Bauer, 2008).

Musculus flexor digitorum longus (fdlh)

-flexor digitorum longus (Russell & Bauer, 2008)

-flexor digitorum longus and musculi lumbricales (Snyder, 1954; Zug, 1971; Walker, 1973)

-flexor digitorum longus + flexor hallucis longus serving digit I + flexor digiti II–IV (Suzuki et al., 2011)

Although other names experience more acceptance across taxa and literature on which this study is based on, fdlh was given priority just to pay tribute to simplification of the terminology (Table 3).

An fdlh origin area on the posteroventral fibular epicondyle of the plesiosaur femur is equally supported by EPB to be proximal or distal to the ge origin. A ventral fibular origin of fdlh is also well supported by all three groups of the EPB. A tibial origin is supported by crocodilians and some turtle taxa. The lumbricals originate in turtles, lepidosaurs, and crocodiles from the flexor plate of fdlh, so this might as well have been the case in plesiosaurs (Fig. 3A). In turtles, fdlh originates from the posteroventral fibular epicondyle proximal to the ge origin (Zug, 1971; Walker, 1973), while in crocodilians adjacently to it but more ventrally (Suzuki et al., 2011), and in lepidosaurs distal to it (Snyder, 1954; Russell & Bauer, 2008). Fdlh also arises from the fibula. In turtles this origin area is situated along the ventral shaft (Walker, 1973). In lepidosaurs fdlh arises relatively proximal from the anterior fibula (Russell & Bauer, 2008) and in crocodilians from the ventral proximal tibia and fibula (Suzuki et al., 2011). A tibial origin is also reported by Zug (1971) (for Trionyx). In lepidosaurs, the fibular origin gives rise to two muscle bellies. Lepidosaurs have additional contributions to fdlh from the astragalocalcaneum anteroventrally, the metatarsocalcaneal posteroventrally, from metatarsal V, from the distal calcaneum, and fdlf heads that arise aponeurotically (Russell & Bauer, 2008). In crocodilians, there are heads from metatarsals II–IV (Suzuki et al., 2011). In all Sauropsida, fdlh forms a flexor plate (Snyder, 1954; Zug, 1971; Walker, 1973; Russell & Bauer, 2008; Suzuki et al., 2011). From the flexor plate arise muscles, often termed lumbricals, in crocodilians, turtles, and lepidosaurs (Walker, 1973; Russell & Bauer, 2008; Suzuki et al., 2011).

An insertion comparable to the “lumbrical” insertion on digit III and IV is best supported for plesiosaurs by all three taxa, but one on digit II (crocodilians) or digit V (turtles) is possible (Fig. 3A). It seems likely that the lumbricals are either highly aponeurotic in plesiosaurs, that they do not differentiate much, or that they are relatively reduced in size. Fdlh inserts into the terminal phalanges of digit I–IV in most turtle taxa (Zug, 1971; Walker, 1973), but in some, e.g., Cheloniidae, also onto the terminal phalanx of digit V (Walker, 1973). The insertion patterns of fdlf are highly complex, but generalized one can say, all of the heads of fdlh in lepidosaurs contribute to tendons that insert into the terminal phalanges of digits I–V according to Snyder (1954) and to the terminal phalanges of digits I–IV in Iguana and crocodilians (Russell & Bauer, 2008; Suzuki et al., 2011). The “lumbricals” that arise from the flexor plate insert into the terminal phalanges of digit II, III, and IV in turtles (Walker, 1973), to III and IV in lepidosaurs (Russell & Bauer, 2008), and to digit I–IV in crocodilians but in digit I at midlength of the shaft of phalanx I posteroventrally, in digit II and III at proximoposterior phalanx I, and anteriorly on midshaft length on the proximal phalanx in digit IV (Suzuki et al., 2011).

Musculus pronator profundus (pp)

No synonyms in the literature on which this study is based on (Snyder, 1954; Zug, 1971; Walker, 1973; Russell & Bauer, 2008; Suzuki et al., 2011) (Table 3).

Best supported by EPB, is an origin area for pp on the anterior plesiosaur fibula, anterior to the fibular origin site of fdlh. Due to the closely associated tibia and fibula and to their disc-shaped form and the loss of a clearly demarcated long bone shaft in Cryptoclidus eurymerus (IGPB R 324), this seems highly unlikely. It can be reconstructed on the ventral fibula (Fig. 3A) as in turtles or additionally on the ventral tibia as in crocodilians. Pp originates from the fibula in turtles and lepidosaurs (Walker, 1973; Russell & Bauer, 2008). In lepidosaurs, its origin area is situated on the ventral and anterior fibula extending across most of the distal shaft. In Testudines it extends proximodistally along the ventral fibula (Walker, 1973; Russell & Bauer, 2008). In crocodilians, it originates from the posterior tibia and the anterior fibula (Suzuki et al., 2011). In all three taxa it arises anteriorly to the origin of the fibular head of fdlh (Walker, 1973; Russell & Bauer, 2008; Suzuki et al., 2011).

A pp insertion in plesiosaurs is best supported (by all three taxa) on the ventral proximal metatarsal I. An attachment on metatarsal II is also likely (by crocodilians and lepidosaurs). An attachment on metatarsal III is less supported (Fig. 3A). In turtles, pp inserts into the proximoposterior metatarsal I, the posterodistal distal tarsal I, and the anterodistal distal tarsal II. In lepidosaurs, this muscle attaches ventrally to the posterior metatarsal I, II, III and proximoventrally on the distal tarsal IV (Russell & Bauer, 2008). In crocodilians, pp attaches to metatarsal I posteriorly and metatarsal II anteriorly (Suzuki et al., 2011).

Muscles of the pes

Dorsal group

Musculi extensores digitores breves (edb)

-extensores digitores breves (Russell & Bauer, 2008)

-extensor digitorum II, III, IV and ext. hallucis brevis (Suzuki et al., 2011)

-extensores digitorum breves and abductor hallucis (Walker, 1973)

-extensores digitores breves and interossei dorsales (Zug, 1971)

-extensor digitorum brevis (Snyder, 1954)

Extensores digitores breves by Russell & Bauer (2008) was chosen as it seems to be generally accepted across taxa and as it simplifies terminology in comparison to other options (Table 3).

An origin of edb in plesiosaurs on the metatarsals is well supported by crocodilians and lepidosaurs and partially by turtles. An origin on the astragalocalcaneum in lepidosaurs and on the calcaneum in crocodilians suggests a posterior tarsal origin of edb in plesiosaurs. In crocodilians and lepidosaurs, the proximalmost origin of edb involves the tibia or the fibula which are both equally likely to be reconstructed for Cryptoclidus eurymerus (IGPB R 324) (Fig. 3B). In turtles, edb originate dorsally from the anteroproximal metatarsal I and the bordering anterodistal distal tarsal, and distal tarsal III and IV (Walker, 1973). In crocodilians, edb originate from the shafts of metatarsal I–IV dorsally. Additionally, edb I–III arise from the distal dorsal tibia and edb IV from the calcaneum posterodorsally (Suzuki et al., 2011). In lepidosaurs, the origins of edb are complex as they often arise from various associated ligaments of the crus. It is impossible to reconstruct the ligamentous origins for plesiosaurs, therefore these will be left out in the following description. Generally, the muscle bellies of edb arise adjacently to the digits from the tarsus. The heads serving digit I and II originate from proximal dorsal metatarsal I and III. The latter has a second origin area on distal dorsal metatarsal II. The edb to digit III, IV, and V have an astragalocalcaneal origin. The edb portion of digit III has two additional muscle bellies which originate from distal metatarsal II anteriorly and distal dorsal metatarsal IV. Digit IV head of edb also arises from metatarsal IV and from the tibiofibular ligament. Digit V comprises three muscle heads from the astragalocalcaneum, metatarsal V, and from a fascia overlying part of the astragalocalcaneum (Russell & Bauer, 2008).

An insertion on the ungual phalanges of digit I–IV is confidently reconstructed for plesiosaurs as it is supported by crocodilians, turtles, and lepidosaurs. An insertion on the ungual phalanx of digit V is based on lepidosaur myology (Fig. 3B). The loss of digit V in crocodilians does not support this nor contradicts it. In turtles, the insertion of the muscle slip on digit V may be misidentified or missed out or the slip is reduced, and another muscle emulates its function. In all three extant EPB groups, edb inserts into the ungual phalanx of digit I–IV (Walker, 1973; Russell & Bauer, 2008; Suzuki et al., 2011). In turtles, it is reported that edb attaches to the connective tissue of the joint capsules of the penultimate and terminal phalanx, i.e., in turtles edb also attaches to the distal dorsal penultimate phalanx of digit I–IV. Additionally, there is a muscle inserting into the terminal phalanx of digit V in Figure 30 (Walker, 1973) but it is designated as being part of the peroneus anterior, so it is possible that this could be synonymous with edb V tendon (Walker, 1973). Contrastingly, crocodilians only have four toes, the fifth is reduced (Suzuki et al., 2011). Accessory edb tendons that attach to other phalanges than the ungual ones are found in lepidosaurs in digit II–V and in crocodilians in digit I–IV (Russell & Bauer, 2008; Suzuki et al., 2011).

Ventral group

Musculi flexores digitores breves (fdb)

-flexor digitorum communis sublimis (Zug, 1971; Walker, 1973)

-flexores digitores breves (Russell & Bauer, 2008)

-flexor digitorum brevis profundus and superficialis (Suzuki et al., 2011)

Flexores digitores breves by Russell & Bauer (2008) was given priority as it reflects better that it is the counterpart to extensores digitores breves (Table 3).

These muscles represent the deeper layers of the fdlh. If these were differentiated in plesiosaurs, then they are of an aponeurotic origin and much reduced or fused with the overlying muscle. In crocodilians, fdb arise from the ventral proximal, anterior, and posterior calcaneum, the posterior edge of metatarsal V (Suzuki et al., 2011). In turtles and crocodilians, fdb originate from the flexor plate of fdlh (Zug, 1971; Walker, 1973; Suzuki et al., 2011), from the aponeurosis of the deep ge, and from the plantar tubercle of metatarsal V in lepidosaurs (Russell & Bauer, 2008).

If fdb are not reduced in plesiosaurs, then they insert into digits I–IV as supported by crocodilians, turtles, and lepidosaurs (Fig. 3A). Into which exact phalanges fdb insert is difficult to reconstruct, as there appears to be no consensus across EPB so all options mentioned below are equally likely. In Testudines, the tendons of insertion bifurcate and insert anteroventrally and posteroventrally into phalanx I of digit I–IV. In sea turtles these muscles are absent (Walker, 1973). In lepidosaurs, they insert into the proximal phalanx I of digit I, onto the proximal and posterior phalanx I, the proximal phalanx II of digit II, the proximal phalanx I on digit III (Snyder, 1954). In crocodilians, fdb insert ventrally and proximally into all phalanges of digit I–IV and into the ungual phalanges (Suzuki et al., 2011).

Musculus extensor hallucis proprius (ehp)

-extensor hallucis proprius (Zug, 1971; Walker, 1973)

-adductor hallucis dorsalis (Suzuki et al., 2011)

-adductor et extensor hallucis indicus (Russell & Bauer, 2008)

The origin of ehp is on the distal dorsal fibula as suggested by all three taxa in plesiosaurs (Fig. 3B). Ehp has its origin area on the distal dorsal fibula in turtles, crocodilians, and lepidosaurs, distal and adjacently to pb and pl origins (Walker, 1973; Russell & Bauer, 2008; Suzuki et al., 2011). Additionally, in lepidosaurs it arises from the astragalocalcaneum (Russell & Bauer, 2008).

An insertion on metatarsal I in plesiosaurs is well supported by the EPB. An attachment on metatarsal II or to the penultimate phalanx of digit I is possible of which the latter could aid in extending the first digit during the upstroke and flipper twisting (Figs. 3B and 5B). In turtles ehp inserts distoposteriorly into metacarpal I, proximoposterior into phalanx I, and anterodistal into phalanx I which is also the penultimate phalanx in Pseudemys (Walker, 1973). In crocodilians, ehp inserts into the anteroproximal metatarsal I (Suzuki et al., 2011). In lepidosaurs, ehp inserts by a bifurcated tendon into metatarsal I anteriorly and posteriorly and into metatarsal II (Russell & Bauer, 2008).

Musculus adductor digiti quinti (addV)

This muscle is only described for lepidosaurs by Russell & Bauer (2008), neither for turtles (Walker, 1973), nor for crocodylians (Suzuki et al., 2011). There are no synonyms (Russell & Bauer, 2008) (Table 3).

The origin and insertion site of addV is reconstructed as it was found in lepidosaurs for plesiosaurs as it adds to flipper twisting (Fig. 3A). In lepidosaurs, addV arises by two heads from the tubercle of metatarsal V anteroventrally and proximally from it (Russell & Bauer, 2008). It attaches to proximoventral phalanx I and to the penultimate phalanx of digit V (Russell & Bauer, 2008).

Musculus flexor hallucis (fh)

-flexor hallucis (Russell & Bauer, 2008)

-flexor hallucis brevis superficialis and flexor hallucis brevis profundus (Suzuki et al., 2011)

This muscle has not been reported for turtles. The term fh by Russell & Bauer (2008) was given priority as it was decided to summarize the subportions of this muscle which would be expressed by the term employed by Suzuki et al. (2011) (Table 3).

For plesiosaurs, an origin area on the fibulare or the adjacent distal tarsal element is equally likely (Fig. 3A). In crocodilians, the two muscle bellies of fh arise from the posteroventral calcaneum (Suzuki et al., 2011). In lepidosaurs, fh originates from the anterior surface of the distal tarsal IV (Russell & Bauer, 2008).

In plesiosaurs, fh inserts into the proximoventral phalanx I of digit I as in lepidosaurs and crocodilians (Fig. 3A). In crocodilians, fh inserts into the anteroventral shaft of metatarsal I and into the posteroventral and proximal phalanx I of digit I (Suzuki et al., 2011). In lepidosaurs, fh attaches to the metatarsophalangeal joint of digit I (Russell & Bauer, 2008).

Muscle functions

Foreflipper

Flipper twisting, in addition to flipper rotation, is a crucial component of lift-based locomotion (or underwater flight) in vertebrates, as has been documented by observational studies and by hydrodynamic studies (Davenport, 1987; Walker & Westneat, 2000; Walker & Westneat, 2002; Witzel, Krahl & Sander, 2015; Witzel, 2020). Founded on the EPB and in comparison to functionally analogous extant secondarily aquatic tetrapods, plesiosaur muscles were reconstructed that could have contributed to underwater flight and flipper twisting.

Several of the reconstructed pectoral muscles turn out to be humeral protractors. These are dc, ds, and the respectively anterior portions of sc, p, scs, and eventually the most cranial portion of ld (s. Abbreviations). Cb, cl, bb, and the posterior portions of sc, p, scs, ld act as humeral retractors. Ds, scs (elevation via deflection on the tuberosity), tb, and ld (deflection on tuberosity) elevate the humerus. Shp may have a minor elevational function. Depressors of the humerus are cb, cl, dc, bb, p, and sc. Pectoral muscles that are able to rotate the humerus and hence the leading edge of the flipper downward are dc, shp, bb, p (posterior portion), scs (anterior portion), and tb. Humeral rotators that enable an upwards rotation of the flipper leading edge are ds, cb, cl, p (anterior portion), scs (posterior portion), and ld (Figs. 4A and 4B; Table 1).

Ecu crosses the carpus diagonally from anteroproximal to posterodistal. It displaces the ulna slightly relatively dorsally to the humerus. Alternatively, although weaker supported by the EPB, but necessary to perform underwater flight, it could allow flexion of metacarpal V on the adjacent distal carpal. Edc is aponeurotic as in other sauropsids and it extends the metacarpals on the adjacent distal carpal elements. Sl and ecr relocate the radius slightly dorsally relative to the humerus. An insertion on the radiale allows displacement of the whole radial side of the carpus. Sm abducts metacarpal I on the adjacent distal carpal element and it allows extension to a minor degree. Edbp and edbs extend all digits (Figs. 4A and 4B; Table 1).

Pte crosses the carpus diagonally from posteroproximal to anterodistal and shifts the radius slightly relatively ventral to the humerus (s. Abbreviations). Fcu displaces the ulnar side of the carpus slightly ventrally in relation to the humerus. Additionally, an insertion on metacarpal V allows the plesiosaur to flex metacarpal V on the adjacent distal carpal. Fdlf (and fdls) forms an aponeurosis with five tendons that allow the flexion of all digits. Fcr flexes metacarpal I on the distal carpal element. An insertion on the radial side of the carpus allows displacement of the whole side of the carpus. AbdV abducts and slightly flexes digit V. Apb could either abduct or flex digit I on the metacarpophalangeal joint or flex metacarpal I on the adjacent distal carpal. Adm adducts and flexes digit V on the metacarpophalangeal joint (Figs. 4A and 4B; Table 1).

Three muscle insertions were suggested in the results due to functional reasons, i.e., that could aid in flipper twisting as described above. These are: (1) flexor carpi ulnaris insertion into metacarpal V instead of ulna and ulnare, (2) flexor carpi radialis insertion into metacarpal I instead of radiale and distal carpal I, and (3) an abductor pollicis brevis insertion into metacarpal I instead of the first phalanx of the first digit. The states solely implied by the EPB would not compromise the suggested flipper twisting mode of the foreflipper. This is because flexor carpi ulnaris abductor digit V and flexor carpi radialis abductor pollicis brevis could fully by themselves take on the respective contribution to flipper twisting. The difference in muscle insertion of the abductor pollicis brevis to either metacarpal I or the first digit would not imply any change in its general function regarding its aid in flipper twisting.

Hindflipper

Muscles that enable an elevation of the femur are pi, it, ife, ifi, cfb, cfl, fte (portion from ilium), and fti (portion to vertebral column) (s. Abbreviations). Pe, af, i, pi, fte (portion from ischium), and fti (ischial portion) power femoral depression. Pe (pubic portion), pi (pubic portion; to vertebral column), a, pti aid in femoral protraction. Pe (ischial portion, but only if femur protracted, minor function), pi (ischial and iliac portion only minorly and only if femur protracted), af (lateroposterior ischial portion; i, ifi (minorly), it, ife, cfb, cfl, fte, and fti retract the plesiosaur femur and flipper (Figs. 5A and 5B; Table 2).

Responsible for the downward rotation of the flipper leading edge are i, cfb, cfl, pi (pubis portion), pe (ischium portion), ifi: clockwise rotation (as long as fibula below origin) (s. Abbreviations), pit, both fti portions, both fte portions, a (if femur elevated), and pti (if femur elevated). Ife, pi (ischium and ilium portion), pe (pubis portion), it, a (if femur depressed), ifi (as long as fibula above origin), and pti (if femur depressed) may rotate the flipper leading edge upward (Figs. 5A and 5B; Table 2).

Pl and pb extend metatarsal V on adjacent tarsal element and abduct metatarsal V. F, a, and it contribute to a slight dorsal displacement of the tibia on the distal femur. Ta abducts metatarsal I relatively anteriorly on the adjacent distal tarsal. Edl extends digits I–IV (on tarsometatarsal joints). Ehp aids in extension or adduction of metatarsal I (on tarsometatarsal joint) depending on how it is reconstructed. Edb extend the phalanges of digit I–V (Table 2). Ge and gi is a flexor of all five digits in all phalangeal joints. It also acts on metatarsal I and V. Fdlh flexes the phalanges of all five digits and fdb are the flexors of digits I–IV lying deep to the former. AddV is the flexor of digit V and fh flexes digit I. Pp is responsible for flexion of tarsometatarsal joints of digit I, and digit II and III (Figs. 5A and 5B; Table 2).

Two muscle insertions were suggested in the results section due to functional reasons, i.e., that they could aid in flipper twisting as described above. The first one is the insertion of extensor hallucis proprius into the penultimate phalanx. An insertion into metatarsal I is more parsimoniously supported by the EPB. However, this would not change this muscle’s function as adding to hindflipper twisting. Second, reconstruction of adductor digiti quinti is supported by lepidosaurs. In crocodiles and turtle, this muscle has not been found. If this muscle were not reconstructed, a more independently (from the aponeurosis) acting gastrocnemius internus and externus muscle belly could aid in flexing digit V instead. It also would be possible that the water flow passively shaped the hindflipper trailing edge.

Discussion

Muscle attachment sites

Plesiosaur muscles have been reconstructed before by Watson (1924), Tarlo (1958), Robinson (1975), Lingham-Soliar (2000), Carpenter et al. (2010), and Araújo & Correia (2015). It is not clearly stated on which taxa the plesiosaur muscle reconstructions are based on by Watson (1924), Tarlo (1958), Robinson (1975), and Lingham-Soliar (2000) which makes it difficult to reproduce their results and hamper comparison. Carpenter et al. (2010) and Araújo & Correia (2015) based their plesiosaur myology on the extant phylogenetic bracket (EPB). Carpenter et al. (2010) used lepidosaurs (tuatara) and turtles for the EPB. Araújo & Correia (2015) used lepidosaurs, crocodilians, and turtles for the EPB.

So far, none of the studies have reconstructed the complete fore- and hindflipper musculature of plesiosaurs. Araújo & Correia (2015), Tarlo (1958), and Watson (1924) reconstructed muscles of the pectoral girdle. Robinson (1975), Lingham-Soliar (2000), and Carpenter et al. (2010) reconstructed muscles of the pectoral and pelvic girdle. P, scs, sc, cb, cl, ds, and ld were reconstructed by all six above mentioned studies (s. Abbreviations). Dc was not reconstructed by Tarlo (1958). Scapulohumeralis anterior was not reconstructed by Araújo & Correia (2015) and Carpenter et al. (2010) and this study. Shp was not reconstructed by Tarlo (1958), Watson (1924), and Robinson (1975). Tb and bb were only considered by Robinson (1975) and Araújo & Correia (2015) and only bb by Carpenter et al. (2010). An attempt at reconstructing distal plesiosaur humerus and flipper musculature has been made exclusively by Robinson (1975) who reconstructed a highly reduced foreflipper (fcr, fcu, and long flexors) which appears almost cetacean-like (compare to Cooper et al., 2007).

Cfb, cfl, pe, pi, and ife have been reconstructed by Robinson (1975), Lingham-Soliar (2000), and Carpenter et al. (2010) (s. Abbreviations). It, ifi, and pit have been reconstructed by Robinson (1975) and Lingham-Soliar (2000). Only Robinson (1975) has reconstructed f, a, i, af, and distal hindflipper musculature (peroneus, and ta+plantar aponeurosis). Pti, fti, and fte are present in Sauropsida (Romer, 1923; Snyder, 1954; Zug, 1971; Walker, 1973; Russell & Bauer, 2008; Otero, Gallina & Herrera, 2010; Suzuki et al., 2011) but have not been reconstructed by any of the authors. Almost no muscles that originate from the distal femur or hindflipper have been reconstructed for plesiosaurs until the current study.

The current study is the first study, that provides complete muscle reconstructions of the plesiosaur (Fig. 1A) fore- and hindflipper based on a firm foundation, i.e., by combining inference from the EPB (Lepidosauria, Crocodylia, Testudines) (Figs. 1B and 1C) with a comparison to functional analogues. The functional analogues helped to identify “boundary conditions”, i.e., helped to inform on broader flipper myology patterns. This led to the reconstruction of 52 locomotory muscles in total. 26 foreflipper muscles (12 pectoral, eight antebrachial, and six manual muscles) and 26 hindflipper muscles (15 pelvic, six crural muscles, and five muscles of the pes) (Figs. 2 and 3). Further, ligaments in the pectoral and pelvic girdle appear unlikely due to lacking osteological evidence and their hypothetical courses. Contrastingly, flexor and extensor retinacula and intermetacarpal/intermetatarsal and metacarpodigital/metatarsodigital ligaments were reconstructed as part of the flipper twisting mechanism.

Overall in terms of muscle origin sites on the pectoral girdle, this study can mostly confirm the results of Araújo & Correia (2015). Generally, the current study (Figs. 2 and 3) shows similarity in plesiosaur pectoral girdle myology to all other muscle reconstructions published so far, too. Especially in the hindflipper, it becomes apparent that most authors only reconstructed a fraction of sauropsid hind limb musculature, which makes it difficult to compare the results. Generally, the reconstructed pelvic girdle muscles of plesiosaurs (Figs. 3A and 3B) differ more than the results of the pectoral girdle muscles in comparison to the published literature. This could be due to the numerous two-joint muscles in the hind limb of sauropsids which show considerable variability (Walker, 1973, Zug, 1971; Romer, 1923; Otero, Gallina & Herrera, 2010; Suzuki et al., 2011, Gatesy, 1997 Snyder, 1954; Russell & Bauer, 2008). This variability leads to more ambiguous muscle reconstruction options offered by the EPB. Furthermore, Robinson (1975) reconstructed an iliopubic ligament that serves as attachment site for muscles, while this study did not reconstruct it. As a consequence, muscles that take origin from this ligament in Robinson’s (1975) study (e.g., m. iliotibialis) had to be placed onto adjacent bony areas in accordance with the EPB in the current study.

Differences between reconstructed pectoral girdle muscles between the different plesiosaur myology studies often result from a variable muscle size or from different extant taxa used for the EPB and therefore varying muscle attachment sites. This can be exemplified by the reconstruction of p: Araújo & Correia (2015) completely reduced p, based on the premise, that it arises in crocodiles and lepidosaurs mostly from the interclavicula and the sternum. As both are either absent or diminutive in plesiosaurs, Araújo & Correia (2015) argue that there would be no support for a p. Further, they write that EPB would not support a spreading of p onto other bony elements. Yet, in turtles, the p origin has spread onto the plastron and p is also the largest locomotory muscle of the forelimb (Walker, 1973; Wyneken, 2001; Krahl et al., 2019). Additionally, there is no extant sauropsid which has a reduced p to the knowledge of the authors. Therefore, the current study reconstructed p onto homologous bony areas in Cryptoclidus eurymerus which agrees with the muscle’s general position in the sauropsid body comparable to Robinson (1975), Lingham-Soliar (2000), and Carpenter et al. (2010). Tarlo (1958) did not figure the p origin and Watson (1924) reconstructed only an attachment onto the gastralia. The two studies that reconstructed a small to reduced p, Watson (1924) and Araújo & Correia (2015), interpreted plesiosaurs as rowers, while the studies that reconstructed a larger p, Tarlo (1958), Robinson (1975), Lingham-Soliar (2000), Carpenter et al. (2010), and the current study, interpret plesiosaurs as (partial) underwater fliers. As the interpretation of the EPB seems to be subjectively biased depending on the outcome one would like to find, the authors of the current study give an explanation why a specific option was chosen each time to counter this effect and make the decisions more transparent. This has been suggested by Witmer (1995) for working with the EPB in general and Araújo & Correia (2015) did so, too, in their plesiosaur muscle reconstructions. To sum up, the differences in the reconstructed plesiosaur pectoral and pelvic girdle muscles may result from different extant taxa used to infer plesiosaur muscles, incompletely reconstructed fore- and hindflipper musculature, the presence or absence of ligaments, and from different decisions based on what the authors possibly presumed plesiosaurs would have been swimming like.

The comparison of muscle insertions into the proximal plesiosaur humerus and femur are hampered due to several reasons: Robinson (1975) reconstructed the anterior side of the humerus and femur as pointing ventrally, so the anterior side becomes ventral and the dorsal side anterior. This consequently leads to completely different muscle attachment areas as in most other studies. The proximal humerus muscle insertions figured by Watson (1924) are partially also similarly shifted onto another humerus side as by Robinson (1975). The reconstructed pectoral musculature by Tarlo (1958) and Lingham-Soliar (2000) and the pelvic girdle musculature by Lingham-Soliar (2000) are rather schematic, lack detail, and are more superficial as the reconstructions by Watson (1924), Robinson (1975), Carpenter et al. (2010), Araújo & Correia (2015), and the current study. Carpenter et al. (2010) figured three dimensionally how muscles attach to humerus and femur, but they did not figure actual muscle attachment sites, which leaves room for interpretation. Additionally, plesiosaur humeri and femora have a derived morphology, e.g., both have only one proximal dorsal process (tuberosity on the humerus and trochanter on the femur). Plesiosaurs have lost common morphological structures like a markedly set off deltopectoral crest, or the femoral trochanters which are present in the three extant sauropsid groups used for the EPB in the current study (e.g., Walker, 1973; Russell & Bauer, 2008; Meers, 2003). This makes it difficult to place muscle attachment sites on the plesiosaur long bones, because the direct association of muscles with common morphological sauropsid long bone structures is not as apparent and probably leads to different muscle attachment reconstructions. Broadly, the proximal humerus muscle insertion sites presented here resemble those of Araújo & Correia (2015). Differences, like e.g., the completely reduced p by Araújo & Correia (2015), are transferred to the muscle insertion areas on the humerus as well.

The results (Figs. 2 and 3) agree mostly with the reconstruction of the distal plesiosaur fore- and hindflipper musclulature by Robinson (1975). Nevertheless, the current study provides many more fore- and hindflipper muscles for the first time that are arranged in a complex flipper twisting mechanism.

It has to be mentioned that there is considereable variability in plesiosaur fore- and hindflipper morphology leading to different flipper forms. Cryptoclidus eurymerus appears to reflect a rather rare flipper morphology in plesiosaurs, with anteriorly and posteriorly expanded humeri and femora (Krahl, 2021). Within Lepidosauria, Crocodylia, and Testudines there is considerable myological intraspecific variability (e.g. Russell & Bauer, 2008; Meers, 2003; Walker, 1973). In these three extant taxa, these differences may be due to different locomotory styles (e.g., walking, rowing, and underwater flight in turtles (Walker, 1973)). Contrastingly, the myological differences between modern crocodiles do not seem to imply fundamentally different locomotory styles (Meers, 2003). It is probable that plesiosaur flipper variability does not reflect greatly different locomotory styles, because overall the locomotory apparatus in plesiosaurs is very similar, but that the variability reflects different ecological adaptations as seen in e.g., modern cetaceans (Woodward, Winn & Fish, 2006). We are confident that the current muscle reconstructions discuss the options for variability enough to adapt the muscle reconstructions to different plesiosaur flipper morphologies in future works.

Muscle functions

While Watson (1924), Tarlo (1958), Robinson (1975) assigned functions to every muscle they reconstructed for plesiosaurs, Lingham-Soliar (2000) only assigned functions to p, cb, cl, and sc, and Carpenter et al. (2010) deduced functions for most reconstructed muscles except for dc, ds, and bb. Araújo & Correia (2015) reconstructed ld but assigned no function to it. They also suggest that p is reduced in plesiosaurs and therefore did not assign any muscle functions to it. Hindflipper muscle functions were given by Carpenter et al. (2010) and Robinson (1975) for all muscles they reconstructed (Tables 1 and 2).

In the foreflipper we found that p, sc, scs, and ld probably had muscular subdivisions, that were recruited during different phases of the limb cycle with different functions (e.e., Russell & Bauer, 2008) (Figs. 4A and 4B). Similarly, in the hindflipper the reconstruction of various muscular heads of pe and pi (Figs. 5A and 5B) led to more numerous and more variable functions of each muscle that contribute to different parts of the limb cycle. Except for Watson (1924), who subdivided scs, too, no other author has provided potential muscular compartementalization so far, although it is common in extant sauropsids (s. Abbreviations). All other studies appear to have divised only the main muscle functions, so they usually identified one or two muscle functions. The current study aimed at identification of all functions each muscle can have including minor functions because they all contribute to the flipper beat cycle. This led to a great discrepancy in comparison to all other studies simply because for many muscles no one has tried to identify its potential functions or it was rather focused on the main function (Tables 1 and 2).

Limb cycle and myological mechanism for flipper twisting

Foreflipper

In vertebrates that employ lift-based locomotion, i.e., underwater flight, flipper twisting has been shown to be necessary for efficiency by direct observations and by hydrodynamic modelling, additionally to flipper rotation (Davenport, 1987; Walker & Westneat, 2000; Walker & Westneat, 2002; Witzel, Krahl & Sander, 2015; Witzel, 2020). The current study provides muscles, based on muscle reconstructions with the EPB and in comparison to functional analogues, that could have enabled plesiosaurs to sustain underwater flight including flipper twisting: At the beginning of the downstroke the flipper leading edge is rotated downwards by approximately 19° (Witzel, Krahl & Sander, 2015; Witzel, 2020) by the humeral rotators (bb, scs (anterior portion), dc) (s. Abbreviations). To eliminate angle-dependent drag, an angle of rotation of 19° was needed, especially in the case of highest downstroke velocity, to generate optimal lift, i.e., for efficient underwater flight. This downwards rotation is synchronized by an upward rotation of the flipper trailing edge (shp, p, and tb). Then, the humerus is depressed and retracted accompanied by flipper twisting along the flipper length axis by slightly displacing the carpus ventrally on the flipper leading edge and dorsally on the flipper trailing edge and flexion of digit I while the following digits are decreasingly depressed towards digit V. The pte slightly displaced the radius ventrally against the humerus in concert with b and fcr that flexes metacarpal I on the adjacent distal carpal or displaces the radial side of the carpus slightly ventrally. Both, pte and fcr, slightly compressed and bulged the carpus. Apb acts together with them and flexes digit I. At the same time the posterior side of the carpus and manus experience upward twisting. The humeral tb head and ecu displace ulna and ulnare slightly dorsally to the humerus, ecu alternatively extended metacarpal V on the adjacent distal carpal in a concerted action with an individually acting edbs and edbp slip to digit V (Table 1). The ligaments (extensor and flexor retinaculum and the intermetacarpal and metacarpodigital ligaments) contribute passively to flipper twisting. When flipper twisting is initialized by flexion or extension of e.g., digit I, the ligamentous system of the hand passively induces digit II, III, IV, and V to successively follow the movement of digit I (Table 1) (Figs. 4A and 4B; Table 1).

The foreflipper upstroke begins with an upwards rotation of the flipper leading edge and a simultaneous downwards rotation of the flipper trailing edge (ds and ld rotate the humerus anteriorly upward, while cb, cl, p (anterior portion), ld, and scs (posterior portion) rotate the humerus posteriorly downward) (s. Abbreviations). Then, humeral elevators and protractors draw the flipper back by drawing the humerus into the starting position for the downstroke. At the same time the flipper is being twisted along the flipper length axis but in the opposing direction as described above for the downstroke. Twisting upwards of the anterior carpus and manus is enabled by muscles on the dorsal side of the plesiosaur foreflipper. Sm and ecr displaces the radius slightly dorsally to the humerus or even the whole radial side of the carpus while sm extends metacarpal I on the distal carpal, while an individually movable edbs and edbp slip to digit I extends the first finger. It is well possible that the two latter may also be involved in a very moderate hand-cupping, which could be opposed by adm on the palm. Muscles on the ventral foreflipper side twist the flipper leading edge relatively downwards in a concerted action. Fcu displaces ulna and ulnare slightly downwards relatively to the humerus or flexes metacarpal V on the adjacent distal carpal and abdV initiates the flexion of digit V. Sm dorsally and adm could induce dorsal, respectively ventral cupping of the plesiosaur carpus (Figs. 4A and 4B; Table 1).

Hindflipper

The downstroke is initialized by the femoral rotators. Pit, both fti portions, both fte portions, a (if femur elevated), pi (pubis portion), pe (ischium portion), and pti (if femur elevated) rotate the femur anteriorly downward and i, cfb, cfl, ifi rotate it posteriorly upward (s. Abbreviations). Femoral depressors and retractors move the flipper downwards and backwards through the water. At the same time the hindflipper is twisted in a similar fashion as the foreflipper. The leading edge is twisted increasingly downwards along the flipper length axis and the trailing edge upward. Downwards twisting of the hindflipper is managed by muscles that are situated on the ventral tarsus and pes. Pp and ta flex metatarsal I on distal tarsal I and fh flexes digit I independently from the other digits. Fte, fti, or pti displace the tibia relatively ventral to the femur (s. Abbreviations). It is likely that a contraction of the ge could also induce flipper twisting. Muscles situated on the dorsal plesiosaur tarsus and pes aid in curling the flipper trailing edge upwards during downstroke. Pb and pl extend metatarsal V on the distal tarsal. An edb slip running to digit V which is individually controllable is responsible for extension of digit V. Ifi displaces the fibula slightly dorsally to the femur. The extensor and flexor retinaculum and the intermetacarpal and metacarpodigital ligaments could contribute passively to flipper twisting as described above for the plesiosaur foreflipper (s. above; Figs. 5A and 5B; Table 2).

At the lowest point of the limb cycle, the flipper is rotated in the opposite direction by approximately 19° (Witzel, Krahl & Sander, 2015; Witzel, 2020) by muscles that either rotate the anterior femur upward or the posterior femur downward (it, a (if femur depressed) pti (if femur depressed) ifi (as long as fibula above origin), pi (ischium and ilium portion), pe (pubis portion), and ife) or those that insert further distally into the tarsus (s. Abbreviations). Then, femoral elevators and protractors return the flipper into its initial position. Simultaneously, the hindflipper is twisted (flipper leading edge curled upward and flipper trailing edge curled downward). Flipper twisting is initiated by f (a and it) which displaces the tibia slightly dorsally to the femur (s. Abbreviations). Ehp extends metatarsal I on the distal tarsal in concert with the slip to digit I of edb which extends the first toe. It is difficult to determine how flipper twisting is induced on the ventral plesiosaur hindflipper, but it is likely that a contraction of the gi head initializes it. Further, addV aids by independently flexing digit V. In addition, fh and pp induce ventral cupping of the tarsus of the hindflipper. Their respective dorsally situated counterpart is ehp (Figs. 5A and 5B; Table 2).

Especially in the hindlimb, complex patterns of tendons interconnecting seemingly independent muscles are known: cfl tendon which is associated with ifi and ge in crocodilians (Otero, Gallina & Herrera, 2010) inserts into ge, fte and fti are tendinously associated and insert into gi (Romer, 1923; Walker, 1973; Russell & Bauer, 2008; Otero, Gallina & Herrera, 2010; Suzuki et al., 2011). Gi and ge themselves are a highly complex, layered, partially independent, partially dependently acting muscles (Snyder, 1954; Zug, 1971; Walker, 1973; Russell & Bauer, 2008; Otero, Gallina & Herrera, 2010; Suzuki et al., 2011) (s. Abbreviations). It appears that the functional implications of these interconnections in recent Sauropsida are not studied in such depth, that it is possible to interpret their functions in too much detail for fossil taxa (Table 2).

Conclusions

Most studies on plesiosaur locomotion point towards underwater flight as their main locomotory style (Robinson, 1975, 1977; Tarlo, 1958; Lingham-Soliar, 2000; Carpenter et al., 2010; Liu et al., 2015; Muscutt et al., 2017; Krahl, 2021). Underwater flight is characterized by a mostly dorsoventral excursion of the flippers or fins with a minor anteroposterior component (Rivera, Rivera & Blob, 2013). Additionally, extant underwater flying vertebrates including plesiosaurs twist their flippers or fins along the length axis for increased efficiency (Davenport, 1987; Walker & Westneat, 2000; Walker & Westneat, 2002; Witzel, Krahl & Sander, 2015; Witzel, 2020). This crucial component of underwater flight has largely been lacking from plesiosaur locomotory studies (Krahl, 2021). Therefore, the authors aimed at reconstructing muscles for the plesiosaur Cryptoclidus eurymerus (IGPB R 324) (Fig. 1A), that do not only support the dorsoventral fore- and hindflipper down- and upstroke including flipper rotation, but also active flipper twisting.

A comparison to the myology of functional analogues shows, that a reduction of muscles takes place in the distal flipper muscles, especially of muscles that spread the digits. Further, with the help of the EPB (lepidosaurs, crocodilians, and turtles) (Figs. 1B and 1C) plesiosaur fore- and hindflipper muscles were reconstructed (Figs. 2 and 3). Muscle functions were deduced geometrically (Figs. 4 and 5).

Fifty-two locomotory muscles were reconstructed in total posing the first comprehensive study that provides muscle reconstructions for the entire fore- and hindflipper of a plesiosaur (Figs. 2 and 3). Generally, the pectoral girdle muscle attachment sites are very similar to the reconstruction by Araújo & Correia (2015). Differences often derive from different taxon choices for the EPB and are given by the incompleteness of former studies. The interpretation of muscle functions is overall relatively different to other studies. This is partially due to the fact that most authors seem to have focused on main muscle functions and not on minor muscle functions. Further differences arise because we identified functional subdivisions in large muscles (e.g., p, scs), i.e., that may show different functions which were recruited during different or partially overlapping phases of the limb cycle.

Numerous humeral and femoral elevators, depressors, protractors, retractors, and rotators were identified (Figs. 4 and 5). Additionally, ligaments (flexor and extensor retinaculum, intermetacarpal/intermetatarsal and metacarpodigital/metatarsodigital ligaments) and muscles that contribute to a complex mechanism enabling flipper twisting are described.

We suggest that a plesiosaur flipper beat cycle could have looked like this: The humerus and femur were slightly rotated downwards (dc, shp, bb, p (posterior portion), scs (anterior portion), tb, i, cfb, cfl, pi (pubis portion), pe (ischium portion), ifi, pit, both fti portions, both fte portions, a, and pti) (~19°) and were strongly depressed (cb, cl, dc, bb, p, sc, pe, af, i, pi, fte (portion from ischium), and fti (ischial portion)) and little retracted (cb, cl, bb, and the posterior portions of sc, p, scs, ld, pe (ischial portion), pi (ischial and iliac portion), af (lateroposterior ischial portion; i, ifi, it, ife, cfb, cfl, fte, and fti) during the downstroke. Additionally, flipper length axis twisting was induced by muscles that twisted the flipper leading edges downward (pte, b, fcr, apb, pp, ta, fh, fte, fti, pti, ge) and the trailing edges upward (tb, ecu, edbs, edbp, pb and pl, edb, ifi). The successive digits followed the actively induced twisting of the first digit passively because of the intermetacarpal/intermetatarsal and metacarpodigital/metatarsodigital ligaments. Then, the propodials were rotated upwards (ds, cb, cl, p (anterior portion), scs (posterior portion), and ld, ife, pi (ischium and ilium portion), pe (pubis portion), it, a, ifi, and pti) by ~38° (= the sum of app. 19° downwards and 19° upwards rotation) during the fore- and hindflipper upstroke and were strongly elevated (ds, scs, tb, ld, shp, pi, it, ife, ifi, cfb, cfl, fte (portion from ilium), and fti (portion to vertebral column)) and little protracted (dc, ds, anterior portions of sc, p, scs, cranial portion of ld, pe (pubic portion), pi (pubic portion; to vertebral column), a, pti). At the same time, the flippers were twisted into the opposite direction, i.e., the leading edge was turned upward and the trailing edge downward myologically (sm, ecr, edbs and edbp, fcu, abdV, f, a, it, ehp, edb, gi, addV). Active flipper profile manipulation may have been possible (sm, adm, fh, pp, and ehp) which would have led to an asymmetry of the fore- and hindflipper profiles which in turn would have provided an increased efficiency over a symmetrical profile (Figs. 4 and 5). The current plesiosaur muscle reconstruction including the flipper twisting mechanism may provide the basis for e.g., future hydrodynamic or maybe finite element modeling studies that could possibly confirm the presence and importance of flipper twisting in plesiosaurs.

Supplemental Information

Supplemental Information 1 Homology of plesiosaur foreflipper and hindflipper myology.

Click here for additional data file.

The authors would like to thank D. Suzuki of Sapporo Medical University, Japan and S. Hayashi, of Okayama University of Science, Japan for discussion of crocodilian limb bone orientation. Thank you to Ian Freeman, at University of Surrey, Alton, Hampshire, UK for information on penguin anatomy and myology. The authors are grateful for comments by P. M. Sander at the Section of Paleontology, Institute of Geoscience, Rheinische Friedrich-Wilhelms Universität Bonn, Germany on earlier versions of this manuscript which led to its great improvement. We would like to thank G. Oleschinski at the Section of Paleontology, Institute of Geoscience, Rheinische Friedrich-Wilhelms Universität Bonn, Germany for kindly providing us with the great picture of IGPB R 324. Last but not least, we would like to thank Kiersten Formoso and two anonymous reviewers for their constructive criticism which led to a greatly improved manuscript.

Additional Information and Declarations

Competing Interests

Author Contributions

Data Availability

The authors declare that they have no competing interests.

Anna Krahl conceived and designed the experiments, performed the experiments, analyzed the data, prepared figures and/or tables, authored or reviewed drafts of the paper, and approved the final draft.

Ulrich Witzel conceived and designed the experiments, authored or reviewed drafts of the paper, and approved the final draft.

The following information was supplied regarding data availability:

This is a descriptive study, so there is no other raw data.

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
