# Peer review of "Foreflipper and hindflipper muscle reconstructions of Cryptoclidus eurymerus in comparison to functional analogues: introduction of a myological mechanism for flipper twisting"

_PeerJ, doi:10.7717/peerj.12537_

## Round 0.1 · original submission · Major Revisions

Three reviewers have commented on the manuscript in detail. The comments may feel overwhelming, but take "the forest for the trees"-- overall, they are supportive and would like to see this published. The writing needs some restructuring as recommended and some extra proofreading.

Reviewer 2 is quite critical but please take their tone as constructive-- they are trying to help improve the paper, even though their perspective on the ideal strategy for publishing this work differs. Not all recommendations must be followed, you can make a case in your Response why not, but the majority should be.

It is your manuscript so you need not split it into 2 papers (there is no length limit here) but you could. Most importantly, the science must be sound (which reviewers generally agree it is) and clear (which is where some work is needed), but you should feel relieved that the majority of work to be done is structural (wording/organization) rather than analytic.

We look forward to the revised manuscript. If you would like guidance on revising you may contact us for advice.

Reviewer 1 ·

Basic reporting

The clarity of the manuscript could be improved by changes to the structure, additional figures, language editing.

1. Most importantly, the manuscript's structure is confusing. The divisions between introduction, results, discussion, and conclusions seem arbitrary, and the order of the sections is hard to follow. For example, the detailed results of previous plesiosaur muscle reconstructions (90-122) and the summary of functional anatomy of extant analogues (124-309) from the introduction both seem more appropriate for the discussion. Also, the paragraph about sea turtle humeri (346-358) does not belong in Methods. Also, the Myology section at the end of the discussion (2573-2631) really belongs in methods. Finally, the subject matter seems to jump around a lot, making it difficult to find relevant information or follow a thought to its conclusion. Having a clear, strong structure is especially important in a long manuscript like this one.

2. The figures provided are good, but they are insufficient and poorly referenced. There are only two figures showing the reconstructed origins and insertions of muscles on the foreflipper and hind flipper, and only one reference is made to figures within the text (p58). At the very least, there should be a figure showing the actual muscle reconstruction used to deduce muscle function – this would require reconstructing the course/line of action of the muscles in addition to their attachments. Figure references should be added as appropriate; for example in lines 70-86. Additional figures that would strengthen the manuscript include: phylogenetic tree showing EPB of plesiosaurs, appendicular muscles in extant analogues, joint surfaces in plesiosaurs. (Also, what does “?” indicate in figures 1-2? And please indicate anterior-posterior in the figures.) If additional figures and tables were added, the text could be greatly reduced and the results would be easier to understand.

3. It is not clear what raw data were used in the study. Limb skeletons and histological sections referred to in Methods were not described or figured (392-394).

4. The English is generally good, but in certain passages grammatical mistakes and word choices make the meaning unclear. Examples include:
- Abstract: “possible to obtain” – do you mean identify, or reconstruct?
- 63: I think you mean compressed, not depressed
- 66: hemispherical, not round
- 2290-2292: Confusing sentence, please re-word
- 2305: “which could be corroborated” – does this mean your results agree?
- 2360: “could be supported” – again, does this mean your results agree?
- 2365 and throughout: these figures are not muscle reconstructions, they are muscle maps.
- 2610: I think you mean relatives, not predecessors
- 2623: “in the course of reduction” – what does this mean?
- 2646: delete “partially”
- Please spell out muscle abbreviations the first time they are used in a section.

5. - The introduction is generally complete and well referenced. A reference and/or explanation is needed in line 67 for “joints are relatively stiffened.”

Experimental design

The research question is well-defined and fills a knowledge gap, and the choice of methods is appropriate. However, the methods are not well enough described to allow replication. For example: were osteological correlates in fossils used, and if so which ones? Do the “limb skeletons” and histological sections examined refer to extant or extinct taxa, and which particular taxa (391-394)? How were decisions made about equivocal EPB results (e.g., 376-380)? How exactly were muscle functions deduced, particularly rotation (e.g., 419-423)? Precisely what criteria were used to establish homology (line 315) and what does “deducible” mean in this context (lines 363-364)? Which homologies were established by topology, and which were from this study as opposed to previous ones? What is meant by “if EPB was little informative, Sphenodon was considered as well” (lines 324-325)?

Validity of the findings

The results add to the literature by providing a more comprehensive, phylogenetically based muscle reconstruction in plesiosaurs. Provision of underlying data and clearer explanation of interpretation of results would demonstrate their validity.

1. Most importantly, the data used to deduce muscle function are not described or provided. Secondarily, EPB uses osteological correlates in fossils, but no fossils other than the taxon in the reconstruction or osteological correlates are listed. In addition, a table showing which muscles are present in which bracket taxa according to which source would be extremely helpful.

2. It is sometimes unclear what is speculation. For example, the section “Limb cycle and myological mechanism for flipper twisting” does not clearly state whether it is based on extant taxa, literature, modelling, or something else. The underlying data and logic need to be more clearly presented. In general, more justification needs to be provided in the discussion section: not just “we agree” or “we disagree” with previous interpretations, but presenting arguments for each and the logic for preferring one or the other (e.g., 2174-2179, 2206, 2209-2210, 2326).

3. The conclusions section is generally well written, concise, and well supported. However, it does not fully address the original research question: what muscles contribute to flipper twisting? I suggest shortening this section and focusing on the research question and its broader implications.

Additional comments

The research question is, what muscles could have produced flipper twisting in plesiosaurs? To address this question, they authors produced the most comprehensive muscle maps for a plesiosaur to date using extant phylogenetic bracketing and deduced muscle function based on musculoskeletal geometry. The results fill a knowledge gap and could help explain plesiosaur functional morphology and locomotion. There are substantial problems in the methods section and the overall structure of the manuscript that need to be addressed, plus some additional issues with the way data and conclusions are presented and with the manuscript’s language. Currently, the results are difficult to understand and impossible to replicate. In addition, readability and thus impact could be improved by shortening the text and presenting some of the information as figures, tables, or supplementary information.

Reviewer 2 ·

Basic reporting

The manuscript needs extensive proofreading throughout the whole text. Please pay attention to present tense, past tense, and passive voice. This manuscript was written by more than one author, so please refer to “we” instead of “I”. Examples for the above-mentioned problems can be found in my comments in the pdf. The text varies between paragraphs consisting of sometimes extremely short and normal to long sentences. Please unify your writing style.
The literature was properly cited and the topic was well covered by the cited literature.
Restructuring the manuscript would really help to make some things clearer. The texts about extant animal groups in the introduction should be shortened, moved to a new chapter and illustrated by a figure. The abbreviations of all muscle names should be placed where they can be more easily found. Regrouping them in hind- and forelimb would make the lists shorter and easier to overview.
The existing figures need extensive work. Please redraw all lines and close gaps in the lines. Add directions like anterior or posterior to clarify what exactly the reader looks at. Add bone names (in full length or abbreviations) and the thyroid fenestra. This would greatly improve the overview over your figures. Select colors which can be distinguished more easily (for example p, bb, and ecu and pte and ecr might be quite hard to tell apart). Add the missing muscles (b, sl, fdls). Explain the dotted lines and the question marks. Add Roman numerals at the end of each finger/toe to make the finger count clearer. Is there a special order in which you sorted the muscles in your list? If yes, please explain. If no, please sort in a comprehensible order. Resize the whole figure.
Your figures are not mentioned anywhere in the text. Please add mentions to the texts of all described muscles (for example “see Fig 2 a”)
Please add figures of the muscle locations (like your Figures 1 and 2) in the main groups you used as EPB. It would be way clearer if the states in the different groups could be seen in a comparative illustration.
Please add figures for the comparison of all plesiosaur muscle reconstructions. This would allow a direct comparison without looking up everything in several different publications.
The text about EPB misses basic information about EPBs, why you used these groups, and any explanations about the level of inference. The latter should also be mentioned for each individual muscle to describe how similar the insertion and origin is in the different groups used for the EPB.
There is nothing written about the used specimen, except its inventory number and the museum where it is located. Where was it found? What’s its age and find location? Please mention the first describer Phillips 1871 directly behind the name of the species. How complete is the specimen? Please add a photograph or illustration of the specimen you worked with. Your specimen is IGBP R 324. What does IGBP stand for? Please mention this in your list of abbreviations. Add all this information to a new chapter.
It looks like this work was written for a thesis and copied for publication. The manuscript would greatly benefit from shortening and adding illustrations of the muscle settings in the other groups used in the EPB, extant taxa, and reconstructions done by other authors. Overall, the work would profit from a much clearer separation of the EPB, extant, and extinct taxa.

Experimental design

The authors did extensive literature work for their study. The topic is very interesting, and the quite different reconstructions made in the past show that there are still many open questions. Plesiosaurs are the only group of Sauropterygia which survived the past the Triassic and roamed the oceans until the Late Cretaceous. Therefore, they are a very interesting research topic and I greatly support any research performed in this topic.
It is not clear to me how much of this study is based on own experimental work and how much on the comparison of existing literature. The use of histology sampled was mentioned in the introduction (line 303) but there was no further mention anywhere is the text. Was this discarded? And if so, why? The literature comparison can be easily checked by other colleagues and therefore this part of the work is very transparent. Unfortunately, IGBP R 324 itself remains quite unclear. See chapter before. Pictures of muscle scars and rugosities would have helped to make the drawn muscle positions clearer understandable. The Goldfuss Institute in Bonn owns a MicroCT scanner, have you thought about scanning the bones and doing a 3d reconstruction? Fine muscle scars and fine rugosities can be nicely made visible by these techniques… either by studying the slices themselves or by applying hard side light to untextured 3d scans. This would greatly help to support your positionings of each muscle.
In case of some reconstructed muscles, the muscle origin probably lies on the sacrum or caudal vertebrae (for example cfl and cfb). This is not shown in the figures. Why? Showing would be relevant for understanding and checking the geometrical derivation of the muscle functions you made.

Validity of the findings

There is some novelty in this study because the previous reconstructions differ so much from each other. A comparative work is therefore helpful and interesting for all researchers working on Sauropterygia. Unfortunately, no information about the true comparability between the previously studied plesiosaurs and C. eurymerus can be found. The authors state that the muscle setup (head counts etc) in crocodiles or turtles may vary within the group. Can this also be the case in plesiosaurs? There is no information about that.
A large amount of speculation can be found in the results, comparison, and conclusions. The authors generally made this clear by using terms like maybe, potentially etc. Personally, I find it difficult to reconstruct muscles with bad support by the EPB (for example addV, comments about the level of inference by the authors would have helped here) and add them to the final reconstruction without marking them as speculative in the figure.

Additional comments

Although the topic is very interesting, I recommend rejection because the manuscript is flawed and unlikely to be salvageable through revision. Think about rewriting and restructuring the whole manuscript and add illustrations and figures. Maybe think about splitting the monography into two manuscripts, maybe published as part I and II: Part one about the EPB and probably the comparison with extant taxa, part two about the comparison of your results with previous reconstructions.

Annotated reviews are not available for download in order to protect the identity of reviewers who chose to remain anonymous.

·

Basic reporting

Given the review order, I assume this section is the first thing you read, so I'd like to first say that this is a really great study and much needed for the field of marine reptile functional anatomy/morphology and I am eager to see it published!

With the Results section, I recognize that this is a comparative anatomy paper and it is very well done, but some sections read as a description of the analog’s anatomy moreso than Cryptoclidus' with its anatomy not mentioned until the last few sentences of each muscle heading. Instead of the way it is currently written where for example for the Musculus subcoracoscapularis, where you first delve into the anatomy of turtles, crocs and lizards and why, based on that, you have inferred its reconstruction on the posterodorsal proximal plesiosaur humerus, maybe start out with “we reconstruct the Musculus subcoracoscapularis on the posterodorsal proximal plesiosaur humerus…. based on….” and then go into the analogous/homologous comparisons.

In general, and I understand the reasoning, much of the paper reads more as a summary of the anatomy of analogs more than a study of the plesiosaur anatomy. I would do a decent amount of reorganization and some shortening of the extant anatomical description sections focusing very explicitly on what is relevant to the plesiosaur reconstructions and why. This will help shorten the paper.

The current figures are good, though I would not have abbreviations for the figures. However, I do think the results section could be greatly augmented with additional or improved figures highlighting the functions of the reconstructed muscles. Perhaps two large fore- and hindflipper figures with A, B, C, D sections highlighting static views of some key functional actions of the muscles. Though, I do really like the tables for this same purpose. Instead of, however, having “function after x study” in the columns maybe just “function” with the studies in parentheses or "function inferred from (study)." It would also be good to include a column of the assigned abbreviation of the muscle groups so the table can connect better to the main text. In Table 2 the third row says “function, this study” which differs from the language used in Table 1.

Additionally, and perhaps this is a personal preference, but due to the large usage of abbreviations and very numerous muscle groups in this study, the manuscript may flow better with abbreviations akin to what Cieri et al. 2020 utilized, for example abbreviating Coracobrachialis brevis as CoracoBrev. I found myself having to stop and go back to the abbreviations list quite a lot, as not all were intuitive. Additionally, definitely place the abbreviations section much earlier in the text as some abbreviations where used prior to when they were indicated (line 99).

Line 127 and 128, modern day paraxial swimmers may not have four similarly sized and shaped flippers, but this sentence makes it sound like in the extant realm it is either caudal flukes in axially swimming groups like cetaceans OR webbed paddle feet. Paddles are drag-based and there are of course flippered animals today which have more than webbing, rely on lift, and can barely/don't use them on land, including penguins and sea turtles which of course you know as this manuscript delves into them as analogs. Given that I find this sentence’s wording in not highlighting these animals as just a little odd given the content of the rest of the paper. Very easy fix though.

The Muscle Function section of the Discussion and the previous, lightly touched on muscle functions section of the Results create redundancy. Additionally, there are elements of the Discussion that might be included in the Results section. Overall there is a lot of length which could be shaved off by having these sections better complement one another and much would be gained with integrating them in a way that the discussion section doesn’t repeat inferred functions, origins, and insertions indicated in the Results section. The Discussion section might be improved by highlighting how this study differs or aligns with studies past and especially, given the title and presented problem, delving into how the reconstructed myology relates to flipper twisting and why that is important for plesiosaurs' locomotion (which you absolutely discussed throughout the manuscript, it's just not explicitly organized in its own section and highlighted). Again, this may entail taking elements of the Results and putting them into the Discussion and vice versa. All of these things are definitely already highlighted in the manuscript, it just needs organizing in a way that will make the findings more straightforward and less repetitive.

The plesiosaur joint morphology, osteology, and myology section (Line 2462) feels out of place in this current Discussion section. As the manuscript is right now, I believe that portions of this section would be better included with the previous section dedicated to the highlighted functional analogs prior to the Results as I find it distracts from the general functional and Cryptoclidus-focused ending of the manuscript.

There are quite a few grammar, flow, and textual issues throughout, with some examples highlighted by line below:

Line 46, strange wording with the rise of plesiosaurs and their extinction.
Line 55, I would merge these two sentences by eliminating the word today. It will also make the paper timeless. “Today” as the 2020’s won’t be very accurate when this paper is read in a decade or two. Something like “…have been debated until Krahl 2020….”
Line 65, extent not extend
Line 99, abbreviations are used prior to them being defined in-text or referenced, so I would move abbreviations to after the introduction and be sure not to abbreviate in the intro.
Line 124, Should this be “Functional analogues <of the> [plesiosaur] locomotory and musculoskeletal system”?
Line 291 and 292, again extent not extend, and repetitive use of “at least”
Line 293, sentence starting with “However” seems out of place. Perhaps meant to be at the end of two paragraphs previous?
Paragraph starting at 295, seems out of place for this section
Line 303, proof, not prove
Line 319, odd wording, I am not sure what this sentence means.
Line 323, replace sometimes with “may” or equivalent
Line 330, instead put “...who sampled and compared…” and remove “and also compared them.”
Line 411, make clearer what “the former” is in this case.
Line 415, I don't think "Therefore" is needed to start this sentence.
Line 563, don’t start sentence with “Decided.”
Line 1426, starts out as “pit was chosen” when other section had the full muscle name written out to start.
Line 1744, add a space after this paragraph.
Line 2181, cumbersome grammar

Citation of “Wittmer,” 1995 should be Witmer with one "t"

Experimental design

I believe this research to be within the aims and scope of PeerJ.

This is a wonderfully comprehensive study with a sound experimental design based on well-done comparative anatomy that fills a major gap in marine reptile functional morphology. Plesiosaurs as having no true analogs have stumped paleontologists for decades with regards to their locomotion and this study is a great push forward and logical continuation of/sequel to studies like Araújo and Correia, 2015 in understanding the myology which relates to how plesiosaur limbs would have moved in the context of potential swimming locomotory styles. The problem is well-defined and some moving around of elements in the manuscript's Results and Discussion section and new/modified sub-sections (recommended under the Basic Reporting and Validity of the Findings comments) would help to make clearer how the problem is addressed.

Some light critique is in some of the analogs. Though similar in function, otariids and cetaceans as mammals have critical differences in this pectoral girdle osteology and myology, lacking entire bones, the large cartilaginous extensions of reptiles, and entire muscles. For that reason, I wouldn’t put as much weight into the osteological and myological comparisons between plesiosaurs and these groups, but the functional strokes of the flippers, especially in otariids is still great comparison for plesiosaurs.

Line 474, “therefore it is presumed to have become superfluous.” In crocs the medial scapulosternal ligament is the originator of some forelimb muscles relevant to locomotion (Jasinoski et al. 2006). I understand the limitations of the reconstructions of this study, and not reconstructing the scapulosternal ligament for the purposes of this project is definitely fine, but I would just further highlight precisely why you believe this ligament to be functionally superfluous given its inferred course, especially given that you did use comparisons with crocs in other areas of the results section.

Line 2634, what are “the three” extant taxa? This paper highlights lepidosaurs, crocs, cetaceans, otariids, sea turtles, and penguins for comparisons.

The literature background used to inform the reconstructions is very solid as based on current phylogenetic bracketing. It may not be entirely harmful, however, to go slightly deeper into avian literature aside from just penguins given that flying birds are of course generating lift and lots of good functional anatomy work has been done on birds in the context of flying that may assist in informing some more obscure and ambiguous muscles and ligaments. Just a light suggestion or worth a passing mention in text.

Validity of the findings

I do find there to be some slight deviation and tangents from the stated purpose of the study at points within the manuscript. Again, there is some dwelling on the taxa used for comparisons. I interpret this paper as being dedicated to making robust muscle reconstructions of the fore- and hindflippers in plesiosaurs with the goal of determining what the underlying myological mechanisms for flipper twisting and general movement in plesiosaurs are. As taken from the abstract:
“The question whether plesiosaurs employed their four flippers in underwater flight, rowing flight, or rowing has not been settled yet. Plesiosaur locomotory muscles have been reconstructed in the past, but neither the pelvic muscles nor the distal fore- and hindflipper musculature have been reconstructed entirely.”
The muscle reconstructions and the inferred functions are excellent and I think the manuscript clearly addresses the second sentence. I feel as though neither the discussion nor conclusion take these functions and infer precisely what type of swimming these functions would have enabled and address the first sentence directly. I think this could be welcome speculation given the robust results. The word "flight" is used only twice in the discussion section, once used in the context of previous studies, the second used in the context of sea turtles. Similar for use of the word "rowing" used only in the context of analogs. The fourth paragraph of the Conclusion comes close to this, but does not explicitly define this as a lift or drag-based stroke. Lift is greatly inferred, but should be made explicitly clear.


It may be worth adding a sentence or two on encouragement of using the myological reconstructions within this study to further assess plesiosaur swimming function with hydrodynamic and biomechanical methods. To fit the journal's needs.

I would restructure the conclusion to particularly how the myological reconstructions of this study inform the flipper twisting and beat cycle. The conclusion should also restate the problem at hand. Why did you do this study? Currently it only summarizes the results, it does not really *conclude*. Why do this? How did you do this? What were your results? How did they differ from/complement previous studies? How would you encourage future work in this realm?

Additional comments

I really commend the authors for this undertaking that will no doubt expand our knowledge and fill a large hole in the field of Mesozoic marine reptile functional morphology. The reconstructions, functional and anatomical comparisons, as well as inferred beat cycle are sound and quality, and I am excited to have reviewed this and for it to be published. The manuscript, however, does have a fair bit of pacing, flow, length, repetitiveness, and grammar issues which hopefully I provided decent constructive critique on.

Line 2377, I believe this section is really the crux of the paper, what the results lead to, and it should be brought out more. Currently it just feels like a generic section especially given the sort of out of place section which follows it. The Muscle Function section in general should be the "main course" of the Discussion section, but currently feels a bit buried overall. It may also be good to slightly pull back on the definitiveness of this section (Line 2377), though I do not disagree with the assertions. Instead of "At the beginning of the downstroke the flipper leading edge is rotated..." perhaps something along the lines of, "Our reconstructions suggest that at the beginning of the downstroke the flipper leading edge is rotated by..." It would additionally be helpful to restate why it is important that plesiosaurs were able to twist their flippers in the context of their currently estimated locomotion style, and then tie in the results to that problem in estimating a plesiosaur fore- and hindflipper stroke. Additionally, definitely restate the ~19 degrees rotation based on Witzel, Krahl & Sander, 2015 and Witzel, 2020 for the foreflipper in this section. It was mentioned very early on in the manuscript, then for the hindflipper only in this section, then restated in the conclusion section as for both.

---

## Round 0.2 · Major Revisions

Two of the three reviewers have given new feedback which will help improve the manuscript, but substantial revisions are needed to satisfy one of them. Some rewording/structuring, extra figures, and added referencing are necessary. But more importantly, the concern about circularity must be addressed carefully and openly. The EPB usage should be explored more transparently and homology inferences too; I am not so convinced though that innervation and development are better evidence of homology (i.e. that this is consensus; another view is that innervation is just another kind of topology, and development is a different level of evidence about homology) but the debate should be acknowledged. 1 reviewer will need to re-check the revised MS after I do. Thank you.

Reviewer 1 ·

Basic reporting

The clarity of the manuscript is much improved; it is focused and organized around a central hypothesis. The structure is strong and the figures are very good, although some information is still missing from the figures and text. A few additional references are needed. The text is still very long and some portions could be reduced.

1. The organization of the manuscript is logical, but some text could be removed or reduced to make it more concise. For example:
- In the introduction, the significance of the paragraphs on girdle homology is not clear to me (lines 160-188)
- In the results section, each muscle description begins with a list of synonyms and a justification for which term is used in the current ms. This could be replaced by a single statement that the current work uses the term most commonly used for that muscle in the recent sauropsid literature and only the exceptions discussed.
- in the Discussion section there are several paragraphs that comprehensively list previous studies (2027-2046, 2142-2148). I think these are unnecessary because this is not a review paper.
- the Conclusions section reiterates the Discussion section

2. References for muscle homology are insufficient; at the very least Walker (1973) should be listed in this section (199-205) and not just as an afterthought (249). Also see, e.g., Hattori, Soki, and Takanobu Tsuihiji. “Homology and Osteological Correlates of Pedal Muscles among Extant Sauropsids.” Journal of Anatomy 238, no. 2 (2021): 365–99 and Rui Diogo’s work on muscle homology in turtles and lepidosaurs.

3. The figures are very good. However, some important information is not figured, namely the muscle scars referred to in the text (e.g., 566-567, 608, 761-762). This is important because these scars influenced the muscle reconstructions, so they should be figured or a reference provided. In addition, some of the figure captions are extremely long and should be reduced (Figs. 4, 5). Fig. 1c and d should include Sphenodon because it is also used for phylogenetic inference. Finally, in Fig. 4b the humeral head of triceps brachii appears to be missing.

4. Some key background information is lacking from the introduction:
- Lines 45-46: What are the relationships between Plesiosauria, Sauropterygia, and Diapsida? (could be added to Fig. 1)
- Lines 151-156: Why was this species of plesiosaur chosen as a basis for the reconstruction, and what are its relevant anatomical features and phylogenetic relationships within the group?

Experimental design

The research question is well-defined and fills a knowledge gap. The methods are described in detail, but some methods were not rigorously followed and/or explained (establishment of muscle homologies and EPB).

1. Most importantly, the method the authors refer to as “EPB” appears to be simply inferring the condition found in the majority of the 3 extant taxa studied. EPB as defined by Witmer (1995) and Bryant and Russell (1992) is a method of phylogenetic inference that follows a series of steps, taking into account the precise relationship of taxa and the presence of osteological correlates throughout the fossil record of the group. So, for example, the presence of a trait in both crocodylia and testudines is weaker evidence for its presence in sauropterygia than its presence in crocodylia and lepidosauria because lepidosauria is the sister group of either sauropterygia + archosauria (Fig. 1b) or of sauropterygia (fig. 1c). Please clarify the methods used here in the context of the literature and justify differences in methods. Throughout the results section the relationship between extant taxa should be considered more carefully and the most parsimonious inference (not necessarily the most common character state) should be given. (I doubt that this will affect the results noticeably, but it should still be considered.)

2. Secondarily, the methods described for determining muscle homology are vague and insufficient. It is implied that most of the muscle homologies are proposed here for the first time (lines 204-205), but Remes 2007 is cited heavily as well. Furthermore, it is stated that homology was established by topology, whereas innervation and development are generally considered to be the most important criteria for muscle homology (e.g., Russell & Bauer 2008, Walker 1973). This section should be reworked with a comprehensive discussion of a) what parts are original to this study, b) the basis for the homologies where original to this study, including innervation patterns, and c) any controversies and their potential effects on the reconstruction.

3. I don’t understand the logic that chameleons are functionally comparable to plesiosaurs (lines 631-632). Their musculature and locomotion are highly specialized for grasping and perching.

Validity of the findings

The results add to the literature by providing a more comprehensive, phylogenetically based muscle reconstruction in plesiosaurs. However, there seems to be some bias introduced into the methods that undermines the conclusions. Also, it is not always clear when something is an inference or speculation and what it is based on, and the logic behind reconstruction decisions is not always explained.

1. Most importantly, there appears to be some circular reasoning leading to the conclusion that plesiosaurs possessed muscles that could enable flipper twisting and thus underwater flight. For example:
- Lines 315-320: “we tried to find muscles that would enable underwater flight ... some muscle attachment areas can be supported by two of the extant sauropsid groups by the EPB, but favorable for flipper twisting is a muscle attachment area which is only supported by one” (these cases are noted in the results but still used in support of the overall conclusion)
- Lines 412-416: “Due to the flipper shape … plesiosaurs were most likely relying only on lift-based locomotion (underwater flight) … therefore, muscles employed in individual digital movement are not reconstructed”
- Line 1042: “a lepidosaur-like insertion to metacarpal I is functionally favorable as it contributes to flipper twisting”
I strongly suggest not making any reconstruction decisions based on functional hypotheses (especially the one you're trying to test), but at least make it absolutely clear how these decisions affected the result and consider the alternatives.

2. Be careful with language when discussing function in extinct animals; it is always uncertain. E.g., lines 342-343 “The whole flipper is rotated by approximately 19 degrees by flipper rotators and twisted by flipper twisting muscles (refs).” This is a hypothesis based on simulations. Line 1965: Before describing the hypothesized muscle functions it would be good to summarize what the hypotheses are based on.

3. The logic behind some decisions is not clear: Lines 910-911: “The state of edc as represented by turtles poses the plesiomorphic condition” – why? Lines 1937-1940: why is m. adductor digiti quinti reconstructed in plesiosaurs when it is only present in lepidosaurs?

Additional comments

In this revision, the authors have reorganized the manuscript and greatly improved its clarity. Several excellent figures have been added. I still have some questions about the methods as they have taken an unconventional approach to muscle homology and EPB, and I'm concerned that some decisions about the methodology may have been influenced by the result the authors were hoping to find. While these are serious issues, their effect on the results is most likely minor. Therefore, I see this version as being much closer to publication.

Reviewer 2 ·

Basic reporting

The manuscript was greatly improved by the authors. Grammar and spelling were also greatly improved and, except from a few typos, I see absolutely no problems in publishing the manuscript like this. The restructuring done by the authors helped a lot in improving the overall readability and clarity of the manuscript. The newly added figures provide valuable information about the lines of action of the reconstructed muscles and are therefore a very good addition. The results are well presented. I’d recommend moving the description of the complete flipper beat cycle to a new paragraph inside the conclusions to highlight this relevant information a bit more.

Experimental design

The authors addressed my concerns really well. Thank you for that! Everything is fine now.

Validity of the findings

Thank you for the explanations. The manuscript profited from the explanations added by the authors and the previously existing uncertainties were addressed and removed.

Additional comments

You cite Krahl (2021) in several places in the manuscript. This citation doesn’t appear in the reference list. In some cases, there is also a publication named Krahl (2021, in review). Is it the same? If so, please correct or add the publication the reference list.
In my opinion, the manuscript is totally publishable now. The few typos in the text and the one in Fig 2 b can be easily corrected. This also applies for the very few mistakes in the reference list and one wrong reference to a Figure (Fig 3 instead of Fig 4). See comments in the pdf.
Thank you for investing so much work again. The manuscript was greatly enhanced.

Annotated reviews are not available for download in order to protect the identity of reviewers who chose to remain anonymous.

---

## Round 0.3 · Minor Revisions

Very minor changes are needed and that is all -- congratulations!!

Reviewer 1 ·

Basic reporting

I'm happy with the current version, and I defer to the authors on style issues. The figure revisions look good, and I appreciate the paragraph on Cryptoclidus. 3 minor questions/comments:
Line 45. What is meant by "most derived"?
220: Where are homology lists with citations - does this refer to table 3?
Results section: I meant to suggest that the phrase "The term ____ was chosen because it is the most commonly used one in recent works on Sauropsida" only needs to be used once in this section, e.g., "we use the muscle terminology most common in recent works on Sauropsida" in the beginning of results, and then don't mention muscle terminology again unless you're choosing something different.

Experimental design

no comment

Validity of the findings

no comment

Additional comments

The authors have fully addressed my concerns. I appreciate their careful explanations and revisions, and I now am confident that this study has been rigorously carried out and well reported.

---

## Round 0.4 · accepted · Accept

I have checked the changes and they are very well handled. Congratulations-- you're done! And a good job persevering, conducting quite extensive revisions that did help improve the manuscript.